

**Oligomerization Reactions of Criegee Intermediates with**
**Hydroxyalkyl Hydroperoxides: Mechanism, Kinetics, and**
**Structure-Reactivity Relationship**
Long Chen, [1,2] Yu Huang, *[1,2] Yonggang Xue, [1,2] Zhenxing Shen,[3] Junji Cao, *[1,2] Wenliang Wang[4]
[1] *Key Lab of Aerosol Chemistry & Physics, Institute of Earth Environment, Chinese*
*Academy of Sciences, Xi'an, Shaanxi, 710061, China*
[2] *State Key Laboratory of Loess and Quaternary Geology, Institute of Earth*
*Environment, Chinese Academy of Sciences, Xi'an 710061, China*
[3] *Department of Environmental Sciences and Engineering, Xi'an Jiaotong University,*
*Xi'an, 710049, China*
[4] *School of Chemistry and Chemical Engineering, Key Laboratory for*
*Macromolecular Science of Shaanxi Province, Shaanxi Normal University, Xi'an,*
*Shaanxi, 710119, China*

21 *Corresponding author:

22 Prof. Yu Huang, E-mail address: huangyu@ieecas.cn

23 Prof. Junji Cao, E-mail address: cao@loess.llqg.ac.cn



## ABSTRACT:

Although secondary organic aerosols (SOAs) are major components of $PM_{2.5}$ and organic aerosol (OA) particles and therefore profoundly influencing air quality, climate forcing and human health, the mechanism of SOAs formation via Criegee chemistry is poorly understood. Herein, we perform high-level theoretical calculations to study the reactivity and kinetics of four Criegee intermediates (CIs) reactions with four hydroxyalkyl hydroperoxides (HHPs) for the first time. The calculated results show that the sequential addition of CIs to HHPs affords oligomers containing CIs as chain units. The addition of -OOH group in HHPs to the central carbon atom of CIs is identified as the most energetically favorable channel, with a barrier height strongly dependent on both, CI substituent number (one or two) and position (*syn-* or *anti-*). In particular, the introduction of a methyl group into the *anti-*position significantly increase the rate coefficient, dramatic decrease is observed when the methyl group is introduced into the *syn-*position. Based on the collected data, the atmospheric lifetime of *anti-*$CH_3CHOO$ in the presence of HHPs is estimated as ~$5.9 \times 10^3$ s. These findings are expected to broaden the reactivity profile and deepen our understanding of atmospheric SOAs formation processes.



## 1. Introduction


Alkenes are the most abundant volatile organic compounds (VOCs) in the
atmosphere after methane and primarily originate from anthropogenic and biogenic
sources (Lester et al., 2018). Gas-phase ozonolysis of volatile alkenes is extremely
exothermic, and the potential energy surface (PES) following the 1,3-cycloaddition of
ozone to the C=C double bond forming a primary ozonide (POZ) is riddled with
shallow wells and low barriers (Donahue et al., 2011; Aplincourt et al., 2000), which
then dissociate to produce a carbonyl oxide (also called Criegee intermediates (CIs))
and a carbonyl moiety (Johnson et al., 2008; Welz et al., 2012; Criegee, 1975). Alkene
ozonolysis is thought to be an important source of radicals, whose subsequent
reactions lead to the formation of hydroperoxides, organic peroxides, and secondary
organic aerosols (SOAs) (Donahue et al., 2011; Becker et al., 1990; Kroll et al., 2008;
Hallquist et al., 2009; Tobias et al., 2001), and thus influence air quality, climate
forcing and human health (Rissanen et al., 2014; Donahue et al., 2011; 2012). Criegee
intermediates were first proposed by Rudolph Criegee as early as 1975 (Criegee,
1975), and their direct synthesis in the laboratory experiment were performed by the
photolysis of organic iodides in the presence of $O_2$ and the photolytic Cl-initiated
oxidation of dimethyl sulfoxide (DMSO) (Welz et al., 2012; Taatjes et al., 2008).
In view of the highly exothermic nature of alkene ozonolysis, the nascent CIs
often possess a considerable amount of internal energy and are thus highly reactive
(Li et al., 2018; Li et al., 2014). The thermal unimolecular decay of vibrationally
excited CIs is thought to be an important nonphotolytic source of atmospheric
hydroxyl (OH) radicals, particularly in low light conditions, urban environments, and
heavily forested areas (Lester et al., 2018; Foreman et al., 2016; Kidwell et al., 2016;
Green et al., 2017; Zhang et al., 2002). OH radical is one of the most powerful
oxidants that participates in the atmospheric photochemical oxidation of VOCs
(Gligorovski et al., 2015), and thus contributes to tropospheric ozone formation by
being involved in the production of organic peroxy radicals ($RO_2$), which, in turn,
facilitate the cycling of NO to $NO_2$ (Zhang et al., 2002; Gligorovski et al., 2015). The





remaining CIs become collisionally stabilized Criegee intermediates (SCIs) that can
undergo further bimolecular reactions with a number of atmospheric trace gases, such
as $H_2O$, $NO_2$, and $SO_2$ (Chen et al., 2016a,b; Mauldin et al., 2012; Berndt et al., 2014;
Kuwata et al., 2015; Lin et al., 2016; Ouyang et al., 2013; Stone et al., 2014; Chao et
al., 2015; Lin et al., 2017; Taatjes et al., 2017; Long et al., 2018), and contribute to the
nucleation and growth of secondary aerosol (e.g. nitrate, sulfate, SOAs) by
partitioning between gas and particle phases (Foreman et al., 2016; Vereecken et al.,
2017; Huang et al., 2015; Berndt et al., 2012; Zhang et al., 2015; Huang et al., 2014;
Li et al., 2018; Ji et al., 2017; Xu et al., 2014). The bimolecular processes of SCIs at
the air/water interface have been extensively studied both experimentally and
theoretically (Zhu et al., 2016; Kumar et al., 2017, 2018; Zhong et al., 2017, 2018;
Enami et al., 2017; Heine et al., 2017), and the reaction with atmosphere-abundant
water vapour in the gas phase or at the air/water interface has been identified as one of
the dominant degradation pathways of SCIs removal from the atmosphere (Chen et al.,
2016a,b; Lin et al., 2016; Chao et al., 2015; Huang et al., 2015; Zhu et al., 2016;
Zhong et al., 2017; Zhang et al., 2012; Taatjes et al., 2013; Anglada et al., 2016).
Experimentally, Su et al. (2013) investigated the transient infrared absorption
spectrum of $CH_2OO$ using a step-scan Fourier-transform spectrometer, and observed
that the vibrational frequencies are more consistent with a zwitterion rather than a
diradical structure. Taatjes et al. (2013) studied the kinetics of $CH_3CHOO$ reactions
with $H_2O$, $SO_2$, and $NO_2$, and found that *anti*-$CH_3CHOO$ is substantially more
reactive toward water and $SO_2$ than is *syn*-$CH_3CHOO$ with an upper limit rate
coefficient $(1.0 \pm 0.4) \times 10^{-14}$ $cm^3$ molecule$^{-1}$ s$^{-1}$. Also Smith et al. (2015) reached
similar conclusions in their UV absorption of the $CH_2OO + H_2O$ reaction system that
the rate coefficient is determined as $(7.4 \pm 0.6) \times 10^{-12}$ $cm^3$ molecule$^{-1}$ s$^{-1}$ at 298 K,
and it exhibits a large negative T-dependence at temperatures from 283 to 324 K.
Moreover, oligomerization reactions of CIs with typical atmospheric species are
identified as one of the dominate pathways leading to the formation of highly
oxygenated and high-molecular-weight oligomers that have remarkably low vapor
pressure contributing to SOAs formation and growth (Bonn et al., 2008; Heaton et al.,



2007; Wang et al., 2016; Inomata et al., 2014). For example, Sakamoto et al. (2013)
performed laboratory-scale ethylene ozonolysis in a Teflon bag reactor, and revealed
that the sequential addition of $CH_2OO$ to hydroperoxides leads to oligomeric
hydroperoxides and finally affords SOAs. Sadezky et al. (2008) proposed that SOAs
formation is initiated by the reaction of SCI with a $RO_2$ radical, followed by the
sequential addition of SCIs, and chain termination by reaction with $HO_2$ radical.
$$RO_2 + SCI \rightarrow RO_2\text{-}SCI \qquad (1)$$
$$RO_2\text{-}SCI + (n\text{-}1)SCI \rightarrow RO_2\text{-}(SCI)_{n\text{-}1}\text{-}SCI \qquad (2)$$
$$RO_2\text{-}(SCI)_{n\text{-}1}\text{-}SCI + HO_2 \rightarrow RO_2\text{-}(SCI)_n\text{-}H + O_2 \qquad (3)$$
Zhao et al. (2015) studied the ozonolysis of *trans*-3-hexene in a flow reactor and
static chambers in the absence and presence of an OH or SCI scavenger at $295 \pm 1$ K,
arriving at the same conclusion as above. In particular, oligomers having SCIs as
chain units were identified as one of the dominant components of atmospheric SOAs
and were produced by the sequential addition of $C_2H_5CHOO$ to $RO_2$ radical. More
recently, Wang et al. (2016) investigated the heterogeneous ozonolysis of oleic acid
(OL) using an aerosol flow tube, and found that reactions of particulate SCIs generate
high-molecular-weight oligomers, with low volatility that are preferentially
partitioned into the particle phase to promote SOAs formation. They confirmed that
the SCI-based mechanism is dominant pathway in the formation of
high-molecular-weight oligomers.
On the other hand, Ehn et al. (2014) reported a large source of low-volatility
SOAs generated from the ozonolysis of α-pinene and other endocyclic monoterpenes
under atmospheric conditions, and proposed that the mechanism of extremely
low-volatility organic compounds (ELVOCs) formation is driven by $RO_2$ autoxidation.
Also several groups reached similar conclusions that the highly oxygenated molecules
(HOM) are produced via $RO_2$ autoxidation in the cyclohexene and terpenes
ozonolysis systems (Rissanen et al., 2014; Kirkby et al., 2016; Berndt et al., 2018).
Moreover, HOM are major contributors to aerosol particle formation and growth on a
global scale (Tröstl et al., 2016; Stolzenburg et al., 2018). Compared with the $RO_2$



autoxidation pathways, oligomerization reactions involving CIs, preserve carbon
oxidation state and increase carbon number, and therefore lead to a large reduction in
volatility (Wang et al., 2016). Moreover, oligomerization reactions accompany with
the shorter time period during the early stage of SOAs growth (Heaton et al., 2007).
Therefore, we think that it is essential to investigate the Criegee chemistry-based
mechanism of SOAs formation and growth.

Aplincourt et al. (2000) investigated the mechanism of $CH_2OO$ reactions with

$CH_2O$, $H_2O$, $SO_2$, and $CO_2$ at the CCSD(T) level of theory, and found that the
reactions with $H_2O$, $CH_2O$, and $SO_2$ are preferable, whereas that with $CO_2$ is unlikely
to occur. Because of the high concentration of $H_2O$ ($[H_2O] \approx 7.0 \times 10^{17}$
molecules $cm^{-3}$) under atmospheric conditions (Zhang et al., 2014; Zhang et al., 2015),
the reaction with water vapour is the dominant chemical sink (Aplincourt et al., 2000).
Also Ryzhkov et al. (2003; 2004; 2006) reached similar conclusions that the most
energetically favourable pathway of carbonyl oxide reaction with the water vapour is
the formation of hydroxyalkyl hydroperoxide. Zhao et al. (2017) investigated the
mechanisms and kinetics of four SCIs reactions with four $RO_2$ radicals, and found that
the addition of terminal oxygen in $RO_2$ to central carbon in SCI is the most kinetically
favourable channel. Unfortunately, there is very little study to do on the reactivity of
SCIs toward hydroxyalkyl hydroperoxides (HHPs) generated from the reaction with
water vapour. Moreover, the effect of substituents on the reactivity of carbonyl oxides
is still poorly understood.

Recently, Vereecken et al. and Anglada et al. tried to mechanistically characterize

the reaction of $CH_2OO$ with $CH_3OO$ radical using different quantum chemistry
methods (Vereecken et al., 2012; Anglada et al., 2013), and revealed that the above
reaction initially proceeds via the formation of a strong pre-reactive complex followed
by a submerged electronic-energy barrier for the subsequent addition of the $CH_3OO$
terminal oxygen atom to the $CH_2OO$ central carbon atom. An analogous conclusion
was obtained by investigating the reactions of *anti*-$CH_3CHOO$ with $HO_2$ and $H_2O_2$
molecules, i.e., the sequential addition of SCIs is a favorable reaction mode for SOAs
formation (Chen et al., 2017). Vereecken et al. (2017) also investigated the reactions





of CH$_2$OO with acids and enols using the CCSD(T)//M06-2X/aug-cc-pVTZ method,
and found that the 1,4-insertion mechanism allows for barrierless reactions with high
rate coefficients. The above milestone investigations provide fundamental insights
and lay solid foundations for further studies of the Criegee chemistry-based
mechanism of SOAs formation.

In this study, we mainly focus on the oligomerization reaction of carbonyl oxides

with HHPs leading to the formation of high-molecular-weight oligomers under
atmospheric conditions. This reaction represent the initial step of oligomer formation
and growth during alkene ozonolysis, and therefore need to be extensively
characterized to gain deeper insights into the fundamental chemical composition of
these oligomers in the atmosphere. Moreover, structure-reactivity relationship plays
an important role in determining the rates and outcomes of bimolecular processes.
Herein, we employ high-level theoretical calculations in conjunction with kinetics
analysis to study the mechanism and kinetics of the reactions of four carbonyl oxides
with four HHPs, and describe the effects of carbonyl oxide conformation on reaction
rate. The carbonyl oxides considered in this work (CH$_2$OO, *syn-/anti*-CH$_3$CHOO, and
(CH$_3$)$_2$COO) are anticipated upon ozonolysis of ethylene, propylene, isobutene and
2,3-dimethyl-2-butene, while the investigated HHPs are assumed to arise from
bimolecular reactions with water vapour in the troposphere.

## 2. Computational details

The geometries of all stationary points on PES are optimized and characterized

by the M06-2X functional (Zhao et al., 2006) in combination with the
6-311+G(2df,2p) basis set (Zheng et al., 2009), since the M06-2X functional allows
one to reliably compute the energies and stability of non-covalent interactions (Zhao
et al., 2008a,b). Harmonic vibrational frequencies are performed at the same level of
theory to verify that the nature of each structure is either a minimum (NIMAG = 0) or
a transition state (NIMAG = 1) and to provide the zero point vibrational energy
(ZPVE) corrections. A scale factor of 0.98 is applied to scale all the
M06-2X/6-311+G(2df,2p) frequencies to account for the thermodynamic contribution



to the Gibbs free energy and enthalpy at 298 K and 1 atm (Zheng et al., 2009). The
reactant-product connectivity on either side is established by intrinsic reaction
coordinate (IRC) calculations (Fukui, 1981).

The barrier heights of some elementary reactions are calculated for single-point

energies Y/X (Y = M06-2X, CCSD(T), X = 6-311+G(2df,2p), def2-TZVP)
determined using M06-2X/6-311+G(2df,2p) optimized geometries. The obtained
results (Table S1) indicate that the largest deviations of electronic-energy ($\Delta E_a^{\#}$) and
free-energy ($\Delta G_a^{\#}$) barriers between CCSD(T)/6-311+G(2df,2p) and
M06-2X/def2-TZVP methods are 1.6 and 1.5 kcal·mol$^{-1}$, respectively, while the
respective mean absolute deviations (MAD) are 0.99 and 0.95 kcal·mol$^{-1}$. Thus, the
M06-2X method afford energies similar to those determined by the accurate and well
recognized CCSD(T) level calculation. Considering the computational costs, the
M06-2X/def2-TZVP method is selected to perform the single-point energy calculation
for the title reaction system. The rate coefficients are calculated using canonical
transition state theory with quantum mechanical tunneling (Eckart) at temperatures
relevant in the troposphere (273-400 K) in the high-pressure limit (Zhao et al., 2017;
Chen et al., 2017).
$$k^{\text{TST}}(T) = \sigma \frac{k_{\text{b}}T}{h} (\frac{RT}{P^0})^{\Delta n} \exp\left(\frac{-\Delta G^{\dagger}(T)}{k_{\text{b}}T}\right) \qquad (4)$$

where $\Delta G^{\dagger}(T)$ is activation Gibbs free energy, $\sigma$ is reaction symmetry number, $k_{\text{b}}$

is Boltzmann's constant, $T$ is the temperature in Kelvin, h is Planck's constant, and
$\Delta n$= 0 and 1 for unimolecular and bimolecular reactions, respectively (Zhao et al.,
2017). The quantum chemical calculations are executed using the Gaussian 09
program suite (Frisch et al., 2009). The rate coefficients are calculated by
implementing the KiSThelP program (Canneaux et al., 2014).
**3. Results and discussion**
**3.1 Bimolecular reaction of SCIs with water vapour**

Equations (5) and (6) represent the two types of bimolecular reactions between

carbonyl oxides and water vapour. Previous investigations have shown that some



carbonyl oxides are largely removed by their reactions with water dimer (Chen et al.,
2016a,b; Chao et al., 2015; Taatjes et al., 2013; Anglada et al., 2016 ) to generate
HHPs (Chen et al., 2016a,b; Anglada et al., 2011), which are important atmospheric
oxidants initiating vegetation damage (Becker et al., 1990). Further mechanistic
details of the above reaction can be found in our previous works (Chen et al., 2016a,b;

2018).

$$R_1R_2COO \;+\; H_2O \;\rightarrow\; R_1R_2C(OH)OOH \tag{5}$$
$$R_1R_2COO \;+\; (H_2O)_2 \;\rightarrow\; R_1R_2C(OH)OOH \;+\; H_2O \tag{6}$$
Figure 1 presents a simplified scheme for the reactions of several distinct
carbonyl oxides ($CH_2OO$, *syn-/anti*-$CH_3CHOO$, $(CH_3)_2COO$) with water dimer to
form HHPs. In all cases, each reaction begins with the formation of a strong
pre-reactive complex and then surmounts a small barrier that is still lower in energy
than the reactants before product generation. Table 1 contains the relative energies of
stationary points and the activation energies of elementary reactions. In Figure 1 and
Table 1, labels A, B, C, and D correspond to the relative energies of the pre-reactive
complex (RC), transition state (TS), post-reactive complex (PC) and the product (P).
R1 and R2 denote *syn-* and *anti*-positions of the substituent, respectively. These four
transition states are located by rotation of two dihedral angles (DO2H4O4H3,
DO4H2O3H1). Based on the energies given in Table 1, products **Pa** and **Pb** are
near-isoenergetic conformers differing only in the orientation of the H1 atom along
the C1-O3 bond. As has been mentioned above, HHPs are key reactive intermediates
that possess with -OH and -OOH functional groups, and can therefore sequentially
react with carbonyl oxide to generate oligomers. Considering the fact that Pa and Pb
are structurally and energetically similar, the former is judiciously selected for
studying oligomerization reactions, whereas the latter is merely listed in the Figures
S1-S3.
**3.2 PES for the reaction of $CH_2OO$ with $HO-CH_2OO-H$**
$CH_2OO$, the simplest Criegee intermediate, originates from the ozonolysis of all
exocyclic alkenes, e.g., isoprene, monoterpenes, and sesquiterpenes (Nguyen et al.,



2016), which makes its chemistry particular important for forest and urban
environments. The largest sink of $CH_2OO$ corresponds to its bimolecular reaction
with water dimer in the troposphere, which generates $HO-CH_2OO-H$ as the dominant
product (Lewis et al., 2015; Kumar et al., 2014). Figure 2 shows the schematic PES
for the reaction of $CH_2OO$ with $HO-CH_2OO-H$, with the optimized geometries of all
stationary points on this PES given in Figure S4.
Figure 2 shows that the differences between the relative free energies and the
electronic energies for all stationary points are significant (~ 10-24 kcal mol$^{-1}$),
implying that the addition reactions of the parent carbonyl oxide with $Pa_1$ are
characterized by obvious contributions of entropy effect. Similar behaviors are also
observed for oligomerization reactions of other carbonyl oxides with HHPs (see
Figures 3-5). Thus, unless otherwise stated, the discussion in the following sections
refers to free-energy barriers ($\Delta G_a^{\#}$).
The formation of oligomers P2a, P2b, P2c and P2d (containing $CH_2OO$ as the
repeating unit) is strongly exothermic (>104 kcal mol$^{-1}$), and the apparent activation
energies $E_{app}$ observed for all elementary reactions are negative values, signifying that
these reactions are both thermochemically and dynamically feasible under
atmospheric condition. Product $Pa_1$ ($HO-CH_2OO-H$) formed in the reaction of $CH_2OO$
with water dimer has two functional groups (-OH and -OOH), both of which can be
involved in addition reactions. The addition reactions of $2CH_2OO + Pa_1$ begin with
the formation of loosely bound pre-reactive complexes IM1a and IM1b, of -3.5 and
-3.1 kcal mol$^{-1}$ stability. They are formed by a hydrogen bond between the terminal
$CH_2OO$ oxygen atom and the hydrogen atom of the -OOH group in $Pa_1$, and a van der
Waals (vdW) bond between the central carbon atom of $CH_2OO$ and the oxygen atom
of the -OH group in $Pa_1$. The above complexes are immediately converted into
products P1a and P1b via transition states TS1a and TS1b with barriers of 8.8 and
12.2 kcal mol$^{-1}$, respectively, while the corresponding reaction exothermicities are
estimated as 43.4 and 40.5 kcal mol$^{-1}$, respectively. The above result shows that the
most favorable channel is the addition of the -OOH group of $Pa_1$ to the parent
carbonyl oxide. The detailed mechanism mainly involves that the $HO-CH_2OO$ moiety





released from the breaking O-H bond in $Pa_1$ binds to the central carbon atom of
$CH_2OO$, and simultaneously the remnant hydrogen atom transfers to the terminal
oxygen leading to product P1a.

The addition pathway opens the door for other subsequent reactions leading to

SOAs via Criegee chemistry, which may result in aerosol formation and thus impact
climate. As pointed out in previous studies (Chen et al., 2016a,b; Kumar et al., 2014),
the thermal unimolecular decay of $Pa_1$ can occur via two competitive pathways,
namely (i) $HO\text{-}CH_2OO\text{-}H \rightarrow CH_2O + H_2O_2$  and (ii) $HO\text{-}CH_2OO\text{-}H \rightarrow HCOOH +$
$H_2O$. However, since the corresponding barriers are much higher than that of the
bimolecular reaction with $CH_2OO$ (~35 kcal $mol^{-1}$), the thermal unimolecular decay
of HHPs is not taken into consideration in this work.

The secondary addition reaction $CH_2OO$ + P1a is equivalent to that of $CH_2OO$ +

$Pa_1$ reaction, and hence features an analogous pathway, i.e., the formation of
pre-reactive complexes IM2a and IM2b in entrance channels, is followed by the
addition of -OH and -OOH groups of P1a to the $CH_2OO$ central carbon atom to
produce P2a and P2b. The barrier heights predict TS2a and TS2b to lie -36.3 and
-35.9 kcal $mol^{-1}$, respectively, below the energies of the separate reactants, and 7.5
and 7.4 kcal $mol^{-1}$ above the energies of the corresponding pre-reactive complexes
IM2a and IM2b. The above result shows that these two addition reactions (R2a and
R2b) equally contribute to the title reaction system. Compared to the first $CH_2OO$
addition reaction, the second one features a lower barrier. Finally, the addition
reaction $CH_2OO$ + P1b proceeds via mechanism fairly similar to those described
above for the $CH_2OO$ + P1a system and do not discussing in detail to avoid
redundancy.

## 3.3 PES for the reaction of $CH_3CHOO$ with $HO\text{-}CH_3CHOO\text{-}H$

The methyl-substituted parent Criegee intermediate can exist in two

conformations, *syn-* and *anti-*$CH_3CHOO$, depending on whether the methyl group is
located on the same or opposite side of the terminal oxygen (Yin et al., 2017).
Numerous theoretical studies have proven that the presence of an intramolecular



hydrogen bond in the *syn*-conformer makes it more stable than the *anti*-conformer
(Anglada et al., 2011; 2016). The interconversion of these two conformers via rotation
around the C-O bond has a very high barrier (~ 42 kcal mol$^{-1}$), which implies that one
can treat *syn*- and *anti*-CH$_3$CHOO as independent species existing in the atmosphere
(Yin et al., 2017). It is well known that the predominant pathway of unimolecular
reaction of *syn*-CH$_3$CHOO is isomerization to vinyl hydroperoxide (VHP) via the
hydrogen atom transfer, whereas the preferable route of unimolecular reaction of
*anti*-CH$_3$CHOO is ring-closure to dioxirane via an oxygen atom transfer (Donahue et
al., 2011; Taatjes et al., 2013; Long et al., 2016). Both of the prompt and thermal
unimolecular decay of the energized VHP may dissociate to OH radicals, and their
yields are strongly pressure and temperature dependents (Kroll et al., 2011a,b). The
dioxirane can finally isomerize to acetic acid via the "hot acid" channel (Kroll et al.,
2011a,b).
Long et al. (2016) proposed that the enthalpic barrier of *syn*-CH$_3$CHOO
isomerization to VHP is more than ~3 kcal higher than that of the addition reaction
*syn*-CH$_3$CHOO + H$_2$O, indicating that the latter reaction is the dominant pathway.
Also Taatjes et al. (2013) reached same conclusions that CH$_3$CHOO reaction with
water is the dominate tropospheric removal pathway. Moreover, the high rate
coefficients of the reactions of CH$_3$CHOO with water vapour (Taatjes et al., 2013;
Anglada et al., 2011; 2016) ($k$(*syn*-CH$_3$CHOO + H$_2$O) = 4.0 × 10$^{-15}$ cm$^3$ molecule$^{-1}$ s$^{-1}$,
$k$(*anti*-CH$_3$CHOO + H$_2$O) = 1.0 ± 0.4 × 10$^{-14}$ cm$^3$ molecule$^{-1}$ s$^{-1}$) suggest that water
can effectively scavenge CH$_3$CHOO to generate low volatile HO-C(CH$_3$)HOO-H and
thus promote SOAs formation. The energy diagram of addition reactions between
CH$_3$CHOO and HO-C(CH$_3$)HOO-H is given in Figure 3. The optimized geometries of
all stationary points are shown in Figures S5 and S6.
Figure 3(a) demonstrates that the sequential additions of *anti*-CH$_3$CHOO to Pa$_2$
are strongly exothermic and spontaneous, indicating that the occurrence of these
consecutive reactions in the atmosphere is thermochemically feasible. The addition
reactions of 2*anti*-CH$_3$CHOO + Pa$_2$ start with the barrierless formation of pre-reactive
complexes IM3a and IM3b held together by weak hydrogen bonds and vdW forces.



Subsequently, the -OH and -OOH fragments in $Pa_2$ immediately add to the central
carbon atom of *anti*-$CH_3CHOO$ to produce P3a and P3b. The barriers of these two
addition reactions are 6.2 and 8.8 kcal $mol^{-1}$ with the concomitant release 39.5 and
34.1 kcal $mol^{-1}$ of energies. This result confirms that the most favorable channel, both
thermochemically and dynamically, corresponds to the -OOH group addition pathway.
Notably, the above reaction barriers are lower than that of the $2CH_2OO$ + $Pa_1$ system
by ~ 3.0 kcal $mol^{-1}$, indicating that *anti*-$CH_3CHOO$ is significantly more reactive than
the parent carbonyl oxide. This finding is further corroborated by the results of
Anglada et al. (2016) and Chen et al., (2017) who found that $CH_2OO$ is significantly
less reactive than *anti*-$CH_3CHOO$ towards $H_2O$, $HCOOH$, and $CH_3COOH$.
The addition reaction *anti*-$CH_3CHOO$ + $Pa_2$ results in the formation of P3a,
which can subsequently react with *anti*-$CH_3CHOO$ via channels R4a and R4b. Both
of these pathways start with the formation of pre-reactive complexes IM4a and IM4b
in entrance channels, that is followed by the addition of -OH and -OOH groups of P3a
to the central carbon atom of *anti*-$CH_3CHOO$ to produce P4a and P4b. According to
the predicted barrier heights, TS4a and TS4b lie 7.3 and 16.7 kcal $mol^{-1}$ above
complexes IM4a and IM4b, respectively, which re-confirms that the most favorable
reaction channel is the -OOH group addition pathway.
At this point, it is worth noting that the addition reactions in the $2syn$-$CH_3CHOO$
+ $Pa_{2'}$ system proceed through a similar mechanism and are thus only briefly
discussed in the following section. As revealed by Figure 3(b), both R5a and R5b
pathways start with the formations of vdW complexes IM5a and IM5b, that are
spontaneously converted into products P5a and P5b. The barriers of these two
addition reactions are estimated as 12.5 and 12.0 kcal $mol^{-1}$, respectively, and are
therefore significantly higher than those calculated for comparable $2anti$-$CH_3CHOO$
+ $Pa_2$ system. This discrepancy is ascribed to the fact that the steric repulsion between
the methyl group and the terminal oxygen in *syn*-$CH_3CHOO$ results in decreased
hydrogen transfer ability and hinders the formation of pre-reactive complexes.
The above result is further supported by recent reports, which claim the
*syn*-conformer to be substantially less reactive than the *anti*-conformer toward key

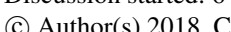



atmospheric species, such as $H_2O$, $SO_2$ and $NO_2$ (Taatjes et al., 2013; Anglada et al.,
2011; Sheps et al., 2014). The lowest-energy channel among the P5a/P5b +
*syn*-$CH_3CHOO$ reaction pathways, TS6a, involves the addition of the -OOH group
with a barrier of only 11.0 kcal mol$^{-1}$ and a large exothermicity of ~ 40 kcal mol$^{-1}$.

### 3.4 PES for the reaction of $(CH_3)_2COO$ with HO-$(CH_3)_2COO$-H

The dimethyl-substituted Criegee intermediate $(CH_3)_2COO$ is generated in the

ozonolysis of 2,3-dimethyl-2-butene (Lester et al., 2018; Drozd et al., 2017). The
unimolecular reaction of $(CH_3)_2COO$ and its bimolecular reaction with water vapour
strongly depend on temperature (Long et al., 2018). For example, the unimolecular
reaction is the dominant decay pathway above 240 K, whereas it reaction with $SO_2$
can compete well with the corresponding unimolecular reaction below 240 K (Long et
al., 2018). Although a fraction of $(CH_3)_2COO$ may proceed unimolecular
decomposition or react with $SO_2$ under some specific conditions, the removal of this
species from the atmosphere mainly occurs via its reaction with water vapour due to
its higher atmospheric concentration (Kuwata et al., 2015; Long et al., 2018; Huang et
al., 2015), which afford HO-$C(CH_3)_2OO$-H as the major product. The PES of addition
reactions $(CH_3)_2COO$ + HO-$C(CH_3)_2OO$-H is given in Figure 4, and the optimized
geometries of all stationary points are shown in Figure S7.

As shown in Figure 4, the vdW complexes IM7a and IM7b are 5.0 and 5.1

kcal mol$^{-1}$ lower in energy than the reactants, while the corresponding transition states
TS7a and TS7b leading to products P7a and P7b are 10.3 and 14.2 kcal mol$^{-1}$ higher
in energy than the respective complexes. The formation of P7a and P7b is strongly
exothermic, with the reaction energies of -31.0 and -25.8 kcal mol$^{-1}$. Again, this result
shows that the addition of the -OOH group in $Pa_3$ to the central carbon atom of
$(CH_3)_2COO$ is both thermochemically and dynamically favorable. Compared with the
barriers of 2*anti*-$CH_3CHOO$ + $Pa_2$ system given in Figure 3(a), one can notice that the
dimethyl-substituted parent carbonyl oxide leads to the barrier increasing by ~ 5
kcal mol$^{-1}$. The secondary addition reaction $(CH_3)_2COO$ + P7a is found to be similar
to that described for the $(CH_3)_2COO$ + $Pa_3$ system. The pre-reactive complexes IM8a



and IM8b are formed in entrance channels with over 5.0 kcal mol$^{-1}$ stabilization
energies, and followed by the addition of -OH and -OOH groups in P7a to the central
carbon atom of $(CH_3)_2COO$ to generate P8a and P8b. According to the predicted
barrier heights, TS8a and TS8b lie 12.0 and 13.0 kcal mol$^{-1}$ above complexes IM8a
and IM8b, respectively, which shows that the second addition reactions R8a and R8b
are nearly equally accessible.
**3.5 PES of distinct SCI reactions with HO-CH$_2$OO-H**
To gain deeper insights into the substituent-influenced modification atmospheric
oligomer composition, one should elucidate the origin of the substituent influence on
the reactivity and kinetics of carbonyl oxides. Therefore, an understanding of
structure-reactivity relationships is important for determining bimolecular processes
and reaction products. Since the addition of the -OOH group to the central carbon
atom of SCIs is shown to be both thermochemically and dynamically preferable, this
type of addition reaction is selected to study the effect of substituents on the reactivity
of carbonyl oxides. The PES of addition reactions SCIs + HO-CH$_2$OO-H is given in
Figure 5, whereas those for bimolecular reactions with other HHPs are displayed in
Figures S8-S10.
As shown in Figure 5, each reaction begins with the formation of a strong
pre-reactive complex and then surmounts a medium barrier that is higher in energy
relative to the reactants before forming the corresponding products. The bimolecular
reaction of CH$_2$OO with HO-CH$_2$OO-H to form P1a(HO-(CH$_2$OO)$_2$-H) is
characterized by a barrier of 8.8 kcal mol$^{-1}$ and an exothermicity of 43.4 kcal mol$^{-1}$.
Hence, the small barrier and large stability of the hydroperoxide species imply that its
formation is both thermochemically and kinetically favoured. Notably, the
introduction of a methyl group at the *anti*-position reduces the barrier by ~ 1.0
kcal mol$^{-1}$ relative to that of the CH$_2$OO + HO-CH$_2$OO-H system, whereas the
corresponding *syn*- and dimethyl substitutions increase the above barrier by 4.2 and
3.3 kcal mol$^{-1}$, respectively. These results indicate that the *anti*-conformer is
substantially more reactive toward HO-CH$_2$OO-H than *syn*-, dimethyl- and parent



conformers in the atmosphere. A similar conclusion has been obtained by studying the
reactions of *syn-/anti*-CH$_3$CHOO with water and SO$_2$, i.e., the rate coefficient of the
*anti*-CH$_3$CHOO reaction was calculated to be one to two orders of magnitude higher
than that of the *syn*-CH$_3$CHOO system (Lin et al., 2016; Huang et al., 2015; Taatjes et
al., 2013; Anglada et al., 2016). Therefore, it is concluded that the position and
number of methyl groups significantly affect barrier heights and reaction rates. On the
other hand, the exothermicities of other reaction pathways are lower than that of the
parent system, which implies that methyl substitution is thermochemically
unfavorable. A similar trend is observed for the bimolecular reactions of SCIs with
other HHPs (Figures S8-S10). In order to avoid redundancy, we do not repeat them
here in detail.

## 434 3.6 Kinetics and implications in atmospheric chemistry

To better understand the effect of substituents on reaction kinetics, the rate
coefficients of distinct SCI reactions with HO-CH$_2$OO-H are computed using a
combination of canonical transition state theory and Eckart tunneling correction at
temperatures between 273 and 400 K, with the obtained results listed in Table 2.
Table 2 shows that the predicted rate coefficients for the reaction of CH$_2$OO with
HO-CH$_2$OO-H(R1a) decrease with increasing temperature, with a similar trend
observed for *syn*-CH$_3$CHOO(R9), *anti*-CH$_3$CHOO(R10), and (CH$_3$)$_2$COO +
HO-CH$_2$OO-H(R11) systems. The above behavior is ascribed to the fact that the
apparent activation barriers $E_{app}$ of these four addition reactions are significantly
negative, as previously observed for the reaction of CH$_3$O$_2$ with BrO (Shallcross et al.,
2015). These findings imply that a significant fraction of atmospheric carbonyl oxides
may survive under high temperature conditions and react with peroxy radical or
organic acid to generate SOAs.
The obtained data shows that the rate coefficient depends on the relative position
and number of methyl groups in the parent carbonyl oxide, e.g., the rate coefficient
increases by two orders of magnitude when methyl substitution occurs at the
*anti*-position, whereas a reduction by four orders of magnitude is observed for methyl





substitution at the *syn*-position. Thus, the relative position of the methyl group plays
an important role in determining SCI reactivity, in particular, *anti*-substitution
promotes the reaction with HHPs and accelerates the formation of oligomers in the
atmosphere. Anglada et al. arrived at the same conclusion by studying the reactions of
SCIs with water vapour, showing that the *anti*-conformer is significantly more
reactive than the parent carbonyl oxide and the *syn*-conformer (Anglada et al., 2011;
2016). On the other hand, the introduction of two methyl groups does not result in a
marked rate coefficient change compared to the parent system, since the addition
reaction R11 is mediated by the pre-reactive hydrogen-bonded complex.
As discussed above, the reaction of *anti*-CH$_3$CHOO with HO-CH$_2$OO-H is
preferred over the other three pathways. Therefore, it would be interesting to
investigate whether the reaction between *anti*-CH$_3$CHOO and HO-CH$_2$OO-H can
compete with the reaction between *anti*-CH$_3$CHOO and formic acid, which represents
a substantially dominant atmospheric degradation pathway (Welz et al., 2014).
Assuming that the concentration of HO-CH$_2$OO-H is approximately equal to that of
SCIs (~5.0 $\times$ 10$^4$ molecules cm$^{-3}$; within an order of magnitude uncertainty) in the
boreal forest and rural environments of Finland and Germany (Novelli et al., 2016,
2017), the lifetime of *anti*-CH$_3$CHOO can be calculated as 5.9 $\times$ 10$^3$ s. The rate
coefficient of the bimolecular reaction of *anti*-CH$_3$CHOO with HCOOH
approximately equals (5$\pm$3) $\times$ 10$^{-10}$ cm$^3$ molecule$^{-1}$ s$^{-1}$ at 298 K (Welz et al., 2014),
corresponding to an estimated *anti*-CH$_3$CHOO lifetime of ~ 0.03 s at an average
daytime concentration of [HCOOH] = 2.0 $\times$ 10$^{11}$ molecules cm$^{-3}$ (Zhang et al., 2014).
Therefore, the *anti*-CH$_3$CHOO + HO-CH$_2$OO-H reaction may not compete with the
*anti*-CH$_3$CHOO + HCOOH reaction during daytime. However, the concentration of
formic acid dramatically decreases in the nighttime, allowing the *anti*-CH$_3$CHOO +
HO-CH$_2$OO-H reaction to compete with the *anti*-CH$_3$CHOO + HCOOH reaction at
temperatures below 273 K when the concentration of HCOOH equals 9.0 $\times$ 10$^7$
molecules cm$^{-3}$.
**4. Conclusion**



The reactivity and kinetics of oligomerization reactions of Criegee intermediates
with HHPs are studied using quantum-chemical methodologies in conjunction with
statistical theory calculations. The main conclusions are summarized as follows:
(a)  The oligomerization reactions of SCIs with HHPs are strongly exothermic

and spontaneous, signifying that the consecutive reactions are feasible

thermochemically in the atmosphere.

(b)  The addition of -OOH group in HHPs to the central carbon atom of SCIs is

both thermochemically and dynamically preferable as compared with the -OH

group addition pathway.

(c)  The reaction barrier and kinetics strongly depend on both, the number of the

substituents in the Criegee intermediate and on its position (*syn-* or *anti-*).

(d)  The rate coefficients show a significant increase when adding a methyl group

on the *anti*-position, whereas it displays a dramatical decrease on the *syn*-position.

On the other hand, the addition of dimethyl group does not cause much variation

in the rate coefficients.

## Acknowledgments


This work was supported by the National Key Research and Development
Program of China (2016YFA0203000) and the National Natural Science Foundation
of China (Nos. 41573138, 21473108 and 41805107). It was also partially supported
by the Key Project of International Cooperation of the Chinese Academy of Sciences
(GJHZ1543), Research Grants Council of Hong Kong (PolyU 152083/14E), Open
Foundation of State Key Laboratory of Loess and Quaternary Geology
(SKLLQG1627), and Shaanxi Province Postdoctoral Science Foundation (No.
2017BSHEDZZ62). Yu Huang is also supported by the "Hundred Talent Program" of
the Chinese Academy of Sciences.

## Supporting Information


The barriers for the addition reactions of carbonyl oxides with HHPs; PESs for
addition reactions of $CH_2OO$ + $HO-CH_2OO-H(Pb_1)$, *syn-/anti-*$CH_3CHOO$ +
$HO-C(CH_3)HOO-H$, $(CH_3)_2COO$ + $HO-C(CH_3)_2OO-H$ $(Pb_3)$, SCIs +



HO-CH(CH$_3$)OO-H(Pa$_2$),    SCIs    +    HO-CH(CH$_3$)OO-H(Pa$_{2'}$)    and    SCIs    +
HO-C(CH$_3$)$_2$OO-H; optimized geometries of all the stationary points.


Competing interests. The authors declare that they have no conflict of interest.
Author contributions. LC designed the study. LC and YH wrote the paper. LC
performed theoretical calculation. YX, ZS, JC, and WW analyzed the data. All authors
reviewed and commented on the paper.




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

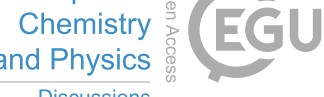



**Table 1** Relative energies (kcal mol$^{-1}$) for the stationary points and activation energies for the elementary pathways of carbonyl oxides reactions with water dimer. Labels A, B, C, and D are defined in Figure 1

| Entry | R1 | R2 | DO2H4O4H3 | DO4H2O3H1 | A | B | C | D | $\Delta E_a^{\#}$ |
|---|---|---|---|---|---|---|---|---|---|
| 1 | H | H | -123.6 | 96.7 | -15.2 | -10.8 | -49.8 | -42.2 | 4.4 |
| 2 | H | H | 124.5 | -94.9 | -15.0 | -11.0 | -49.1 | -42.3 | 4.0 |
| 3 | H | H | -143.8 | -116.9 | -14.7 | -10.6 | -48.7 | -42.3 | 4.1 |
| 4 | H | H | 143.0 | 122.7 | -14.7 | -9.5 | -49.3 | -42.4 | 5.2 |
| 5 | CH$_3$ | H | -126.2 | 100.9 | -16.0 | -6.6 | -43.9 | -36.0 | 9.4 |
| 6 | CH$_3$ | H | 130.0 | -90.4 | -15.5 | -6.5 | -43.0 | -36.3 | 9.0 |
| 7 | CH$_3$ | H | -146.0 | -116.3 | -15.8 | -6.9 | -42.5 | -36.3 | 8.9 |
| 8 | CH$_3$ | H | 138.1 | 126.3 | -15.7 | -5.0 | -43.3 | -36.0 | 10.2 |
| 9 | H | CH$_3$ | -122.1 | 95.0 | -18.0 | -11.2 | -48.5 | -40.9 | 6.8 |
| 10 | H | CH$_3$ | 125.6 | -93.6 | -17.4 | -11.1 | -47.6 | -40.9 | 6.3 |
| 11 | H | CH$_3$ | -138.1 | -120.5 | -18.0 | -10.8 | -46.9 | -40.9 | 7.2 |
| 12 | H | CH$_3$ | 139.2 | 123.4 | -17.5 | -9.7 | -47.9 | -40.9 | 7.8 |
| 13 | CH$_3$ | CH$_3$ | -125.4 | 101.5 | -18.4 | -7.5 | -43.0 | -35.1 | 10.9 |
| 14 | CH$_3$ | CH$_3$ | 128.5 | -89.8 | -18.0 | -7.1 | -41.9 | -35.3 | 10.9 |
| 15 | CH$_3$ | CH$_3$ | -145.4 | -117.6 | -18.5 | -7.3 | -41.3 | -35.3 | 11.2 |
| 16 | CH$_3$ | CH$_3$ | 136.3 | 129.4 | -18.1 | -5.5 | -42.5 | -35.1 | 12.6 |






**Table 2** Rate coefficients ($cm^3$ molecule$^{-1}$ s$^{-1}$) of SCIs reactions with HO-CH$_2$OO-H computed at
different temperatures

| T/K | $k_{CH2OO}$ | $k_{anti-CH3CHOO}$ | $k_{syn-CH3CHOO}$ | $k_{(CH3)2CHOO}$ |
|---|---|---|---|---|
| 273 | $2.5 \times 10^{-11}$ | $1.4 \times 10^{-8}$ | $1.7 \times 10^{-14}$ | $6.9 \times 10^{-12}$ |
| 280 | $1.9 \times 10^{-11}$ | $9.1 \times 10^{-9}$ | $1.5 \times 10^{-14}$ | $5.7 \times 10^{-12}$ |
| 298 | $1.1 \times 10^{-11}$ | $3.4 \times 10^{-9}$ | $1.2 \times 10^{-14}$ | $3.6 \times 10^{-12}$ |
| 300 | $9.9 \times 10^{-12}$ | $3.1 \times 10^{-9}$ | $1.2 \times 10^{-14}$ | $3.4 \times 10^{-12}$ |
| 320 | $5.6 \times 10^{-12}$ | $1.2 \times 10^{-9}$ | $9.9 \times 10^{-15}$ | $2.2 \times 10^{-12}$ |
| 340 | $3.4 \times 10^{-12}$ | $5.4 \times 10^{-10}$ | $8.4 \times 10^{-15}$ | $1.5 \times 10^{-12}$ |
| 360 | $2.2 \times 10^{-12}$ | $2.6 \times 10^{-10}$ | $7.3 \times 10^{-15}$ | $1.1 \times 10^{-12}$ |
| 380 | $1.5 \times 10^{-12}$ | $1.4 \times 10^{-10}$ | $6.4 \times 10^{-15}$ | $8.2 \times 10^{-13}$ |
| 400 | $1.0 \times 10^{-12}$ | $7.7 \times 10^{-11}$ | $5.8 \times 10^{-15}$ | $6.4 \times 10^{-13}$ |






# Figure Captions:


**Figure 1.** Schematic PES for the bimolecular reaction of SCIs with water dimer
**Figure 2.** PES ($\Delta G$ and $\Delta E$ (*italic*)) for the reaction of $CH_2OO$ with $HO\text{-}CH_2OO\text{-}H$ ($Pa_1$)
computed at the M06-2X/def2-TZVP//M06-2X/6-311+G(2df,2p) level of theory
**Figure 3.** PES ($\Delta G$ and $\Delta E$ (*italic*)) for the reactions of $HO\text{-}C(CH_3)HOO\text{-}H$ with *anti*-(a) and
*syn*-$CH_3CHOO$(b) calculated at the M06-2X/def2-TZVP//M06-2X/6-311+G(2df,2p) level of
theory
**Figure 4.** PES ($\Delta G$ and $\Delta E$ (*italic*)) for the reaction of $(CH_3)_2COO$ with $HO\text{-}C(CH_3)_2OO\text{-}H$($Pa_3$)
calculated at the M06-2X/def2-TZVP//M06-2X/6-311+G(2df,2p) level of theory
**Figure 5.** PES ($\Delta G$ and $\Delta E$ (*italic*)) of distinct SCI reactions with $HO\text{-}CH_2OO\text{-}H$ calculated at the
M06-2X/def2-TZVP//M06-2X/6-311+G(2df,2p) level of theory



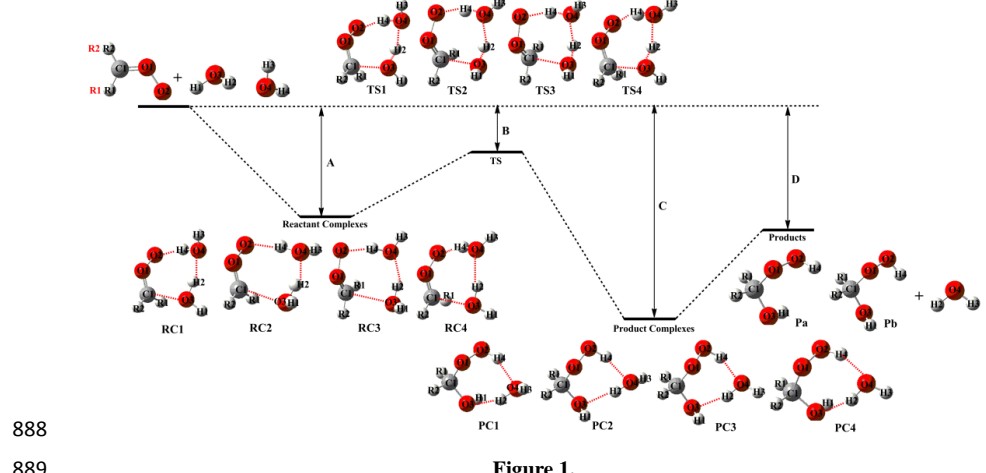


**Figure 1.**





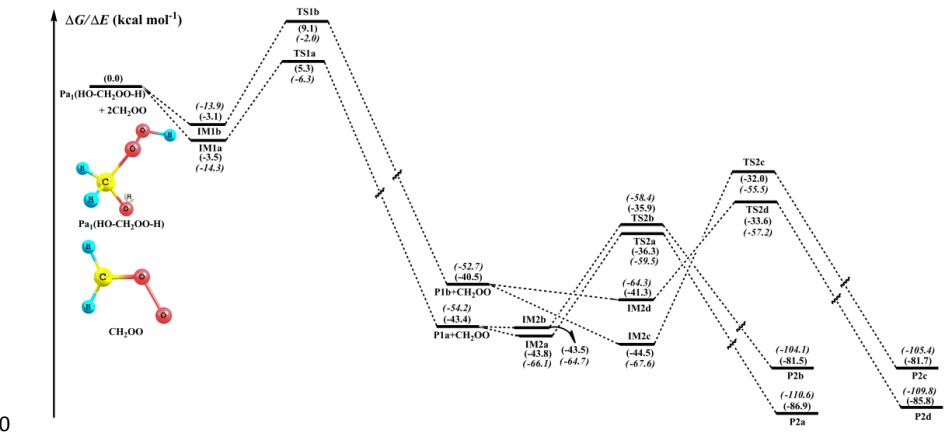

**Figure 2.**





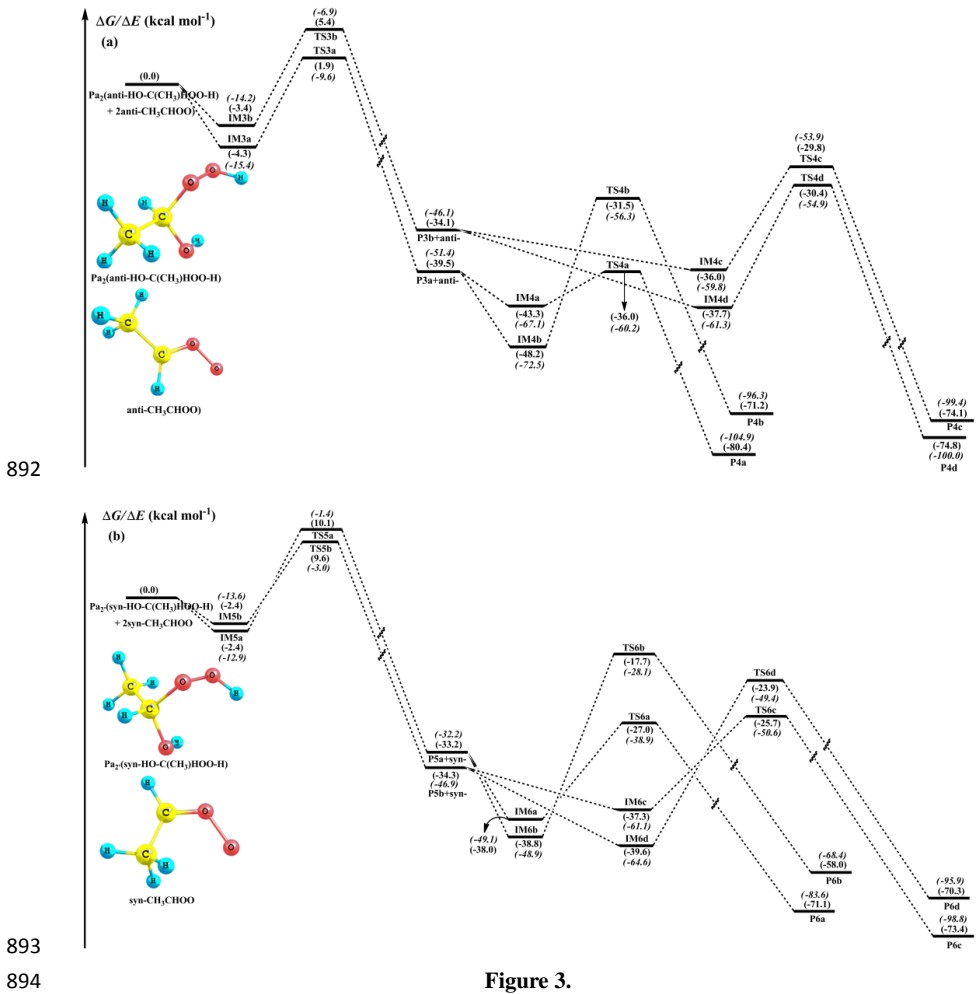

**Figure 3.**





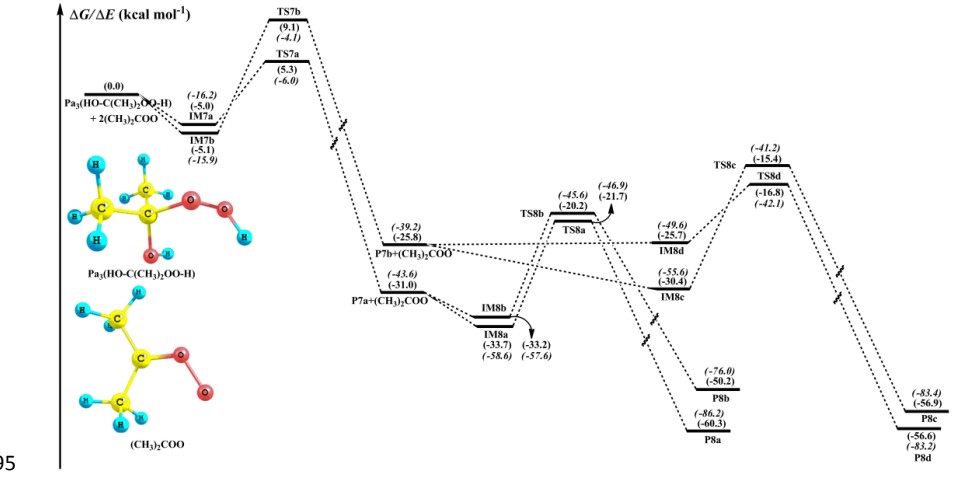

**Figure 4.**





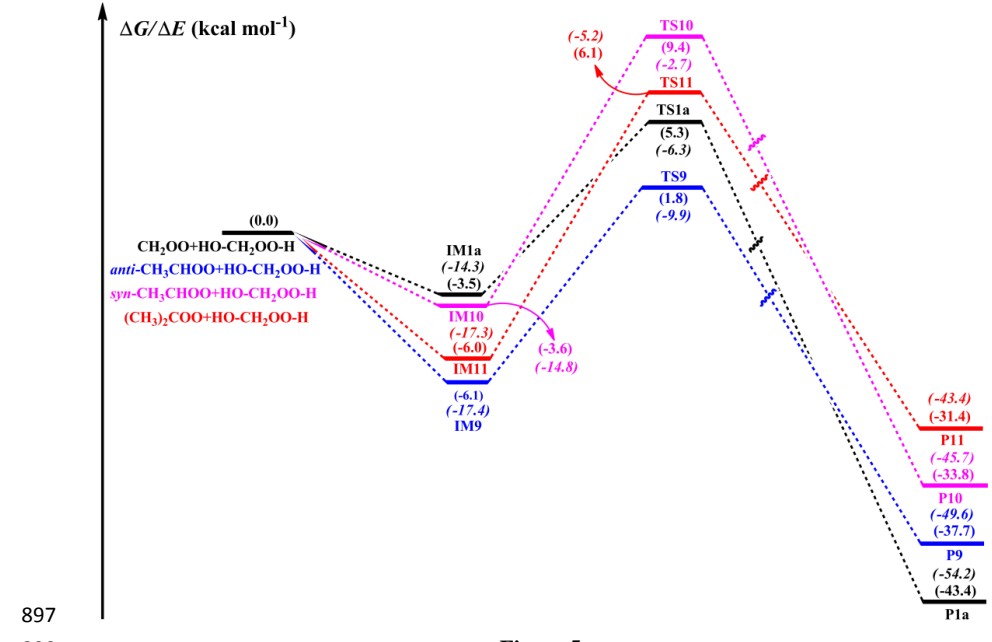

**Figure 5.**