# Peer review of "Oligomerization Reactions of Criegee Intermediates with Hydroxyalkyl Hydroperoxides: Mechanism, Kinetics, and Structure-Reactivity Relationship"

_Atmospheric Chemistry and Physics, 2018_

## Referee Comment (RC1) · Anonymous Referee #1 · 4 Jan 2019

This manuscript describes the gas phase reactions of Criegee intermediates (CI) formed from ozonolysis with hydroxyalkyl hydroperoxides (HHP) to determine the feasibility of these reactions in the formation of oligomeric species in atmospheric ozonolysis. The authors complete this task thoroughly with a number of different CI and HHP. The consecutive reactions for oligomer formation are clearly feasible thermodynamically and kinetically when considered in and of themselves. All the reactions studied proceed via exothermic formation of a pre-reactive complex and have reaction barriers of 4.5-13 kcal/mol. The authors have studied important reactions for the understanding

of new particle formation from ozonolysis, but the exact context in terms of overall CI reactivity is not discussed clearly and the context of their results needs to be clarified. As an overall loss process for CI, the reactions studied are minor, with the apparent lifetimes of nearly 2 hours, while unimolecular loss alone will be much faster (less than 1 second for the dimethyl CI). The investigated reactions may play a role in the processes of SOA and/or new particle formation, as these biomolecular reactions of CI with other organic compounds form material with very low volatility. This is a good contribution to the field. I recommend this manuscript for publication following minor revisions to the text and a substantial increase of the data presented in the supplemental material.

General Comments:

The sentence on lines 45-48 is plagiarized from Donahue et al.(1) I do not have the resources to fully vet the rest of this manuscript, but all text must be in your own words. You cannot copy such fine sentences as if they were your own. ALL instances of copied text must be removed and replaced with text you produce.

Please make clear throughout that the reactions studied are minor loss processes for CI. Your lifetime of 6000 seconds is quite long compared to other CI reactions. In addition, make explicitly clear that you calculate the lifetime with respect to only reaction with HHP. You often speak of just "lifetime" and do not specify reaction with HHP.

The products formed may be subject to other loss processes than thermal unimolecular decay, such as photolysis and reaction with more abundant reaction partners (e.g. OH). The authors need to place the chemistry studied here in a broader context and discuss (even if briefly) other loss processes.

As the major importance of this work relates to formation of secondary organic aerosol, the authors should make a better attempt to compare the possible influence of this specific chemistry on SOA formation and observed SOA formation from ozonolysis experiments, such as that in Ehn et al.(2) Further it should be clearly stated early on in the manuscript that the reactions studied are gas phase. A reaction with HHP might

suggest to many that this chemistry occurs within the particle phase where HHP could have a higher concentration.

All optimized geometries of the stationary points must be added to supplementary material. Pictures are not sufficient, z-matrices must be included. Also The imaginary frequencies which are used in your Eckart calculations should be included.

Be very careful about referring to SOAs as individual aerosols. As this work may relate to new particle formation this may technically be correct, but this usage is often problematic. Line 26 has the INCORRECT usage, it should be "secondary organic aerosol (SOA)", because the general term refers to mass added to existing aerosol due to condensation of material onto existing particles.

You state the importance of entropy in the free energy barriers, but you do not discuss its origin. Please comment on key vibrational modes that contribute to entropic factors.

Specific Comments:

Did the authors look for cyclic products? For example products where both dashed bonds in IM1a becoming full covalent bonds?

Do you rely solely on free energy estimate outputs? Are other methods such as SCTST too costly?

In sections 3.3 and 3.4, the discussion of loss processes for CI is flawed. Unimolecular decay is always a significant loss process. You seem to misinterpret the references you cite, both Drozd et al. and Long et al. find short thermal lifetimes for the CI formed in ozonolysis.(3,4) The authors must revise their statements to make clear that CI generally have significant unimolecular decay.

The energy labels in the supplement are confusing. The italics/non-italics energies are not always in same vertical order. Change the figure labels to make the non-italics energy the upper number in all cases. Do this for the main manuscript figures as well.

[Figure]

There are a number of awkward sentences/phrases, missing words, or grammatically confusing sentences. Carefully check over this document. I have listed a number of these below.

Line 27 change to "influence"

Line 43 Add Guenther et al as a reference for alkene emissions.5

Line 119 change to "is the dominant.."

132 change to "reactions occur during the..."

147 change to "there is little known of the reactivity..."

167 change to "represents"

168 change to "needs"

238 change to "possess OH and OOH"

246 change to "particularly"

267 – It is confusing to see a negative energy of stability, re-word this, and other instances, to be clearer.

315-317 You have an error the citations to Kroll et al. The year should be 2001.(6,7)

(1) Donahue, N. M.; Drozd, G. T.; Epstein, S. A.; Presto, A. A.; Kroll, J. H. Adventures in Ozoneland: Down the Rabbit-Hole. Phys. Chem. Chem. Phys. 2011, 13 (23), 10848–10857. (2) Ehn, M.; Thornton, J. A.; Kleist, E.; Sipila, M.; Junninen, H.; Pullinen, I.; Springer, M.; Rubach, F.; Tillmann, R.; Lee, B.; et al. A Large Source of Low-Volatility Secondary Organic Aerosol. Nature 2014, 506, 476–479. (3) Long, B.; Bao, J. L.; Truhlar, D. G. Unimolecular Reaction of Acetone Oxide and Its Reaction with Water in the Atmosphere. Proc. Natl. Acad. Sci. 2018, 115 (24), 6135–6140. (4) Drozd, G. T.; Kurtén, T.; Donahue, N. M.; Lester, M. I. Unimolecular Decay of the Dimethyl-Substituted Criegee Intermediate in Alkene Ozonolysis: Decay Time Scales

and the Importance of Tunneling. J. Phys. Chem. A 2017, 121 (32), 6036–6045. (5) Guenther, A.; Geron, C.; Pierce, T.; Lamb, B.; Harley, P.; Fall, R. Natural Emissions of Non-Methane Volatile Organic Compounds , Carbon Monoxide , and Oxides of Nitrogen from North America. 2000, 34. (6) Kroll, J. H.; Clarke, J. S.; Donahue, N. M.; Anderson, J. G.; Demerjian, K. L. Mechanism of HO Formation in the Gas-Phase Ozone $-$ Alkene Reaction . 1 . Direct , Pressure-Dependent Measurements of Prompt OH Yields Pressure-Dependent Measurements of Prompt OH Yields †. 2001. (7) Kroll, J. H.; Hanisco, T. F.; Donahue, N. M.; Demerjian, K. L.; Anderson, J. G. Accurate, Direct Measurements of Oh Yields from Gas-Phase Ozone-Alkene Reactions Using an in Situ LIF Instrument. Geophys. Res. Lett. 2001, 28 (20), 3863–3866.

---

## Referee Comment (RC2) · Anonymous Referee #2 · 4 Jan 2019

Comment on the manuscript "Oligomerization Reactions of Criegee Intermediates with Hydroxyalkyl Hydroperoxides: Mechanism, Kinetics, and Structure-Reactivity Relstionship.

by: Long Chen, Yu Huang, Yonggang Xue, Zhenxing Shen, Junji Cao, and Wenliang Wang

Chen and co-workers report an interesting work regarding the reaction of small carbonyl oxides ($H_2COO$, $HCH_3COO$ and $(CH_3)_2COO$) with small hydroxy methyl hy-

droperoxides (HHPs). The issue is interesting for understanding the first steps in the nucleation processes in the aerosol formation. However, in my opinion, there are several points that should be considered before publication.

First of all, I think that the title of this study goes too far. The main focus of this study refers just to the reaction of Carbonyl oxides with HHPs, although a second step, namely the mechanisms for the interaction of the products of these reaction with Carbonyl oxides is taken into account.

Along the text, the authors refer to several reaction products, as for instance, P2c, P2b and so on, but the structure of these compounds is not mentioned, which makes the work difficult to follow.

Some important references misses, as for instance CPL, 2001,337, 199, JPCA, 2001,105,446, JACS, 1997, 119, 330, CPC 2002, 2, 215, JPCA, 2003, 107, 5812, J. Atmos Chem, 2000, 35, 165 and references therein.

Regarding the theoretical approach, the authors state that all stationary points have been computing using the M06-2X functional, and for some selected elementary reactions they have performed single point energy calculations at CCSD(T) level of theory, pointing out that the deviations in the free energy barriers computed with both approaches range between 1.5 – 1.6 kcal/mol. The authors should clarify in which cases they have computed the energy barriers using both approaches, if they have taken into account basis set superposition corrections. They should compare the results of both approaches, for instance with results from the literature involving the reaction with water vapor (section 3.1) with results from the literature, where energy barriers are reported at CCSD(T)/CBS level of theory.

Regarding the kinetics, the authors should clarify if they have considered the pre-reactive complexes in the kinetic study and if they play a role in the temperature dependence of the rate constants.

The authors report rate constants for the reactions of the carbonyl oxides considered with HHP's (Table 2), but no mention is done for the reactions of P1x with Carbonyl oxides. Moreover, that authors should clarify if they have considered all different conformers of the stationary points in the kinetic study. In addition, they should estimate the errors in the these calculated rate constants, since they can be between one and two orders of magnitude according to the errors in the computed free energy barriers.

With respect to the atmospheric implications, the authors compare the reaction rates of the reaction investigated with those between carbonyl oxides with formic acid. In my opinion, the reactions rates of carbonyl oxides with water and water dimer, but also the reactions rates of HHPs with water should be also taken into account, because the high concentration of water vapor in the atmosphere. For the last, there are free energy barriers in the literature to compare with.

An hydrogen misses in the structure of P1a in Figure 2. In addition some addition structures of the P2x compounds should be drawn if the different figures and the numbers should have a larger size.

Please also note the supplement to this comment:
https://www.atmos-chem-phys-discuss.net/acp-2018-935/acp-2018-935-RC2-supplement.pdf

---

## Author Comment (AC1) · 31 Jan 2019

Prof. Yu Huang Key Lab of Aerosol Chemistry & Physics, Institute of Earth Environment, Chinese Academy of Sciences, Xi'an, 710061, China Tel./Fax: (86) 29-62336261 E-mail: huangyu@ieecas.cn Jan. 31, 2019 Dear reviewer, Revision for Manuscript acp-2018-935 We thank you very much for giving us the opportunity to revise our manuscript. We highly appreciate the reviewer for their comments and suggestions on the manuscript entitled "Mechanistic and Kinetics Investigations of Oligomer Formation from Criegee Intermediates Reactions with Hydroxyalkyl Hydroperoxides". We

have made revisions of our manuscript carefully according to the comments and suggestions of reviewer. The revised contents are marked in blue color. The response letter to reviewers is attached at the end of this cover letter. We hope that the revised manuscript can meet the requirement of Atmospheric Chemistry & Physics. Any further modifications or revisions, please do not hesitate to contact us. Look forward to hearing from you as soon as possible.

Best regards, Yu Huang   Comments of reviewer #1 1. The sentence on lines 45-48 is plagiarized from Donahue et al.(1) I do not have the resources to fully vet the rest of this manuscript, but all text must be in your own words. You cannot copy such fine sentences as if they were your own. ALL instances of copied text must be removed and replaced with text you produce. Response: We are sorry for this. The original sentence in the revised manuscript is "Alkene ozonolysis produces a carbonyl oxide (also called Criegee intermediates (CIs)) and a carbonyl moiety". Corresponding descriptions have been revised in the page 3 line 43-46 of the revised manuscript: Alkene ozonolysis produces a carbonyl oxide (also called Criegee intermediates (CIs)) and a carbonyl moiety (Donahue et al., 2011; Aplincourt et al., 2000; Johnson et al., 2008; Welz et al., 2012; Criegee et al., 1975; Taatjes et al., 2013). Moreover, all text in the revised manuscript has been checked carefully using our own words. 2. Please make clear throughout that the reactions studied are minor loss processes for CI. Your lifetime of 6000 seconds is quite long compared to other CI reactions. In addition, make explicitly clear that you calculate the lifetime with respect to only reaction with HHP. You often speak of just "lifetime" and do not specify reaction with HHP. Response: Previous experimental and theoretical investigations have shown that unimolecular decay of CI and its reaction with water are the dominant chemical sinks (Smith, et al., 2015; Taatjes, et al., 2013; Chao et al., 2015; Anglada, et al., 2016; Long et al., 2016, 2018, Lester et al., 2018). The main products of CIs reactions with water are hydroxyalkyl hydroperoxides (HHPs), which are important sources of hydroperoxides and carbonyl compounds (Anglada, et al., 2016; Kumar, et al., 2014). Although the carbonyl oxides reactions with HHPs are minor loss processes, the type of reactions is very important for understanding the first step of new particle formation form alkenes ozonolysis, particularly in heavily forested area. Our results demonstrate that the oligomer formations from CIs reactions with HHPs are both thermochemically and kinetically favoured. As shown in Table 2, the room temperature rate coefficient kR9 is $1.5 \times 10$-9 cm3 molecule-1 s-1. Assuming that the concentration of HO-CH2OO-H (Pa1) is approximately equal to that of SCIs ($\sim 5.0 \times 10$4 molecules•cm-3, within an order of magnitude uncertainty) in the boreal forest and rural environments of Finland and Germany (Novelli et al., 2016, 2017). The fate $\tau$ can be written as eqn (1) (Long et al., 2016): (1) where k and [X] represent the rate coefficient and the reactant concentration. The atmospheric lifetime of anti-CH3CHOO reactivity toward Pa1 is $1.3$-$13 \times 10$3 s. The experimental rate coefficient of anti-CH3CHOO reaction with water dimer approximately equals $\sim 1.0 \times 10$-11 cm3 molecule-1 s-1 at 298 K (Lin, et al., 2016). The concentration of water dimer is $5.5 \times 10$13 molecules•cm-3 at 3 km altitude (Long et al., 2016). The atmospheric lifetime of anti-CH3CHOO reactivity toward water dimer is $1.8 \times 10$-3 s. The result implies that the reactions studied are minor loss processes in the atmosphere. However, the [(H2O)2] decreases significantly with increasing altitudes. For example, at 15 km altitude, the [(H2O)2] is $2.7 \times 10$6 molecules cm-3 (Long et al., 2016), the atmospheric lifetime of anti-CH3CHOO is $3.4 \times 10$4 s. As discussed above, it can be found that the anti-CH3CHOO + Pa1 reaction can compete with the anti-CH3CHOO + (H2O)2 reaction in some regions where the altitude is above 15 km. Corresponding descriptions have been added in the page 20 line 549-564 of the revised manuscript: The room temperature rate coefficient kR9 is $1.5 \times 10$-9 cm3 molecule-1 s-1. Assuming that the concentration of Pa1 is approximately equal to that of SCIs ($\sim 5.0 \times 10$4 molecules•cm-3, within an order of magnitude uncertainty) in the boreal forest and rural environments of Finland and Germany (Novelli et al., 2016; 2017). The atmospheric lifetime of anti-CH3CHOO reactivity toward Pa1 can be estimated as $1.3$-$13 \times 10$3 s. The experimental rate coefficient of reaction R12 approximately equals $\sim 1.0 \times 10$-11 cm3 molecule-1 s-1 at 298 K (Lin et al., 2016). The concentration of water dimer is $5.5 \times 10$13 molecules•cm-3 at 3 km altitude (Long et al., 2016). The atmospheric

lifetime of anti-CH3CHOO reactivity toward water dimer is 1.8 × 10-3 s. The result implies that the reactions studied are minor loss processes in the atmosphere. However, the [(H2O)2] decreases significantly with increasing altitudes(Long et al., 2016). For example, at 15 km altitude, the [(H2O)2] is 2.7 × 106 molecules cm-3, the atmospheric lifetime of anti-CH3CHOO is 3.4 × 104 s. As discussed above, it can be found that the anti-CH3CHOO + Pa1 reaction can compete with the anti-CH3CHOO + (H2O)2 system in some regions where the altitude is above 15 km. 3. The products formed may be subject to other loss processes than thermal unimolecular decay, such as photolysis and reaction with more abundant reaction partners (e.g. OH). The authors need to place the chemistry studied here in a broader context and discuss (even if briefly) other loss processes. Response: Based on the Reviewer's suggestion, the bimolecular reaction of HO-CH2OO-H(Pa1) with OH radical is investigated at the M06-2X/def2-TZVP//M06-2X/6-311+ G(2df,2p) level of theory, and the corresponding potential energy surface is shown in Figure S5. As seen from Figure S5, the hydrogen abstraction reactions between OH and Pa1 are strongly exothermic and spontaneous, indicating that the occurrence of these reactions in the atmosphere is thermochemically feasible. One can notice that the exothermic reaction RH$\gamma$ releasing energy is significantly higher than that of the RH$\alpha$ and RH$\beta$. The pre-reactive complexes (IM$\alpha$H-a, $\beta$H-a, and $\gamma$H-a) are formed in the entrance channel, and followed by proceed the direct hydrogen abstraction processes leading to the products PH$\alpha$, PH$\beta$, and PH$\gamma$ plus H2O. The free-energy ($\Delta$Ga#) barriers predict TSH$\alpha$, TSH$\beta$ and TSH$\gamma$ to lie 6.8, 4.8, and 4.4 kcal•mol-1 above the energies of the corresponding pre-reactive complexes IM$\alpha$H-a, $\beta$H-a, and $\gamma$H-a. The result again shows that the OH abstraction H$\gamma$ atom reaction is the most energetically favorable channel. As shown in Figure 2, the addition reactions of 2CH2OO + Pa1 begin with the formation of loosely bound pre-reactive complexes IM1a and IM1b, of 3.1 and 2.6 kcal•mol-1 stability. They are formed by a hydrogen bond between the terminal CH2OO oxygen atom and the hydrogen atom of the -OOH group in Pa1, and a van der Waals (vdW) bond between the central carbon atom of CH2OO and the oxygen atom of the -OH group in Pa1. The above complexes

are immediately converted into products P1a and P1b via transition states TS1a and TS1b with barriers of 8.4 and 11.7 kcal•mol-1, respectively, while the corresponding reaction exothermicities are estimated as 43.4 and 40.5 kcal•mol-1, respectively. The above result shows that the most favorable channel is the addition of the -OOH group of Pa1 to the parent carbonyl oxide. However, the barrier is higher than that of the corresponding hydrogen abstraction reaction RH$\gamma$ 4.0 kcal•mol-1, indicating that the photochemical oxidation of hydroxyalkyl hydroperoxides is important in the atmosphere. In the present study, we mainly focus on the gas phase reaction mechanism and kinetics of oligomer formation from carbonyl oxides reactions with hydroxyalkyl hydroperoxides. This is because the type of reaction studied is very important for understanding the first step of new particle formation form alkenes ozonolysis, particularly in heavily forested area. Corresponding descriptions have been added in the page 13 line 339-357 of the revised manuscript: Moreover, the bimolecular reaction of Pa1 with OH radical is investigated at the M06-2X/def2-TZVP//M06-2X/6-311+G(2df,2p) level of theory, and the corresponding PES is shown in Figure S5. As seen from Figure S5, the hydrogen abstraction reactions between OH and Pa1 are strongly exothermic and spontaneous, indicating that the occurrence of these reactions in the atmosphere is thermochemically feasible. One can notice that the exothermic reaction RH$\gamma$ releasing energy is significantly higher than that of the RH$\alpha$ and RH$\beta$. The pre-reactive complexes (IM$\alpha$H-a, $\beta$H-a, and $\gamma$H-a) are formed in the entrance channel, and followed by proceed the direct hydrogen abstraction processes leading to the products PH$\alpha$, PH$\beta$, and PH$\gamma$ plus H2O. The barriers predict TSH$\alpha$, TSH$\beta$ and TSH$\gamma$ to lie 6.8, 4.8, and 4.4 kcal•mol-1 above the energies of the corresponding pre-reactive complexes IM$\alpha$H-a, $\beta$H-a, and $\gamma$H-a. The result again shows that the reaction RH$\gamma$ is the most energetically favorable channel. One can notice that the barrier of RH$\gamma$ is lower than that of the corresponding addition reaction R1a 4.0 kcal•mol-1, indicating that the photochemical oxidation of HHPs is important in the atmosphere. In the present study, we mainly focus on the gas phase reaction mechanism and kinetics of oligomer formation from carbonyl oxides reactions with HHPs. This is because the type of reaction

[Figure]

studied is very important for understanding the first step of new particle formation form alkenes ozonolysis, particularly in heavily forested area.

Figure S5. PES ($\Delta$G and $\Delta$E (italic)) for the reaction of HO-CH2OO-H (Pa1) with OH computed at the M06-2X/def2-TZVP//M06-2X/6-311+G(2df,2p) level of theory 4. As the major importance of this work relates to formation of secondary organic aerosol, the authors should make a better attempt to compare the possible influence of this specific chemistry on SOA formation and observed SOA formation from ozonolysis experiments, such as that in Ehn et al.(2) Further it should be clearly stated early on in the manuscript that the reactions studied are gas phase. A reaction with HHP might suggest to many that this chemistry occurs within the particle phase where HHP could have a higher concentration. Response: In the present study, we mainly focus on the gas phase reaction mechanism and kinetics of oligomer formation from carbonyl oxides reactions with HHPs. Our results demonstrate that the consecutive reactions of CIs with HHPs are both thermochemically and kinetically favoured, and the oligomers containing CIs as chain units. Ehn et al. (2014) reported a large source of low-volatility SOA generated from the ozonolysis of $\alpha$-pinene and other endocyclic monoterpenes under atmospheric conditions, and proposed that the mechanism of extremely low-volatility organic compounds (ELVOCs) formation is driven by RO2 autoxidation. The RO2 autoxidation pathway mainly includes the intramolecular hydrogen shift and the sequential O2 addition steps (Rissanen, et al., 2014). Compared with the RO2 autoxidation pathways, oligomerization reactions involving CIs as the repeat units, preserve carbon oxidation state and increase the number of a carbon backbone moiety, and therefore lead to a large reduction in the volatility (Wang, et al., 2016). Moreover, oligomerization reaction proceed over a shorter period of time during the early stage of biogenic SOA formation and growth (Heaton et al., 2007). Therefore, it is essential to investigate the gas phase Criegee chemistry-based mechanism of SOA formation and growth. Corresponding descriptions have been added in the page 5 line 116-127 and page 6 line 128-132 of the revised manuscript: On the other hand, Ehn et al. (2014) reported a large source of low-volatility SOA generated from the ozonolysis of

$\alpha$-pinene and other endocyclic monoterpenes under atmospheric conditions, and proposed that the mechanism of extremely low-volatility organic compounds (ELVOCs) formation is driven by RO2 autoxidation. The RO2 autoxidation pathway mainly includes the intramolecular hydrogen shift and the sequential O2 addition steps (Rissanen et al., 2014). Also several groups obtained similar conclusions that the highly oxygenated molecules (HOM) are produced via RO2 autoxidation in the cyclohexene and terpenes ozonolysis systems (Kirkby et al., 2016; Berndt et al., 2018). Moreover, HOM are major contributors to aerosol particle formation and growth on a global scale (Tröstl et al., 2016; Stolzenburg et al., 2018). Compared with the RO2 autoxidation pathways, oligomerization reactions involving CIs as the repeat units, preserve carbon oxidation state and increase the number of a carbon backbone moiety, and therefore lead to a large reduction in the volatility (Wang et al., 2016). Moreover, oligomerization reaction proceeds over a shorter period of time during the early stage of biogenic SOA formation and growth (Heaton et al., 2007). Therefore, we think that it is essential to investigate the gas phase Criegee chemistry-based mechanism of SOA formation and growth. 5. All optimized geometries of the stationary points must be added to supplementary material. Pictures are not sufiňAcient, z-matrices must be included. Also The imaginary frequencies which are used in your Eckart calculations should be included. Response: Based on the Reviewer's suggestion, the optimized geometries, z-matrices and vibrational frequencies of all stationary points are added in the Figures S4, S6-S8, and the imaginary frequencies of all transition states in Eckart calculations are also added in the Figures S4, S6-S8 in the supplementary material. 6. Be very careful about referring to SOAs as individual aerosols. As this work may relate to new particle formation this may technically be correct, but this usage is often problematic. Line 26 has the INCORRECT usage, it should be "secondary organic aerosol (SOA)", because the general term refers to mass added to existing aerosol due to condensation of material onto existing particles. Response: Based on the Reviewer's suggestion, the secondary organic aerosols (SOA) has been corrected in the revised manuscript. 7. You state the importance of entropy in the free energy barriers, but you do not discuss

its origin. Please comment on key vibrational modes that contribute to entropic factors. Response: According to the principle of statistical thermodynamics, the entropy can be described as eqn (2) (2) where H is the enthalpy, G is the free energy, T is the temperature in Kelvin. The relationship between the entropy and the partition function can be written as eqn (3) (3) where k is the Boltzmann constant, q is the partition function, it mainly includes translation (qtrans), vibration (qvib), external rotation (qrot), electronic (qele) and torsional (qtor) partition functions (q = qtransqvibqrotqeleqtor) (Mendes, et al., 2014). Among these partition functions, the qvib plays an important role in determining the entropic factors. Thus, the qvib is used to evaluate the contribution of vibrational mode to the entropic factors. The qvib is expressed as follows: (4) Table S2 lists the partition function of every vibrational mode involved in the complex IM1a. As shown in Table S2, the partition functions of low frequency vibrational mode (< 200 cm-1) are significantly higher than the high frequency vibrational mode (> 200 cm-1), and their contribution is up to 70.5 percent. The result implies that the low frequency vibrational mode contributes to entropic factor is significant. Similar conclusion is also obtained from the IM2a case (Table S3). Corresponding descriptions have been added in the page 11 line 293-299 of the revised manuscript: The partition function of every vibrational mode involved in the complex IM1a is listed in Table S2. As shown in Table S2, the partition function of low frequency vibrational mode (< 200 cm-1) is significantly higher than the high frequency vibrational mode (> 200 cm-1), and their contribution is up to 70.5 percent. The result implies that the low frequency vibrational mode contributes to entropic factor is significant. Similar conclusion is also obtained from the IM2a case (Table S3). Table S2 The partition function of every vibrational mode involved in the complex IM1a N  qvib percent(%) 1 51.49 4.03 30.43 2 123.66 1.66 12.52 3 149.16 1.37 10.31 4 158.76 1.28 9.66 5 199.39 1.00 7.58 6 242.53 0.81 6.12 7 257.19 0.76 5.74 8 422.78 0.42 3.15 9 501.75 0.33 2.48 10 538.04 0.30 2.24 11 642.00 0.22 1.69 12 723.92 0.18 1.37 13 858.58 0.13 0.97 14 896.89 0.12 0.89 15 958.09 0.10 0.76 16 1070.70 0.08 0.58 17 1093.27 0.07 0.55 18 1112.93 0.07 0.52 19 1134.12 0.07 0.50 20 1270.31 0.05 0.36 21 1300.08 0.04 0.33 22 1397.45 0.03 0.26

23 1429.61 0.03 0.24 24 1456.77 0.03 0.23 25 1506.87 0.03 0.20 26 1588.46 0.02 0.17 27 1674.72 0.02 0.13 28 3088.99 0.00 0.00 29 3148.98 0.00 0.00 30 3162.08 0.00 0.00 31 3162.65 0.00 0.00 32 3305.08 0.00 0.00 33 3868.49 0.00 0.00 Table S3 The partition function of every vibrational mode involved in the complex IM2a N  qvib percent(%) 1 45.57 4.56 19.74 2 66.77 3.10 13.44 3 93.88 2.20 9.52 4 102.82 2.00 8.68 5 122.46 1.67 7.25 6 144.25 1.41 6.12 7 151.51 1.34 5.82 8 186.88 1.08 4.66 9 236.15 0.84 3.62 10 242.41 0.81 3.52 11 324.58 0.58 2.51 12 383.38 0.47 2.05 13 460.65 0.37 1.61 14 475.56 0.35 1.54 15 546.86 0.29 1.25 16 589.00 0.26 1.12 17 676.46 0.20 0.89 18 704.48 0.19 0.82 19 751.93 0.17 0.73 20 889.93 0.12 0.52 21 904.59 0.12 0.50 22 957.33 0.10 0.44 23 1053.66 0.08 0.35 24 1082.87 0.07 0.33 25 1103.67 0.07 0.31 26 1123.39 0.07 0.29 27 1129.87 0.07 0.29 28 1139.95 0.06 0.28 29 1188.07 0.06 0.25 30 1271.30 0.05 0.20 31 1306.97 0.04 0.19 32 1353.81 0.04 0.17 33 1383.99 0.04 0.16 34 1415.10 0.03 0.14 35 1438.67 0.03 0.14 36 1455.62 0.03 0.13 37 1457.70 0.03 0.13 38 1501.08 0.03 0.12 39 1569.37 0.02 0.10 40 1663.59 0.02 0.08 41 3104.57 0.00 0.00 42 3113.88 0.00 0.00 43 3140.47 0.00 0.00 44 3173.49 0.00 0.00 45 3187.95 0.00 0.00 46 3286.9 0.00 0.00 47 3436.06 0.00 0.00 48 3910.66 0.00 0.00 8. Did the authors look for cyclic products? For example products where both dashed bonds in IM1a becoming full covalent bonds? Response: All the pre-reactive complexes and products involved in the title reaction system are the cyclic structures. For example, the bimolecular reaction of CH2OO with HO-CH2OO-H (Pa1) begins with the barrierless formation of pre-reactive complexes IM1a and IM1b held together by weak hydrogen bonds and vdW forces. Then, the -OOH and -OH fragments in Pa1 immediately add to the CH2OO central carbon atom to produce P1a and P1b. The detailed reaction mechanism mainly includes that the -O3(-O1) fragment of Pa1 moves to approach the CH2OO central carbon atom, whereas the -H4(-H3) atom gets attached to the terminal oxygen atom leading to products P1a and P1b (see Table S4). In the revised manuscript, atoms in molecules (AIM) analysis at the bond critical point (BCP) for the forming bonds (B1(O5-H4), B2(O3-C4), B3(O5-H3) and B1(O1-C4)) is performed with M06-2X functional (Biegler et al., 2000). The electronic density (),

Laplacian ($\nabla^2$), and the three eigenvalues of the Hessian of BCP are listed in Table S4. As shown in Table S4, the $\nabla^2$ values of all forming bonds are negative, indicating that they are covalent bonds. The values of B1 and B3 are significantly higher than that of the B2 and B4, showing that the bond strength of the former case is higher than the latter case. Corresponding descriptions have been added in the page 12 line 320-326 of the revised manuscript: The electronic density (), Laplacian ($\nabla^2$), and the three eigenvalues of the Hessian of the complexes IM1a and IM1b are displayed in Table S4. As shown in Table S4, the $\nabla^2$ values of all forming bonds (B1(O5-H4), B2(O3-C4), B3(O5-H3) and B1(O1-C4)) are negative, indicating that they are covalent bonds. The values of B1 and B3 are significantly higher than that of the B2 and B4, showing that the bond strength of the former case is higher than the latter case.

Table S4 AIM properties at the bond critical points for the forming bonds B1-B4 Bond  (e/Å3) $\nabla^2$(e/Å3) Eigenvalue 1 Eigenvalue 2 Eigenvalue 3 B1(O5-H4) 0.370 -2.776 -1.9316 -1.8598 1.0148 B2(O3-C4) 0.271 -0.559 -0.5734 -0.5112 0.5258 B3(O5-H3) 0.364 -2.760 -1.945 -1.8743 1.0588 B4(O1-C4) 0.269 -0.565 -0.5734 -0.5054 0.5139 9.  Do you rely solely on free energy estimate outputs?  Are other methods such as SCTST too costly? Response: The rate coefficients of elementary reactions are calculated using a combination of canonical transition state theory (CTST) and an asymmetric Eckart tunneling correction based on the free energies obtained from the M06-2X method, in the temperature range from 273 to 400 K. The predicted free energies are equal to the thermal correction to Gibbs free energies at the M06-2X/6-311+G(2df,2p) level plus the electronic energies obtained at the M06-2X/def2-TZVP level.  In order to assess the reliability of our kinetics study, the rate coefficients of HO-CH2OO-H (Pa1) + CH2OO (R1a) and HO-CH2OO-H (Pa1) + anti-CH3CHOO (R9) reactions are recomputed employing the canonical variational transition state theory (CVTST) with Eckart tunneling correction.  The calculated result is listed in Table S9.  As shown in Table S9, the predicted rate coefficients kCTST(R1a) and kCVTST(R1a) decrease with increasing temperature, and they exhibit a negative temperature dependence. The difference between kCTST(R1a) and

kCVTST(R1a) decreases in the range of 2.3 (273 K) to 1.8 (400 K). Such discrepancy between CTST and CVTST ones is acceptable. Similar conclusion is also obtained from the rate coefficients between kCTST(R9) and kCVTST(R9). It is concluded that the CTST/Eckart method allows one to reliably describe the kinetics parameters. Corresponding descriptions have been added in the page 18 line 506 and page 19 line 507-514 of the revised manuscript: In order to assess the reliability of our kinetics study, the rate coefficients of some selected reactions R1a and R9 are recomputed employing the canonical variational transition state theory (CVTST) with Eckart tunneling correction. The calculated result is listed in Table S9. As shown in Table S9, the difference between kCTST(R1a) and kCVTST(R1a) decreases in the range of 2.3 (273 K) to 1.8 (400 K). Such rate coefficients discrepancy between CTST and CVTST ones is acceptable. Similar conclusion is also obtained from the rate coefficients between kCTST(R9) and kCVTST(R9). It is concluded that the CTST/Eckart method allows one to reliably describe the kinetics parameters. Table S9 Rate coefficients (cm3 molecule-1 s-1) of elementary reactions R1a and R9 computed T kCTST(R1a) kCVTST(R1a) kCTST(R9) kCVTST(R9) 273 $1.2 \times 10^{-11}$ $3.5 \times 10^{-11}$ $5.5 \times 10^{-9}$ $1.3 \times 10^{-8}$ 280 $9.4 \times 10^{-12}$ $2.6 \times 10^{-11}$ $3.7 \times 10^{-9}$ $8.5 \times 10^{-9}$ 298 $5.4 \times 10^{-12}$ $1.4 \times 10^{-11}$ $1.5 \times 10^{-9}$ $3.2 \times 10^{-9}$ 300 $5.1 \times 10^{-12}$ $1.3 \times 10^{-11}$ $1.3 \times 10^{-9}$ $2.9 \times 10^{-9}$ 320 $3.0 \times 10^{-12}$ $7.2 \times 10^{-12}$ $5.5 \times 10^{-10}$ $1.1 \times 10^{-9}$ 340 $1.9 \times 10^{-12}$ $4.2 \times 10^{-12}$ $2.6 \times 10^{-10}$ $5.0 \times 10^{-10}$ 360 $1.2 \times 10^{-12}$ $2.7 \times 10^{-12}$ $1.4 \times 10^{-10}$ $2.4 \times 10^{-10}$ 380 $8.6 \times 10^{-13}$ $1.8 \times 10^{-12}$ $7.1 \times 10^{-11}$ $1.3 \times 10^{-10}$ 400 $6.3 \times 10^{-13}$ $1.2 \times 10^{-12}$ $4.1 \times 10^{-11}$ $7.3 \times 10^{-11}$ 10. In sections 3.3 and 3.4, the discussion of loss processes for CI is flawed. Unimolecular decay is always a significant loss process. You seem to misinterpret the references you cite, both Drozd et al. and Long et al. find short thermal lifetimes for the CI formed in ozonolysis.(3,4) The authors must revise their statements to make clear that CI generally have significant unimolecular decay. Response: Long et al. (2016) proposed that the predominant pathway of unimolecular decay of syn-CH3CHOO is isomerization to vinyl hydroperoxide (VHP) via the hydrogen atom migration from the methyl group to the terminal oxygen atom, then the

decomposition of VHP produces OH radical. Also Donahue et al. (2011) obtained similar conclusion that syn-CI isomerization to VHP is preferable due to the low ring strain of H-atom transfer transition state. Both the prompt and thermal unimolecular decay of the energized VHP may dissociate to OH radical, and their yields are strongly pressure and temperature dependents (Kroll, et al., 2001a,b). The preferable route of unimolecular decay of anti-CH3CHOO is ring-closure to dioxirane via an oxygen atom transfer (Donahue et al, 2011; Taatjes et al., 2013; Long et al., 2016). The dioxirane can finally isomerize to acetic acid via the "hot acid" channel (Kroll, et al., 2001a). Alternatively, the syn- and anti-CH3CHOO may undergo bimolecular reactions with water vapor lead to the formation of HO-C(CH3)HOO-H (Anglada et al., 2011). Long et al. (2018) proposed that the unimolecular decay of (CH3)2COO is the predominant pathway above 240 K, whereas it can compete with the reaction (CH3)2COO + SO2 below 240 K. Drozd et al. (2017) elucidated that tunneling for both the thermal and prompt unimolecular decay of (CH3)2COO is significant. Also Lester et al. (2018) obtained similar conclusion in the unimolecular decay of (CH3)2COO to OH radical reaction that the contribution of tunneling to the unimolecular decay rates is significant. Alternatively, the (CH3)2COO may react with water vapour leading to the formation of HO-C(CH3)2OO-H (Anglada et al., 2016). Corresponding descriptions have been added in the page 14 line 380-389, page 15 line 390-393 and page 16 line 437-445 of the revised manuscript: Long et al. (2016) proposed that the predominant pathway of unimolecular decay of syn-CH3CHOO is isomerization to vinyl hydroperoxide (VHP) via the hydrogen atom migration from the methyl group to the terminal oxygen atom, then the decomposition of VHP produces OH radical. Also Donahue et al. (2011) obtained similar conclusion that syn-CI isomerization to VHP is preferable due to the low ring strain of H-atom transfer transition state. Both of the prompt and thermal unimolecular decay of the energized VHP may dissociate to OH radical, and their yields are strongly pressure and temperature dependents (Kroll et al., 2001a,b). The preferable route of unimolecular decay of anti-CH3CHOO is ring-closure to dioxirane via an oxygen atom transfer (Donahue et al., 2011; Taatjes et al., 2013;

Long et al., 2016). The dioxirane can finally isomerize to acetic acid via the "hot acid" channel (Kroll et al., 2011b). Alternatively, the syn- and anti-CH3CHOO may undergo bimolecular reactions with water vapor lead to the formation of HO-C(CH3)HOO-H (Anglada et al., 2011). Long et al. (2018) proposed that the unimolecular decay of (CH3)2COO is the predominant pathway above 240 K, whereas it can compete with the reaction (CH3)2COO + SO2 below 240 K. Drozd et al. (2017) elucidated that tunneling for both the thermal and prompt unimolecular decay of (CH3)2COO is significant. Also Lester et al. (2018) obtained similar conclusion in the unimolecular decay of (CH3)2COO to OH radical reaction that the contribution of tunneling to the unimolecular decay rates is significant. Alternatively, the (CH3)2COO may react with water vapour leading to the formation of HO-C(CH3)2OO-H (Anglada et al., 2016). 11. The energy labels in the supplement are confusing. The italics/non-italics energies are not always in same vertical order. Change the figure labels to make the non-italics energy the upper number in all cases. Do this for the main manuscript figures as well. Response: Based on the Reviewer's suggestion, the non-italics energies have been placed in the upper number in the manuscript and supplement figures. 12. There are a number of awkward sentences/phrases, missing words, or grammatically confusing sentences. Carefully check over this document. I have listed a number of these below. Line 27 change to "influence" Line 43 Add Guenther et al as a reference for alkene emissions.5 Line 119 change to "is the dominant.." 132 change to "reactions occur during the..." 147 change to "there is little known of the reactivity..." 167 change to "represents" 168 change to "needs" 238 change to "possess OH and OOH" 246 change to "particularly" 267 - It is confusing to see a negative energy of stability, re-word this, and other in- stances, to be clearer. 315-317 You have an error the citations to Kroll et al. The year should be 2001.(6,7) Response: Based on the Reviewer's suggestion, the sentences/phrases, missing words, and the grammatically confusing sentences have been corrected carefully in the revised manuscript.   References Anglada, J. M., and Solé, A.: Impact of water dimer on the atmospheric reactivity of carbonyl oxides, Phys. Chem. Chem.

[revised manuscript text omitted]

Please also note the supplement to this comment:
https://www.atmos-chem-phys-discuss.net/acp-2018-935/acp-2018-935-AC1-supplement.pdf

[Figure]

**Fig. 1.**

**Supplement:**

Prof. Yu Huang
Key Lab of Aerosol Chemistry & Physics,
Institute of Earth Environment, Chinese Academy
of Sciences, Xi'an, 710061, China
Tel./Fax: (86) 29-62336261
E-mail: huangyu@ieecas.cn

Jan. 31, 2019

Dear reviewer,

**Revision for Manuscript acp-2018-935**

We thank you very much for giving us the opportunity to revise our manuscript. We highly appreciate the reviewer for their comments and suggestions on the manuscript entitled "**Mechanistic and Kinetics Investigations of Oligomer Formation from Criegee Intermediates Reactions with Hydroxyalkyl Hydroperoxides**". We have made revisions of our manuscript carefully according to the comments and suggestions of reviewer. The revised contents are marked in blue color. The response letter to reviewers is attached at the end of this cover letter.

We hope that the revised manuscript can meet the requirement of Atmospheric Chemistry & Physics. Any further modifications or revisions, please do not hesitate to contact us.

Look forward to hearing from you as soon as possible.

Best regards,

Yu Huang

**Comments of reviewer #1**

1. The sentence on lines 45-48 is plagiarized from Donahue et al.(1) I do not have the resources to fully vet the rest of this manuscript, but all text must be in your own words. You cannot copy such fine sentences as if they were your own. ALL instances of copied text must be removed and replaced with text you produce.

**Response:** We are sorry for this. The original sentence in the revised manuscript is "Alkene ozonolysis produces a carbonyl oxide (also called Criegee intermediates (CIs)) and a carbonyl moiety".

Corresponding descriptions have been revised in the page 3 line 43-46 of the revised manuscript:

*Alkene ozonolysis produces a carbonyl oxide (also called Criegee intermediates (CIs)) and a carbonyl moiety (Donahue et al., 2011; Aplincourt et al., 2000; Johnson et al., 2008; Welz et al., 2012; Criegee et al., 1975; Taatjes et al., 2013).*

Moreover, all text in the revised manuscript has been checked carefully using our own words.

2. Please make clear throughout that the reactions studied are minor loss processes for CI. Your lifetime of 6000 seconds is quite long compared to other CI reactions. In addition, make explicitly clear that you calculate the lifetime with respect to only reaction with HHP. You often speak of just "lifetime" and do not specify reaction with HHP.

**Response:** Previous experimental and theoretical investigations have shown that unimolecular decay of CI and its reaction with water are the dominant chemical sinks (Smith, et al., 2015; Taatjes, et al., 2013; Chao et al., 2015; Anglada, et al., 2016; Long et al., 2016, 2018, Lester et al., 2018). The main products of CIs reactions with water are hydroxyalkyl hydroperoxides (HHPs), which are important sources of hydroperoxides and carbonyl compounds (Anglada, et al., 2016; Kumar, et al., 2014). Although the carbonyl oxides reactions with HHPs are minor loss processes, the type of reactions is very important for understanding the first step of new particle formation form alkenes ozonolysis, particularly in heavily forested area. Our results demonstrate that the oligomer formations from CIs reactions with HHPs are both thermochemically and kinetically favoured.

As shown in Table 2, the room temperature rate coefficient $k_{R9}$ is $1.5 \times 10^{-9}$ cm$^3$ molecule$^{-1}$ s$^{-1}$. Assuming that the concentration of HO-CH$_2$OO-H (Pa$_1$) is approximately equal to that of SCIs (~

$5.0 \times 10^4$ molecules cm$^{-3}$, within an order of magnitude uncertainty) in the boreal forest and rural environments of Finland and Germany (Novelli et al., 2016, 2017). The fate $\tau$ can be written as eqn (1) (Long et al., 2016):

$$\tau \ = \ \frac{1}{k[\mathrm{X}]} \tag{1}$$

where $k$ and [X] represent the rate coefficient and the reactant concentration. The atmospheric lifetime of *anti*-CH$_3$CHOO reactivity toward Pa$_1$ is 1.3-13 $\times 10^3$ s. The experimental rate coefficient of *anti*-CH$_3$CHOO reaction with water dimer approximately equals ~ $1.0 \times 10^{-11}$ cm$^3$ molecule$^{-1}$ s$^{-1}$ at 298 K (Lin, et al., 2016). The concentration of water dimer is $5.5 \times 10^{13}$ molecules cm$^{-3}$ at 3 km altitude (Long et al., 2016). The atmospheric lifetime of *anti*-CH$_3$CHOO reactivity toward water dimer is $1.8 \times 10^{-3}$ s. The result implies that the reactions studied are minor loss processes in the atmosphere. However, the [(H$_2$O)$_2$] decreases significantly with increasing altitudes. For example, at 15 km altitude, the [(H$_2$O)$_2$] is $2.7 \times 10^6$ molecules cm$^{-3}$ (Long et al., 2016), the atmospheric lifetime of *anti*-CH$_3$CHOO is $3.4 \times 10^4$ s. As discussed above, it can be found that the *anti*-CH$_3$CHOO + Pa$_1$ reaction can compete with the *anti*-CH$_3$CHOO + (H$_2$O)$_2$ reaction in some regions where the altitude is above 15 km.

Corresponding descriptions have been added in the page 20 line 549-564 of the revised manuscript:

*The room temperature rate coefficient $k_{R9}$ is $1.5 \times 10^{-9}$ cm$^3$ molecule$^{-1}$ s$^{-1}$. Assuming that the concentration of Pa$_1$ is approximately equal to that of SCIs (~$5.0 \times 10^4$ molecules cm$^{-3}$, within an order of magnitude uncertainty) in the boreal forest and rural environments of Finland and Germany (Novelli et al., 2016; 2017). The atmospheric lifetime of anti-CH$_3$CHOO reactivity toward Pa$_1$ can be estimated as 1.3-13 $\times 10^3$ s. The experimental rate coefficient of reaction R12 approximately equals ~ $1.0 \times 10^{-11}$ cm$^3$ molecule$^{-1}$ s$^{-1}$ at 298 K (Lin et al., 2016). The concentration of water dimer is $5.5 \times 10^{13}$ molecules cm$^{-3}$ at 3 km altitude (Long et al., 2016). The atmospheric lifetime of anti-CH$_3$CHOO reactivity toward water dimer is $1.8 \times 10^{-3}$ s. The result implies that the reactions studied are minor loss processes in the atmosphere. However, the [(H$_2$O)$_2$] decreases significantly with increasing altitudes(Long et al., 2016). For example, at 15 km altitude, the [(H$_2$O)$_2$] is $2.7 \times 10^6$ molecules cm$^{-3}$, the atmospheric lifetime of anti-CH$_3$CHOO is $3.4 \times 10^4$ s. As discussed above, it can be found that the anti-CH$_3$CHOO + Pa$_1$ reaction can*

*compete with the anti-CH₃CHOO + (H₂O)₂ system in some regions where the altitude is above 15 km.*

3. The products formed may be subject to other loss processes than thermal unimolecular decay, such as photolysis and reaction with more abundant reaction partners (e.g. OH). The authors need to place the chemistry studied here in a broader context and discuss (even if briefly) other loss processes.

**Response:** Based on the Reviewer's suggestion, the bimolecular reaction of HO-CH₂OO-H(Pa₁) with OH radical is investigated at the M06-2X/def2-TZVP//M06-2X/6-311+ G(2df,2p) level of theory, and the corresponding potential energy surface is shown in Figure S5. As seen from Figure S5, the hydrogen abstraction reactions between OH and Pa₁ are strongly exothermic and spontaneous, indicating that the occurrence of these reactions in the atmosphere is thermochemically feasible. One can notice that the exothermic reaction RHγ releasing energy is significantly higher than that of the RHα and RHβ. The pre-reactive complexes (IMαH-a, βH-a, and γH-a) are formed in the entrance channel, and followed by proceed the direct hydrogen abstraction processes leading to the products PHα, PHβ, and PHγ plus H₂O. The free-energy ($\Delta G_a^{\#}$) barriers predict TSHα, TSHβ and TSHγ to lie 6.8, 4.8, and 4.4 kcal mol$^{-1}$ above the energies of the corresponding pre-reactive complexes IMαH-a, βH-a, and γH-a. The result again shows that the OH abstraction Hγ atom reaction is the most energetically favorable channel.

As shown in Figure 2, the addition reactions of 2CH₂OO + Pa₁ begin with the formation of loosely bound pre-reactive complexes IM1a and IM1b, of 3.1 and 2.6 kcal mol$^{-1}$ stability. They are formed by a hydrogen bond between the terminal CH₂OO oxygen atom and the hydrogen atom of the -OOH group in Pa₁, and a van der Waals (vdW) bond between the central carbon atom of CH₂OO and the oxygen atom of the -OH group in Pa₁. The above complexes are immediately converted into products P1a and P1b via transition states TS1a and TS1b with barriers of 8.4 and 11.7 kcal mol$^{-1}$, respectively, while the corresponding reaction exothermicities are estimated as 43.4 and 40.5 kcal mol$^{-1}$, respectively. The above result shows that the most favorable channel is the addition of the -OOH group of Pa₁ to the parent carbonyl oxide. However, the barrier is higher than that of the corresponding hydrogen abstraction reaction RHγ 4.0 kcal mol$^{-1}$, indicating that the photochemical oxidation of hydroxyalkyl hydroperoxides is important in the atmosphere.

In the present study, we mainly focus on the gas phase reaction mechanism and kinetics of

oligomer formation from carbonyl oxides reactions with hydroxyalkyl hydroperoxides. This is because the type of reaction studied is very important for understanding the first step of new particle formation form alkenes ozonolysis, particularly in heavily forested area.

Corresponding descriptions have been added in the page 13 line 339-357 of the revised manuscript:

*Moreover, the bimolecular reaction of $Pa_1$ with OH radical is investigated at the M06-2X/def2-TZVP//M06-2X/6-311+G(2df,2p) level of theory, and the corresponding PES is shown in Figure S5. As seen from Figure S5, the hydrogen abstraction reactions between OH and $Pa_1$ are strongly exothermic and spontaneous, indicating that the occurrence of these reactions in the atmosphere is thermochemically feasible. One can notice that the exothermic reaction RHγ releasing energy is significantly higher than that of the RHα and RHβ. The pre-reactive complexes (IMαH-a, βH-a, and γH-a) are formed in the entrance channel, and followed by proceed the direct hydrogen abstraction processes leading to the products PHα, PHβ, and PHγ plus $H_2O$. The barriers predict TSHα, TSHβ and TSHγ to lie 6.8, 4.8, and 4.4 kcal mol$^{-1}$ above the energies of the corresponding pre-reactive complexes IMαH-a, βH-a, and γH-a. The result again shows that the reaction RHγ is the most energetically favorable channel. One can notice that the barrier of RHγ is lower than that of the corresponding addition reaction R1a 4.0 kcal mol$^{-1}$, indicating that the photochemical oxidation of HHPs is important in the atmosphere. In the present study, we mainly focus on the gas phase reaction mechanism and kinetics of oligomer formation from carbonyl oxides reactions with HHPs. This is because the type of reaction studied is very important for understanding the first step of new particle formation form alkenes ozonolysis, particularly in heavily forested area.*

[Figure]

***Figure S5.*** *PES (ΔG and ΔE (italic)) for the reaction of HO-CH$_2$OO-H (Pa$_1$) with OH computed at the M06-2X/def2-TZVP//M06-2X/6-311+G(2df,2p) level of theory*

4. As the major importance of this work relates to formation of secondary organic aerosol, the authors should make a better attempt to compare the possible influence of this specific chemistry on SOA formation and observed SOA formation from ozonolysis experiments, such as that in Ehn et al.(2) Further it should be clearly stated early on in the manuscript that the reactions studied are gas phase. A reaction with HHP might suggest to many that this chemistry occurs within the particle phase where HHP could have a higher concentration.

**Response:** In the present study, we mainly focus on the gas phase reaction mechanism and kinetics of oligomer formation from carbonyl oxides reactions with HHPs. Our results demonstrate that the consecutive reactions of CIs with HHPs are both thermochemically and kinetically favoured, and the oligomers containing CIs as chain units. Ehn et al. (2014) reported a large source of low-volatility SOA generated from the ozonolysis of α-pinene and other endocyclic monoterpenes under atmospheric conditions, and proposed that the mechanism of extremely low-volatility organic compounds (ELVOCs) formation is driven by RO$_2$ autoxidation. The RO$_2$ autoxidation pathway mainly includes the intramolecular hydrogen shift and the sequential O$_2$ addition steps (Rissanen, et al., 2014). Compared with the RO$_2$ autoxidation pathways, oligomerization reactions involving CIs as the repeat units, preserve carbon oxidation state and increase the number of a carbon backbone moiety, and therefore lead to a large reduction in the volatility (Wang, et al., 2016). Moreover, oligomerization reaction proceed over a shorter

period of time during the early stage of biogenic SOA formation and growth (Heaton et al., 2007). Therefore, it is essential to investigate the gas phase Criegee chemistry-based mechanism of SOA formation and growth.

Corresponding descriptions have been added in the page 5 line 116-127 and page 6 line 128-132 of the revised manuscript:

*On the other hand, Ehn et al. (2014) reported a large source of low-volatility SOA generated from the ozonolysis of α-pinene and other endocyclic monoterpenes under atmospheric conditions, and proposed that the mechanism of extremely low-volatility organic compounds (ELVOCs) formation is driven by $RO_2$ autoxidation. The $RO_2$ autoxidation pathway mainly includes the intramolecular hydrogen shift and the sequential $O_2$ addition steps (Rissanen et al., 2014). Also several groups obtained similar conclusions that the highly oxygenated molecules (HOM) are produced via $RO_2$ autoxidation in the cyclohexene and terpenes ozonolysis systems (Kirkby et al., 2016; Berndt et al., 2018). Moreover, HOM are major contributors to aerosol particle formation and growth on a global scale (Tröstl et al., 2016; Stolzenburg et al., 2018). Compared with the $RO_2$ autoxidation pathways, oligomerization reactions involving CIs as the repeat units, preserve carbon oxidation state and increase the number of a carbon backbone moiety, and therefore lead to a large reduction in the volatility (Wang et al., 2016). Moreover, oligomerization reaction proceeds over a shorter period of time during the early stage of biogenic SOA formation and growth (Heaton et al., 2007). Therefore, we think that it is essential to investigate the gas phase Criegee chemistry-based mechanism of SOA formation and growth.*

5. All optimized geometries of the stationary points must be added to supplementary material. Pictures are not sufficient, z-matrices must be included. Also The imaginary frequencies which are used in your Eckart calculations should be included.

**Response:** Based on the Reviewer's suggestion, the optimized geometries, z-matrices and vibrational frequencies of all stationary points are added in the Figures S4, S6-S8, and the imaginary frequencies of all transition states in Eckart calculations are also added in the Figures S4, S6-S8 in the supplementary material.

6. Be very careful about referring to SOAs as individual aerosols. As this work may relate to new particle formation this may technically be correct, but this usage is often problematic. Line 26 has the INCORRECT usage, it should be "secondary organic aerosol (SOA)", because the general

term refers to mass added to existing aerosol due to condensation of material onto existing particles.

**Response:** Based on the Reviewer's suggestion, the secondary organic aerosols (SOA) has been corrected in the revised manuscript.

7. You state the importance of entropy in the free energy barriers, but you do not discuss its origin. Please comment on key vibrational modes that contribute to entropic factors.

**Response:** According to the principle of statistical thermodynamics, the entropy can be described as eqn (2)

$$S = \frac{H - G}{T} \tag{2}$$

where H is the enthalpy, G is the free energy, T is the temperature in Kelvin. The relationship between the entropy and the partition function can be written as eqn (3)

$$S = k \ln \frac{q^N}{N!} + NkT \left( \frac{\partial lnq}{\partial T} \right)_{V,N} \tag{3}$$

where $k$ is the Boltzmann constant, $q$ is the partition function, it mainly includes translation ($q_{trans}$), vibration ($q_{vib}$), external rotation ($q_{rot}$), electronic ($q_{ele}$) and torsional ($q_{tor}$) partition functions ($q = q_{trans}q_{vib}q_{rot}q_{ele}q_{tor}$) (Mendes, et al., 2014). Among these partition functions, the $q_{vib}$ plays an important role in determining the entropic factors. Thus, the $q_{vib}$ is used to evaluate the contribution of vibrational mode to the entropic factors. The $q_{vib}$ is expressed as follows:

$$q_{vib} = \exp(-\frac{1}{2}\frac{h\nu}{kT}) \times (\frac{1}{1 - e^{-h\nu/kT}}) \tag{4}$$

Table S2 lists the partition function of every vibrational mode involved in the complex IM1a. As shown in Table S2, the partition functions of low frequency vibrational mode ($< 200$ cm$^{-1}$) are significantly higher than the high frequency vibrational mode ($> 200$ cm$^{-1}$), and their contribution is up to 70.5 percent. The result implies that the low frequency vibrational mode contributes to entropic factor is significant. Similar conclusion is also obtained from the IM2a case (Table S3).

Corresponding descriptions have been added in the page 11 line 293-299 of the revised manuscript:

*The partition function of every vibrational mode involved in the complex IM1a is listed in Table S2. As shown in Table S2, the partition function of low frequency vibrational mode ($< 200$*

$cm^{-1}$) is significantly higher than the high frequency vibrational mode (> 200 $cm^{-1}$), and their contribution is up to 70.5 percent. The result implies that the low frequency vibrational mode contributes to entropic factor is significant. Similar conclusion is also obtained from the IM2a case (Table S3).

**Table S2** The partition function of every vibrational mode involved in the complex IM1a

| N | ν | $q_{vib}$ | percent(%) |
|---|---|---|---|
| 1 | 51.49 | 4.03 | 30.43 |
| 2 | 123.66 | 1.66 | 12.52 |
| 3 | 149.16 | 1.37 | 10.31 |
| 4 | 158.76 | 1.28 | 9.66 |
| 5 | 199.39 | 1.00 | 7.58 |
| 6 | 242.53 | 0.81 | 6.12 |
| 7 | 257.19 | 0.76 | 5.74 |
| 8 | 422.78 | 0.42 | 3.15 |
| 9 | 501.75 | 0.33 | 2.48 |
| 10 | 538.04 | 0.30 | 2.24 |
| 11 | 642.00 | 0.22 | 1.69 |
| 12 | 723.92 | 0.18 | 1.37 |
| 13 | 858.58 | 0.13 | 0.97 |
| 14 | 896.89 | 0.12 | 0.89 |
| 15 | 958.09 | 0.10 | 0.76 |
| 16 | 1070.70 | 0.08 | 0.58 |
| 17 | 1093.27 | 0.07 | 0.55 |
| 18 | 1112.93 | 0.07 | 0.52 |
| 19 | 1134.12 | 0.07 | 0.50 |
| 20 | 1270.31 | 0.05 | 0.36 |
| 21 | 1300.08 | 0.04 | 0.33 |
| 22 | 1397.45 | 0.03 | 0.26 |
| 23 | 1429.61 | 0.03 | 0.24 |
| 24 | 1456.77 | 0.03 | 0.23 |
| 25 | 1506.87 | 0.03 | 0.20 |
| 26 | 1588.46 | 0.02 | 0.17 |
| 27 | 1674.72 | 0.02 | 0.13 |
| 28 | 3088.99 | 0.00 | 0.00 |
| 29 | 3148.98 | 0.00 | 0.00 |
| 30 | 3162.08 | 0.00 | 0.00 |
| 31 | 3162.65 | 0.00 | 0.00 |
| 32 | 3305.08 | 0.00 | 0.00 |
| 33 | 3868.49 | 0.00 | 0.00 |

**Table S3** The partition function of every vibrational mode involved in the complex IM2a

| N | ν | $q_{vib}$ | percent(%) |
|---|---|---|---|
| 1 | 45.57 | 4.56 | 19.74 |
| 2 | 66.77 | 3.10 | 13.44 |
| 3 | 93.88 | 2.20 | 9.52 |
| 4 | 102.82 | 2.00 | 8.68 |
| 5 | 122.46 | 1.67 | 7.25 |
| 6 | 144.25 | 1.41 | 6.12 |
| 7 | 151.51 | 1.34 | 5.82 |
| 8 | 186.88 | 1.08 | 4.66 |

| | | | |
|---|---|---|---|
| 9 | 236.15 | 0.84 | 3.62 |
| 10 | 242.41 | 0.81 | 3.52 |
| 11 | 324.58 | 0.58 | 2.51 |
| 12 | 383.38 | 0.47 | 2.05 |
| 13 | 460.65 | 0.37 | 1.61 |
| 14 | 475.56 | 0.35 | 1.54 |
| 15 | 546.86 | 0.29 | 1.25 |
| 16 | 589.00 | 0.26 | 1.12 |
| 17 | 676.46 | 0.20 | 0.89 |
| 18 | 704.48 | 0.19 | 0.82 |
| 19 | 751.93 | 0.17 | 0.73 |
| 20 | 889.93 | 0.12 | 0.52 |
| 21 | 904.59 | 0.12 | 0.50 |
| 22 | 957.33 | 0.10 | 0.44 |
| 23 | 1053.66 | 0.08 | 0.35 |
| 24 | 1082.87 | 0.07 | 0.33 |
| 25 | 1103.67 | 0.07 | 0.31 |
| 26 | 1123.39 | 0.07 | 0.29 |
| 27 | 1129.87 | 0.07 | 0.29 |
| 28 | 1139.95 | 0.06 | 0.28 |
| 29 | 1188.07 | 0.06 | 0.25 |
| 30 | 1271.30 | 0.05 | 0.20 |
| 31 | 1306.97 | 0.04 | 0.19 |
| 32 | 1353.81 | 0.04 | 0.17 |
| 33 | 1383.99 | 0.04 | 0.16 |
| 34 | 1415.10 | 0.03 | 0.14 |
| 35 | 1438.67 | 0.03 | 0.14 |
| 36 | 1455.62 | 0.03 | 0.13 |
| 37 | 1457.70 | 0.03 | 0.13 |
| 38 | 1501.08 | 0.03 | 0.12 |
| 39 | 1569.37 | 0.02 | 0.10 |
| 40 | 1663.59 | 0.02 | 0.08 |
| 41 | 3104.57 | 0.00 | 0.00 |
| 42 | 3113.88 | 0.00 | 0.00 |
| 43 | 3140.47 | 0.00 | 0.00 |
| 44 | 3173.49 | 0.00 | 0.00 |
| 45 | 3187.95 | 0.00 | 0.00 |
| 46 | 3286.9 | 0.00 | 0.00 |
| 47 | 3436.06 | 0.00 | 0.00 |
| 48 | 3910.66 | 0.00 | 0.00 |

8. Did the authors look for cyclic products? For example products where both dashed bonds in IM1a becoming full covalent bonds?

**Response:** All the pre-reactive complexes and products involved in the title reaction system

are the cyclic structures. For example, the bimolecular reaction of $CH_2OO$ with $HO-CH_2OO-H$ ($Pa_1$) begins with the barrierless formation of pre-reactive complexes IM1a and IM1b held together by weak hydrogen bonds and vdW forces. Then, the -OOH and -OH fragments in $Pa_1$ immediately add to the $CH_2OO$ central carbon atom to produce P1a and P1b. The detailed reaction mechanism mainly includes that the -O3(-O1) fragment of $Pa_1$ moves to approach the $CH_2OO$ central carbon atom, whereas the -H4(-H3) atom gets attached to the terminal oxygen atom leading to products P1a and P1b (see Table S4). In the revised manuscript, atoms in molecules (AIM) analysis at the bond critical point (BCP) for the forming bonds (B1(O5-H4), B2(O3-C4), B3(O5-H3) and B1(O1-C4)) is performed with M06-2X functional (Biegler et al., 2000). The electronic density ($\rho$), Laplacian ($\nabla^2$), and the three eigenvalues of the Hessian of BCP are listed in Table S4. As shown in Table S4, the $\nabla^2$ values of all forming bonds are negative, indicating that they are covalent bonds. The $\rho$ values of B1 and B3 are significantly higher than that of the B2 and B4, showing that the bond strength of the former case is higher than the latter case.

Corresponding descriptions have been added in the page 12 line 320-326 of the revised manuscript:

*The electronic density ($\rho$), Laplacian ($\nabla^2$), and the three eigenvalues of the Hessian of the complexes IM1a and IM1b are displayed in Table S4. As shown in Table S4, the $\nabla^2$ values of all forming bonds (B1(O5-H4), B2(O3-C4), B3(O5-H3) and B1(O1-C4)) are negative, indicating that they are covalent bonds. The $\rho$ values of B1 and B3 are significantly higher than that of the B2 and B4, showing that the bond strength of the former case is higher than the latter case.*

[Figure]

IM1a            TS1a            P1a

IM1b            TS1b            P1b

**Table S4** AIM properties at the bond critical points for the forming bonds B1-B4

| Bond | $\rho$ (e/Å$^3$) | $\nabla^2$(e/Å$^3$) | Eigenvalue 1 | Eigenvalue 2 | Eigenvalue 3 |
|------|------|------|------|------|------|
| B1(O5-H4) | 0.370 | -2.776 | -1.9316 | -1.8598 | 1.0148 |
| B2(O3-C4) | 0.271 | -0.559 | -0.5734 | -0.5112 | 0.5258 |
| B3(O5-H3) | 0.364 | -2.760 | -1.945 | -1.8743 | 1.0588 |
| B4(O1-C4) | 0.269 | -0.565 | -0.5734 | -0.5054 | 0.5139 |

9. Do you rely solely on free energy estimate outputs? Are other methods such as SCTST too costly?

**Response:** The rate coefficients of elementary reactions are calculated using a combination of canonical transition state theory (CTST) and an asymmetric Eckart tunneling correction based on the free energies obtained from the M06-2X method, in the temperature range from 273 to 400 K. The predicted free energies are equal to the thermal correction to Gibbs free energies at the M06-2X/6-311+G(2df,2p) level plus the electronic energies obtained at the M06-2X/def2-TZVP level. In order to assess the reliability of our kinetics study, the rate coefficients of HO-CH$_2$OO-H (Pa$_1$) + CH$_2$OO (R1a) and HO-CH$_2$OO-H (Pa$_1$) + *anti*-CH$_3$CHOO (R9) reactions are recomputed employing the canonical variational transition state theory (CVTST) with Eckart tunneling correction. The calculated result is listed in Table S9. As shown in Table S9, the predicted rate

coefficients $k_{CTST(R1a)}$ and $k_{CVTST(R1a)}$ decrease with increasing temperature, and they exhibit a negative temperature dependence. The difference between $k_{CTST(R1a)}$ and $k_{CVTST(R1a)}$ decreases in the range of 2.3 (273 K) to 1.8 (400 K). Such discrepancy between CTST and CVTST ones is acceptable. Similar conclusion is also obtained from the rate coefficients between $k_{CTST(R9)}$ and $k_{CVTST(R9)}$. It is concluded that the CTST/Eckart method allows one to reliably describe the kinetics parameters.

Corresponding descriptions have been added in the page 18 line 506 and page 19 line 507-514 of the revised manuscript:

*In order to assess the reliability of our kinetics study, the rate coefficients of some selected reactions R1a and R9 are recomputed employing the canonical variational transition state theory (CVTST) with Eckart tunneling correction. The calculated result is listed in Table S9. As shown in Table S9, the difference between $k_{CTST(R1a)}$ and $k_{CVTST(R1a)}$ decreases in the range of 2.3 (273 K) to 1.8 (400 K). Such rate coefficients discrepancy between CTST and CVTST ones is acceptable. Similar conclusion is also obtained from the rate coefficients between $k_{CTST(R9)}$ and $k_{CVTST(R9)}$. It is concluded that the CTST/Eckart method allows one to reliably describe the kinetics parameters.*

**Table S9** *Rate coefficients ($cm^3$ molecule$^{-1}$ s$^{-1}$) of elementary reactions R1a and R9 computed*

| T | $k_{CTST(R1a)}$ | $k_{CVTST(R1a)}$ | $k_{CTST(R9)}$ | $k_{CVTST(R9)}$ |
|---|---|---|---|---|
| 273 | $1.2 \times 10^{-11}$ | $3.5 \times 10^{-11}$ | $5.5 \times 10^{-9}$ | $1.3 \times 10^{-8}$ |
| 280 | $9.4 \times 10^{-12}$ | $2.6 \times 10^{-11}$ | $3.7 \times 10^{-9}$ | $8.5 \times 10^{-9}$ |
| 298 | $5.4 \times 10^{-12}$ | $1.4 \times 10^{-11}$ | $1.5 \times 10^{-9}$ | $3.2 \times 10^{-9}$ |
| 300 | $5.1 \times 10^{-12}$ | $1.3 \times 10^{-11}$ | $1.3 \times 10^{-9}$ | $2.9 \times 10^{-9}$ |
| 320 | $3.0 \times 10^{-12}$ | $7.2 \times 10^{-12}$ | $5.5 \times 10^{-10}$ | $1.1 \times 10^{-9}$ |
| 340 | $1.9 \times 10^{-12}$ | $4.2 \times 10^{-12}$ | $2.6 \times 10^{-10}$ | $5.0 \times 10^{-10}$ |
| 360 | $1.2 \times 10^{-12}$ | $2.7 \times 10^{-12}$ | $1.4 \times 10^{-10}$ | $2.4 \times 10^{-10}$ |
| 380 | $8.6 \times 10^{-13}$ | $1.8 \times 10^{-12}$ | $7.1 \times 10^{-11}$ | $1.3 \times 10^{-10}$ |
| 400 | $6.3 \times 10^{-13}$ | $1.2 \times 10^{-12}$ | $4.1 \times 10^{-11}$ | $7.3 \times 10^{-11}$ |

10. In sections 3.3 and 3.4, the discussion of loss processes for CI is flawed. Unimolecular decay is always a significant loss process. You seem to misinterpret the references you cite, both Drozd et al. and Long et al. find short thermal lifetimes for the CI formed in ozonolysis.(3,4) The

authors must revise their statements to make clear that CI generally have significant unimolecular decay.

**Response:** Long et al. (2016) proposed that the predominant pathway of unimolecular decay of *syn*-CH$_3$CHOO is isomerization to vinyl hydroperoxide (VHP) via the hydrogen atom migration from the methyl group to the terminal oxygen atom, then the decomposition of VHP produces OH radical. Also Donahue et al. (2011) obtained similar conclusion that *syn*-CI isomerization to VHP is preferable due to the low ring strain of H-atom transfer transition state. Both the prompt and thermal unimolecular decay of the energized VHP may dissociate to OH radical, and their yields are strongly pressure and temperature dependents (Kroll, et al., 2001a,b). The preferable route of unimolecular decay of *anti*-CH$_3$CHOO is ring-closure to dioxirane via an oxygen atom transfer (Donahue et al, 2011; Taatjes et al., 2013; Long et al., 2016). The dioxirane can finally isomerize to acetic acid via the "hot acid" channel (Kroll, et al., 2001a). Alternatively, the *syn*- and *anti*-CH$_3$CHOO may undergo bimolecular reactions with water vapor lead to the formation of HO-C(CH$_3$)HOO-H (Anglada et al., 2011).

Long et al. (2018) proposed that the unimolecular decay of (CH$_3$)$_2$COO is the predominant pathway above 240 K, whereas it can compete with the reaction (CH$_3$)$_2$COO + SO$_2$ below 240 K. Drozd et al. (2017) elucidated that tunneling for both the thermal and prompt unimolecular decay of (CH$_3$)$_2$COO is significant. Also Lester et al. (2018) obtained similar conclusion in the unimolecular decay of (CH$_3$)$_2$COO to OH radical reaction that the contribution of tunneling to the unimolecular decay rates is significant. Alternatively, the (CH$_3$)$_2$COO may react with water vapour leading to the formation of HO-C(CH$_3$)$_2$OO-H (Anglada et al., 2016).

Corresponding descriptions have been added in the page 14 line 380-389, page 15 line 390-393 and page 16 line 437-445 of the revised manuscript:

*Long et al. (2016) proposed that the predominant pathway of unimolecular decay of syn-CH$_3$CHOO is isomerization to vinyl hydroperoxide (VHP) via the hydrogen atom migration from the methyl group to the terminal oxygen atom, then the decomposition of VHP produces OH radical. Also Donahue et al. (2011) obtained similar conclusion that syn-CI isomerization to VHP is preferable due to the low ring strain of H-atom transfer transition state. Both of the prompt and thermal unimolecular decay of the energized VHP may dissociate to OH radical, and their yields are strongly pressure and temperature dependents (Kroll et al., 2001a,b). The preferable route of*

*unimolecular decay of anti-CH₃CHOO is ring-closure to dioxirane via an oxygen atom transfer (Donahue et al., 2011; Taatjes et al., 2013; Long et al., 2016). The dioxirane can finally isomerize to acetic acid via the "hot acid" channel (Kroll et al., 2011b). Alternatively, the syn- and anti-CH₃CHOO may undergo bimolecular reactions with water vapor lead to the formation of HO-C(CH₃)HOO-H (Anglada et al., 2011).*

*Long et al. (2018) proposed that the unimolecular decay of (CH₃)₂COO is the predominant pathway above 240 K, whereas it can compete with the reaction (CH₃)₂COO + SO₂ below 240 K. Drozd et al. (2017) elucidated that tunneling for both the thermal and prompt unimolecular decay of (CH₃)₂COO is significant. Also Lester et al. (2018) obtained similar conclusion in the unimolecular decay of (CH₃)₂COO to OH radical reaction that the contribution of tunneling to the unimolecular decay rates is significant. Alternatively, the (CH₃)₂COO may react with water vapour leading to the formation of HO-C(CH₃)₂OO-H (Anglada et al., 2016).*

11. The energy labels in the supplement are confusing. The italics/non-italics energies are not always in same vertical order. Change the figure labels to make the non-italics energy the upper number in all cases. Do this for the main manuscript figures as well.

**Response:** Based on the Reviewer's suggestion, the non-italics energies have been placed in the upper number in the manuscript and supplement figures.

12. There are a number of awkward sentences/phrases, missing words, or grammatically confusing sentences. Carefully check over this document. I have listed a number of these below.

Line 27 change to "influence"

Line 43 Add Guenther et al as a reference for alkene emissions.5

Line 119 change to "is the dominant.."

132 change to "reactions occur during the…"

147 change to "there is little known of the reactivity…"

167 change to "represents"

168 change to "needs"

238 change to "possess OH and OOH"

246 change to "particularly"

267 - It is confusing to see a negative energy of stability, re-word this, and other instances, to be clearer.

315-317 You have an error the citations to Kroll et al. The year should be 2001.(6,7)

**Response:** Based on the Reviewer's suggestion, the sentences/phrases, missing words, and the grammatically confusing sentences have been corrected carefully in the revised manuscript.

---

## Author Comment (AC2) · 31 Jan 2019

Prof. Yu Huang Key Lab of Aerosol Chemistry & Physics, Institute of Earth Environment, Chinese Academy of Sciences, Xi'an, 710061, China Tel./Fax: (86) 29-62336261 E-mail: huangyu@ieecas.cn Jan. 31, 2019 Dear reviewer, Revision for Manuscript acp-2018-935 We thank you very much for giving us the opportunity to revise our manuscript. We highly appreciate the reviewer for their comments and suggestions on the manuscript entitled "Mechanistic and Kinetics Investigations of Oligomer Formation from Criegee Intermediates Reactions with Hydroxyalkyl Hydroperoxides". We

have made revisions of our manuscript carefully according to the comments and suggestions of reviewer. The revised contents are marked in blue color. The response letter to reviewers is attached at the end of this cover letter. We hope that the revised manuscript can meet the requirement of Atmospheric Chemistry & Physics. Any further modifications or revisions, please do not hesitate to contact us. Look forward to hearing from you as soon as possible.

Best regards, Yu Huang   Comments of reviewer #2 1. First of all, I think that the title of this study goes too far. The main focus of this study refers just to the reaction of Carbonyl oxides with HHPs, although a second step, namely the mechanisms for the interaction of the products of these reaction with Carbonyl oxides is taken into account. Response: The title of this study has been revised to "Mechanistic and Kinetics Investigations of Oligomer formation from Criegee Intermediates Reactions with Hydroxyalkyl Hydroperoxides". 2. Along the text, the authors refer to several reaction products, as for instance, P2c, P2b and so on, but the structure of these compounds is not mentioned, which makes the work difficult to follow. Response: Based on the Reviewer's suggestion, the structures of all reactants and products are added in the revised manuscript and supplement figures.

Figure 2. PES ($\Delta G$ and $\Delta E$ (italic)) for the reaction of CH2OO with HO-CH2OO-H (Pa1) computed at the M06-2X/def2-TZVP//M06-2X/6-311+G(2df,2p) level of theory

Figure 3. PES ($\Delta G$ and $\Delta E$ (italic)) for the reactions of HO-C(CH3)HOO-H with anti-(a) and syn-CH3CHOO(b) calculated at the M06-2X/def2-TZVP//M06-2X/6-311+G(2df,2p) level of theory

Figure 4. PES ($\Delta G$ and $\Delta E$ (italic)) for the reaction of (CH3)2COO with HO-C(CH3)2OO-H(Pa3) calculated at the M06-2X/def2-TZVP//M06-2X/6-311+G(2df,2p) level of theory

Figure 5. PES ($\Delta G$ and $\Delta E$ (italic)) of distinct SCI reactions with HO-CH2OO-H (Pa1) calculated at the M06-2X/def2-TZVP//M06-2X/6-311+G(2df,2p) level of theory

3. Some important references misses, as for instance CPL, 2001,337, 199, JPCA, 2001,105,446, JACS, 1997, 119, 330, CPC 2002, 2, 215, JPCA, 2003, 107, 5812, J. Atmos. Chem, 2000, 35, 165 and references therein. Response: The references on the unimolecular decay of HHPs generated from isoprene ozonolysis, and OH radicals production from alkene ozonolysis are added in the Introduction of the revised manuscript. Thanks for suggesting these closely related references, which have now been cited in the revised manuscript. The corresponding sentences have been added in the page 4 line 90-92, page 6 line 145-148 and page 3 line 57-61 of the revised manuscript in blue color: Winterhalter et al. (2001) studied the mechanism and products of gas phase ozonolysis of $\beta$-pinene, and found that the main products are the excited C9-CI plus HCHO. Aplincourt et al. (2003) investigated the unimolecular decay and water-catalyzed decomposition of HHPs generated from isoprene ozonolysis, and found that the main products are methyl vinyl ketone (MVK) or methacroleine (MAC) plus H2O2. The thermal unimolecular decay of vibrationally excited CIs is thought to be an important nonphotolytic source of atmospheric hydroxyl (OH) radicals, particularly in low light conditions, urban environments, and heavily forested areas (Lester et al., 2018; Foreman et al., 2016; Kidwell et al., 2016; Green et al., 2017; Zhang et al., 2002; Cremer et al., 2001; Anglada et al., 2002). 4. Regarding the theoretical approach, the authors state that all stationary points have been computing using the M06-2X functional, and for some selected elementary reactions they have performed single point energy calculations at CCSD(T) level of theory, pointing out that the deviations in the free energy barriers computed with both approaches range between 1.5-1.6 kcal/mol. The authors should clarify in which cases they have computed the energy barriers using both approaches, if they have taken into account basis set superposition corrections. They should compare the results of both approaches, for instance with results from the literature involving the reaction with water vapor (section 3.1) with results from the literature, where energy barriers are reported at CCSD(T)/CBS level of theory. Response: In the present study, the geometries of all stationary points on PES are optimized at the M06-2X/6-311+G(2df,2p) level of theory. For improved energies, single point calculations at

the M06-2X/def2-TZVP level are performed. In order to evaluate the reliability of M06-2X functional in computing energies, the single point energies of species involved in the some selected elementary reactions (R1a-R1d, R3a-R3d and R5a-R5d) are recalculated at the CCSD(T)/6-311+G(2df,2p) level based on the M06-2X/6-311+G(2df,2p) optimized geometries. Furthermore, the basis set superposition error (BSSE) is performed using the counterpoise method proposed by Boys and Bernardi (Boys et al., 1970). The electronic-energy ($\Delta Ea\#$) and free-energy ($\Delta Ga\#$) barriers comparisons of both approaches for considering BSSE correction and not considering are listed in Table S1. As shown in Table S1, the BSSE correction contributes to the barriers $\sim$ 1.4 (CCSD(T)) and $\sim$ 0.4 (M06-2X) kcalâ$\breve{A}$ćmol-1, respectively. The mean absolute deviations (MAD) of both approaches are 0.98 ($\Delta Ea\#$) and 0.96 ($\Delta Ga\#$) kcalâ$\breve{A}$ćmol-1 when without considering BSSE correction, while they are 0.38 ($\Delta Ea\#$) and 0.34 ($\Delta Ga\#$) kcalâ$\breve{A}$ćmol-1 when considering BSSE correction. The result shows that the M06-2X method in combination with the BSSE correction afford energies similar to those determined by the accurate and well recognized CCSD(T) level calculation. Considering the computational costs, the M06-2X/def2-TZVP method is selected to perform the single-point energy calculation for the title reaction system. For the bimolecular reaction of carbonyl oxide with water dimer, the barriers ($\Delta Ea\#$ and ($\Delta Ga\#$) are calculated using the M06-2X/def2-TZVP//M06-2X/6-311+G(2df,2p) method, and compared to the literature results. As shown in Table 1, the barrier differences between the computational and literature ones that were derived from the CCSD(T)/aug-cc-pVTZ//B3LYP/6-311+G(2df,2p) method (Anglada et al., 2016) are 0.5-1.0 kcal mol-1. Such discrepancies may be attributed to the different theoretical methods used in computing energies. The barrier of Entry 4 is 3.3 kcal mol-1, which is lower than the corresponding CCSD(T)/CBS result by 0.5 kcal mol-1 (Anglada et al., 2016). The results show that the M06-2X method provides energies similar to those determined by the CCSD(T) level calculation. Corresponding descriptions have been added in the page 8 line 198-215 and page 10 line 264-271 of the revised manuscript: In order to obtain a better evaluation on the reliability of M06-2X functional in computing energies, the single point

energies of species included in some selected elementary reactions (R1a-R1d, R3a-R3d and R5a-R5d) are recalculated at the CCSD(T)/6-311+G(2df,2p) level based on the M06-2X/6-311+G(2df,2p) optimized geometries. Furthermore, the basis set superposition error (BSSE) is performed using the counterpoise method described by Boys and Bernardi (1970) to estimate the stability of the pre-reactive complexes. The electronic-energy ($\Delta Ea\#$) and free-energy ($\Delta Ga\#$) barriers comparisons of both approaches for considering BSSE correction and not considering are listed in Table S1. As shown in Table S1, the contributions of BSSE corrections to the barriers are $\sim$ 1.4 (CCSD(T)) and $\sim$ 0.4 (M06-2X) kcal•mol-1, respectively. The mean absolute deviations (MAD) of both approaches are 0.98 ($\Delta Ea\#$) and 0.96 ($\Delta Ga\#$) kcal•mol-1 when without considering BSSE correction, while they become 0.38 ($\Delta Ea\#$) and 0.34 ($\Delta Ga\#$) kcal•mol-1 when considering BSSE correction. The result shows that the M06-2X method in combination with the BSSE correction afford energies similar to those determined by the accurate and well recognized CCSD(T) level calculation. Considering the computational costs, the M06-2X/def2-TZVP method is selected to perform the single-point energy calculation for the title reaction system. As shown in Table 1, the barrier differences between the computational and literature ones that were derived from the CCSD(T)/aug-cc-pVTZ//B3LYP/6-311+G(2df,2p) method (Anglada et al., 2016) are 0.5-1.0 kcal mol-1. Such discrepancies may be attributed to the different theoretical method used in computing energies. The barrier of Entry 4 is 3.3 kcal mol-1, which is lower than the corresponding CCSD(T)/CBS result by 0.5 kcal mol-1 (Anglada et al., 2016). The results show that the M06-2X method provides energies similar to those determined by the CCSD(T) level calculation. Table 1 Relative free-energies (kcal mol-1) for the stationary points and free-energy ($\Delta Ga\#$) barriers for the elementary pathways of distinct carbonyl oxides reactions with water dimer calculated at the M06-2X/def2-TZVP//M06-2X/6-311+G(2df,2p) level of theory. Labels A, B, C, and D are defined in Figure 1 Entry R1 R2 DO2H4O4H3 DO4H2O3H1 A B C D $\Delta Ga\#$ 1 H H -123.6 96.7 -2.5(-2.9) 0.2 -40.5 -39.9 2.7 [2.2] 2 H H 124.5 -94.9 -2.5(-3.0) 0.2 -39.6 -40.2 2.7 3 H H -143.8 -116.9 -2.2(-2.6) 0.2 -39.1 -40.2 2.4 4 H H 143.0 122.7 -1.9(-2.3) 1.4 -40.0 -40.3

3.3 {3.8} 5 CH3 H -126.2 100.9 -2.5(-3.0) 4.1 -32.4 -31.9 6.6 [6.0] 6 CH3 H 130.0 -90.4 -2.0(-2.5) 4.3 -31.6 -32.2 6.3 7 CH3 H -146.0 -116.3 -2.1(-2.6) 4.1 -30.9 -32.2 6.2 8 CH3 H 138.1 126.3 -2.3(-2.8) 4.7 -31.7 -31.9 7.0 9 H CH3 -122.1 95.0 -4.3(-4.8) 0.6 -36.6 -36.2 4.9 [3.9] 10 H CH3 125.6 -93.6 -3.9(-4.3) 0.7 -35.7 -36.5 4.6 11 H CH3 -138.1 -120.5 -4.4(-4.9) 1.1 -34.9 -36.5 5.5 12 H CH3 139.2 123.4 -3.6(-4.1) 2.0 -36.0 -36.2 5.6 13 CH3 CH3 -125.4 101.5 -4.6(-5.1) 4.2 -29.7 -29.1 8.8 [7.8] 14 CH3 CH3 128.5 -89.8 -4.2(-4.8) 4.5 -28.6 -29.7 8.7 15 CH3 CH3 -145.4 -117.6 -4.9(-5.3) 4.6 -28.2 -29.7 9.5 16 CH3 CH3 136.3 129.4 -4.5(-5.0) 5.2 -29.2 -29.1 9.7 Values in parenthesis correspond to without considering the BSSE correction, values in brackets correspond to CCSD(T)/aug-cc-pVTZ//B3LYP/6-311+G(2df,2p), values in braces correspond to CCSD(T)/CBS//B3LYP/6-311+G(2df,2p) Table S1 Y/X (Y = M06-2X, CCSD(T), X = 6-311+G(2df,2p), def2-TZVP) calculated energy barrier(ΔEa#, ΔGa#) for the addition reactions of carbonyl oxides with HHPs based on the M06-2X/6-311+G(2df,2p) optimized geometries (kcal•mol-1) Reactions CCSD(T)/6-311+G(2df,2p) M06-2X/def2-TZVP ΔEa# ΔGa# ΔEa# ΔGa# R1a 9.4a; 7.9b 10.1a; 8.6b 8.0a; 7.6b 8.8a; 8.4b R1b 13.0a; 11.6b 13.3a; 11.9b 11.9a; 11.4b 12.2a; 11.7b R1c 8.1a; 6.7b 9.2a; 7.8b 6.9a; 6.4b 7.9a; 7.4b R1d 8.6a; 7.6b 10.2a; 9.2b 7.0a; 6.7b 8.7a; 8.4b R3a 6.6a; 4.9b 7.1a; 5.4b 5.8a; 4.4b 6.2a; 5.8b R3b 8.4a; 7.3b 9.7a; 8.6b 7.3a; 6.8b 8.8a; 8.3b R3c 6.5a; 4.8b 7.3a; 5.6b 5.8a; 5.4b 6.6a; 6.2b R3d 8.4a; 6.8b 9.7a; 8.1b 7.3a; 6.8b 8.6a; 8.1b R5a 12.3a; 10.9b 13.2a; 11.8b 11.5a; 11.1b 12.5a; 12.1b R5b 11.0a; 9.4b 12.4a; 10.8b 10.6a; 10.1b 12.0a; 11.5b R5c 12.0a; 10.6b 13.3a; 11.9b 11.3a; 10.9b 12.5a; 12.1b R5d 12.2a; 11.0b 13.8a; 12.6b 11.4a; 11.0b 13.0a; 12.6b a and b represent without and with considering the BSSE correction 5. Regarding the kinetics, the authors should clarify if they have considered the pre-reactive complexes in the kinetic study and if they play a role in the temperature dependence of the rate constants. Response: As shown in Figure 5, the CH2OO + HO-CH2OO-H (Pa1) reaction proceeds according to a two-step mechanism: (i) a fast thermal equilibrium between the reactants and intermediate IM1a, (ii) the addition of CH2OO leading to the formation of product P1a. The whole reaction process is expressed as follows: (5) Applying the steady state approximation

(SSA) for the intermediate IM1a, the overall rate coefficient is extrapolated to the eqn (6): (Zhang et al., 2012; Liu et al., 2015) (6) If $k2 \ll k-1$, the overall rate coefficient is written as follows: (Ryzhkov et al., 2006; Chen et al., 2016) (7) The equilibrium coefficient Keq is expressed as eqn (8): (8) where $\sigma$ is reaction symmetry number, QIM(T), QA(T) and QB(T) denote the products of electronic, translational, rotational, torsional, and vibrational canonical partition functions for the intermediate, reactants A and B, respectively (Mendes, et al., 2014), T is the temperature in Kelvin, GR and GIM are the total free-energies of the reactant and intermediate, respectively. Table R1 lists the partition functions, equilibrium coefficients (Keq), and rate coefficients (k2(IM1a-TS1a) and kovr(R1a)) of the bimolecular reaction of CH2OO with HO-CH2OO-H (Pa1). As shown the Table R1, the partition functions of reactants and intermediate increase with raising temperature, whereas the kovr(R1a), Keq and k2(IM1a-TS1a) decrease with increasing temperature, indicating that there is not a direct correlation between kovr(R1a) and partition function of pre-reactive intermediate. The result shows that the kinetics parameters strongly depend on the interaction between Keq and k2(IM1a-TS1a). Similar conclusions are also obtained from those of the anti-CH3CHOO + HO-CH2OO-H(R9), syn-CH3CHOO + HO-CH2OO-H(R10) and (CH3)2CHOO + HO-CH2OO-H(R11) systems (Table R2-R4). Table R1 The partition function, equilibrium coefficient (Keq) (cm3 molecule-1), and rate coefficients (k2(IM1a-TS1a) and kovr(R1a)) (cm3 molecule-1 s-1) of CH2OO reactions with HO-CH2OO-H (Pa1) computed at different temperatures

| T/K | QCH2OO | QHOCH2OOH | QIMa | Keq | k2(IM1a-TS1a) | kovr(R1a) |
|---|---|---|---|---|---|---|
| 273 | $7.9 \times 10^{10}$ | $1.6 \times 10^{12}$ | $1.1 \times 10^{15}$ | $2.8 \times 10^{-6}$ | $4.3 \times 10^{-6}$ | $1.2 \times 10^{-11}$ |
| 280 | $8.8 \times 10^{10}$ | $1.9 \times 10^{12}$ | $1.5 \times 10^{15}$ | $2.4 \times 10^{-6}$ | $4.0 \times 10^{-6}$ | $9.4 \times 10^{-12}$ |
| 298 | $1.2 \times 10^{11}$ | $2.7 \times 10^{12}$ | $2.7 \times 10^{15}$ | $1.6 \times 10^{-6}$ | $3.3 \times 10^{-6}$ | $5.4 \times 10^{-12}$ |
| 300 | $1.2 \times 10^{11}$ | $2.8 \times 10^{12}$ | $2.9 \times 10^{15}$ | $1.6 \times 10^{-6}$ | $3.2 \times 10^{-6}$ | $5.1 \times 10^{-12}$ |
| 320 | $1.6 \times 10^{11}$ | $4.2 \times 10^{12}$ | $5.6 \times 10^{15}$ | $1.1 \times 10^{-6}$ | $2.7 \times 10^{-6}$ | $3.0 \times 10^{-12}$ |
| 340 | $2.1 \times 10^{11}$ | $6.2 \times 10^{12}$ | $1.1 \times 10^{16}$ | $8.1 \times 10^{-7}$ | $2.3 \times 10^{-6}$ | $1.9 \times 10^{-12}$ |
| 360 | $2.7 \times 10^{11}$ | $9.0 \times 10^{12}$ | $2.0 \times 10^{16}$ | $6.2 \times 10^{-7}$ | $2.0 \times 10^{-6}$ | $1.2 \times 10^{-12}$ |
| 380 | $3.5 \times 10^{11}$ | $1.3 \times 10^{13}$ | $3.6 \times 10^{16}$ | $4.8 \times 10^{-7}$ | $1.8 \times 10^{-6}$ | $8.6 \times 10^{-13}$ |
| 400 | $4.5 \times 10^{11}$ | $1.8 \times 10^{13}$ | $6.6 \times 10^{16}$ | $3.9 \times 10^{-7}$ | $1.6 \times 10^{-6}$ | $6.3 \times 10^{-13}$ |

Table R2 The partition function, equilibrium coefficient (Keq) (cm3 molecule-1), and rate coefficients (k2(IM9-TS9) and kovr(R9)) (cm3 molecule-1 s-1) of anti-CH3CHOO reactions with HO-CH2OO-H (Pa1) computed at different temperatures T/K Qanti-CH3CHOO QHOCH2OOH QIM9 Keq k2(IM9-TS9) kovr(R9) 273 1.4 × 1012 1.6 × 1012 9.1 × 1015 1.3 × 10-4 4.4 × 10-5 5.5 × 10-9 280 1.6 × 1012 1.9 × 1012 1.2 × 1016 9.6 × 10-5 3.9 × 10-5 3.7 × 10-9 298 2.4 × 1012 2.7 × 1012 2.5 × 1016 5.0 × 10-5 2.9 × 10-5 1.5 × 10-9 300 2.5 × 1012 2.8 × 1012 2.7 × 1016 4.7 × 10-5 2.9 × 10-5 1.3 × 10-9 320 3.7 × 1012 4.2 × 1012 5.8 × 1016 2.5 × 10-5 2.2 × 10-5 5.5 × 10-10 340 5.4 × 1012 6.2 × 1012 1.2 × 1017 1.5 × 10-5 1.7 × 10-5 2.6 × 10-10 360 7.8 × 1012 9.0 × 1012 2.5 × 1017 9.1 × 10-6 1.4 × 10-5 1.3 × 10-10 380 1.1 × 1013 1.3 × 1013 5.1 × 1017 5.9 × 10-6 1.2 × 10-5 7.1 × 10-11 400 1.6 × 1013 1.8 × 1013 1.0 × 1018 4.1 × 10-6 1.0 × 10-5 4.1 × 10-11

Table R3 The partition function, equilibrium coefficient (Keq) (cm3 molecule-1), and rate coefficients (k2(IM10-TS10) and kovr(R10)) (cm3 molecule-1 s-1) of syn-CH3CHOO reactions with HO-CH2OO-H (Pa1) computed at different temperatures T/K Qsyn-CH3CHOO QHOCH2OOH QIM10 Keq k2(IM10-TS10) kovr(R10) 273 1.0 × 1012 1.6 × 1012 6.6 × 1015 1.5 × 10-6 5.6 × 10-9 8.1 × 10-15 280 1.2 × 1012 1.9 × 1012 8.8 × 1015 1.2 × 10-6 6.0 × 10-9 7.5 × 10-15 298 1.7 × 1012 2.7 × 1012 1.8 × 1016 8.6 × 10-7 7.3 × 10-9 6.3 × 10-15 300 1.8 × 1012 2.8 × 1012 1.9 × 1016 8.3 × 10-7 7.5 × 10-9 6.2 × 10-15 320 2.6 × 1012 4.2 × 1012 4.1 × 1016 5.8 × 10-7 9.1 × 10-9 5.3 × 10-15 340 3.7 × 1012 6.2 × 1012 8.7 × 1016 4.3 × 10-7 1.1 × 10-8 4.6 × 10-15 360 5.3 × 1012 9.0 × 1012 1.8 × 1017 3. 3 × 10-7 1.3 × 10-8 4.1 × 10-15 380 7.5 × 1012 1.3 × 1013 3.6 × 1017 2.6 × 10-7 1.5 × 10-8 3.8 × 10-15 400 1.1 × 1013 1.8 × 1013 7.2 × 1017 2.1 × 10-7 1.7 × 10-8 3. 5 × 10-15

Table R4 The partition function, equilibrium coefficient (Keq) (cm3 molecule-1), and rate coefficients (k2(IM11-TS11) and kovr(R11)) (cm3 molecule-1 s-1) of (CH3)2CHOO reactions with HO-CH2OO-H (Pa1) computed at different temperatures T/K Q(CH3)2CHOO QHOCH2OOH QIM11 Keq k2(IM10-TS10) kovr(R10) 273 7.2 ×

1012 1.6 × 1012 4.0 × 1016 8.8 × 10-5 3.1 × 10-8 2.7 × 10-12 280 8.5 × 1012 1.9 × 1012 5.5 × 1016 6.9 × 10-5 3.4 × 10-8 2.3 × 10-12 298 1.3 × 1013 2.7 × 1012 1.2 × 1017 3.7 × 10-5 4.1 × 10-8 1.5 × 10-12 300 1.4 × 1013 2.8 × 1012 1.3 × 1017 3.5 × 10-5 4.2 × 10-8 1.5 × 10-12 320 2.2 × 1013 4.2 × 1012 3.2 × 1017 1.9 × 10-5 5.2 × 10-8 1.0 × 10-12 340 3.5 × 1013 6.2 × 1012 7.3 × 1017 1.2 × 10-5 6.2 × 10-8 7.2 × 10-13 360 5.5 × 1013 9.0 × 1012 1.7 × 1018 7.4 × 10-6 7.4 × 10-8 5.4 × 10-13 380 8.6 × 1013 1.3 × 1013 3.8 × 1018 5.0 × 10-6 8.6 × 10-8 4.2 × 10-13 400 1.3 × 1014 1.8 × 1013 8.3 × 1018 3.5 × 10-6 9.8 × 10-8 3.4 × 10-13 The corresponding sentences have been added in the page 9 line 219-237 of the revised manuscript in blue color: As shown in Figure 5, the CH2OO + HO-CH2OO-H (Pa1) reaction proceeds according to a two-step mechanism: (i) a fast thermal equilibrium between the reactants and intermediate IM1a, (ii) the addition of CH2OO leading to the formation of product P1a. The whole reaction process is expressed as follows: (4) Applying the steady state approximation (SSA) for the intermediate IM1a, the overall rate coefficient is extrapolated to the eqn (5) (Zhang et al., 2012; Liu et al., 2015) (5) If k2 « k-1, the overall rate coefficient is written as follows: (Chen et al., 2016b; Ryzhkov et al., 2006) (6) The equilibrium coefficient Keq is expressed as eqn (7): (7) where $\sigma$ is reaction symmetry number, QIM(T), QA(T) and QB(T) denote the products of electronic, translational, rotational, torsional, and vibrational canonical partition functions for the intermediate, reactants A and B, respectively (Mendes et al., 2014), T is the temperature in Kelvin, GR and GIM are the total free-energies of the reactant and complex, respectively. 6. The authors report rate constants for the reactions of the carbonyl oxides considered with HHP's (Table 2), but no mention is done for the reactions of P1x with Carbonyl oxides. Moreover, that authors should clarify if they have considered all different conformers of the stationary points in the kinetic study. In addition, they should estimate the errors in the these calculated rate constants, since they can be between one and two orders of magnitude according to the errors in the computed free energy barriers. Response: Based on the Reviewer's suggestion, the rate coefficients of carbonyl oxides reactions with P1x, P3x, P5x, and P7x are calculated using a combination of canonical transition

state theory (CTST) and an asymmetric Eckart tunneling correction at 273-400 K. And the different conformers of stationary points in the kinetics study are taken into account, with the obtained results listed in Table S5-S8. As shown in Table S5, the predicted rate coefficients for the reaction of CH2OO with P1a decrease with increasing temperature, with a similar trend observed for CH2OO + P1b, CH2OO + P1c and CH2OO + P1d systems. The result implies that the oligomer formation from CH2OO reaction with HHP is preferable under low temperature conditions. Similar conclusions are obtained from the anti-CH3CHOO + P3x (Table S6), syn-CH3CHOO + P5x (Table S7) and (CH3)2CHOO + P7x (Table S8) systems. In order to avoid redundancy, we do not repeat them here in detail. Considering the errors of the computed free energy barriers, the uncertainty of rate coefficient is estimated within an order of magnitude. Corresponding descriptions have been added in the page 18 line 500-506 and page 19 line 521-528 of the revised manuscript: the rate coefficients of distinct SCI reactions with HO-CH2OO-H (Pa1) are computed using a combination of canonical transition state theory (CTST) and an asymmetric Eckart tunneling correction based on the free energies obtained at the M06-2X level, in the temperature range from 273 to 400 K. And the different conformers of stationary points in the kinetics study are taken into account, with the results listed in Table 2 and Table S5-S8. As shown in Table S5, the rate coefficients of CH2OO + P1a, CH2OO + P1b, CH2OO + P1c and CH2OO + P1d reactions decrease with increasing temperature, indicating that the oligomer formation from CH2OO reactions with HHP is preferable under low temperature conditions. Similar conclusions are also obtained from the anti-CH3CHOO + P3x (Table S6), syn-CH3CHOO + P5x (Table S7) and (CH3)2CHOO + P7x (Table S8) systems. In order to avoid redundancy, we do not repeat them here in detail. Considering the errors of the computed free energy barriers, the uncertainty of rate coefficient is estimated within an order of magnitude. Table S5 Rate coefficients (cm3 molecule-1 s-1) of CH2OO reactions with P1a, P1b, P1c and P1d computed at different temperatures T/K k(CH2OO+P1a) k(CH2OO+P1b) k(CH2OO+P1c) k(CH2OO+P1d) 273 5.1 × 10-12 3.5 × 10-12 8.2 × 10-13 4.8 × 10-11 280 4.0 × 10-12 2.9 × 10-12 7.1 × 10-13 3.7 × 10-11 298 2.3 × 10-12 1.9 × 10-12

5.2 × 10-13 1.9 × 10-11 300 2.2 × 10-12 1.8 × 10-12 5.0 × 10-13 1.8 × 10-11 320 1.3 × 10-12 1.2 × 10-12 3.7 × 10-13 1.0 × 10-11 340 8.3 × 10-13 8.5 × 10-13 2.8 × 10-13 5.9 × 10-12 360 5.6 × 10-13 6.3 × 10-13 2.3 × 10-13 3.8 × 10-12 380 4.0 × 10-13 4.9 × 10-13 1.9 × 10-13 2.5 × 10-12 400 3.0 × 10-13 3.9 × 10-13 1.6 × 10-13 1.8 × 10-12

Table S6 Rate coefficients (cm3 molecule-1 s-1) of anti-CH3CHOO reactions with P3a, P3b, P3c and P3d computed at different temperatures T/K k(anti+P3a) k(anti+P3b) k(anti+P3c) k(anti+P3d) 273 3.5 × 10-9 2.3 × 10-9 1.7 × 10-7 2.1 × 10-9 280 2.3 × 10-9 1.4 × 10-9 1.0 × 10-7 1.4 × 10-9 298 8.8 × 10-10 5.8 × 10-10 2.9 × 10-8 5.9 × 10-10 300 8.0 × 10-10 5.3 × 10-10 2.6 × 10-8 5.4 × 10-10 320 3.2 × 10-10 2.3 × 10-10 7.9 × 10-9 2.3 × 10-10 340 1.4 × 10-10 1.1 × 10-10 2.8 × 10-9 1.1 × 10-10 360 7.1 × 10-11 5.8 × 10-11 1.1 × 10-9 5.9 × 10-11 380 3.8 × 10-11 3.3 × 10-11 4.9 × 10-10 3.4 × 10-11 400 2.2 × 10-11 2.0 × 10-11 2.4 × 10-10 2.0 × 10-11

Table S7 Rate coefficients (cm3 molecule-1 s-1) of syn-CH3CHOO reactions with P5a, P5b, P5c and P5d computed at different temperatures T/K k(syn+P5a) k(syn+P5b) k(syn+P5c) k(syn+P5d) 273 2.1 × 10-11 1.7 × 10-13 2.1 × 10-11 2.4 × 10-13 280 1.5 × 10-11 1.4 × 10-13 1.5 × 10-11 2.0 × 10-13 298 7.5 × 10-12 1.0 × 10-13 7.6 × 10-12 1.4 × 10-13 300 6.9 × 10-12 9.7 × 10-14 7.0 × 10-12 1.4 × 10-13 320 3.5 × 10-12 6.9 × 10-14 3.6 × 10-12 9.6 × 10-14 340 1.9 × 10-12 5.2 × 10-14 2.0 × 10-12 7.1 × 10-14 360 1.1 × 10-12 4.0 × 10-14 1.2 × 10-12 5.5 × 10-14 380 7.1 × 10-13 3.0 × 10-14 7.3 × 10-13 4.4 × 10-14 400 4.7 × 10-13 2.6 × 10-14 4.9 × 10-13 3.6 × 10-14

Table S8 Rate coefficients (cm3 molecule-1 s-1) of (CH3)2CHOO reactions with P7a, P7b, P7c and P7d computed at different temperatures T/K k((CH3)2CHOO +P7a) k((CH3)2CHOO +P7b) k((CH3)2CHOO +P7c) k((CH3)2CHOO +P7d) 273 7.8 × 10-14 9.1 × 10-14 1.8 × 10-12 4.5 × 10-13 280 6.8 × 10-14 8.2 × 10-14 1.5 × 10-12 3.9 × 10-13 298 5.0 × 10-14 6.6 × 10-14 1.0 × 10-12 2.8 × 10-13 300 4.8 × 10-14 6.5 × 10-14 9.9 × 10-13 2.7 × 10-13 320 3.6 × 10-14 5.3 × 10-14 6.9 × 10-13

[Figure]

1.9 × 10-13 340 2.8 × 10-14 4.5 × 10-14 5.1 × 10-13 1.5 × 10-13 360 2.2 × 10-14 3.9 × 10-14 3.9 × 10-13 1.2 × 10-13 380 1.9 × 10-14 3.5 × 10-14 3.1 × 10-13 9.6 × 10-14 400 1.6 × 10-14 3.1 × 10-14 2.5 × 10-13 8.0 × 10-14 7. With respect to the atmospheric implications, the authors compare the reaction rates of the reaction investigated with those between carbonyl oxides with formic acid. In my opinion, the reactions rates of carbonyl oxides with water and water dimer, but also the reactions rates of HHPs with water should be also taken into account, because the high concentration of water vapor in the atmosphere. For the last, there are free energy barriers in the literature to compare with. Response: As shown in Table 2, the rate coefficient of anti-CH3CHOO + Pa1 reaction (R9) is significantly higher than that of the other three pathways (R1a, R10 and R11). Therefore, it would be interesting to investigate whether the anti-CH3CHOO + Pa1 reaction can compete well with the anti-CH3CHOO + (H2O)2 (R12) system because the latter reaction is the dominant chemical sink (Anglada et al., 2016; Taatjes, et al., 2013). The ratio of reaction rates of R9 and R12 is expressed as follows (9) The room temperature rate coefficient kR9 is 1.5 × 10-9 cm3 molecule-1 s-1. Assuming that the concentration of Pa1 is approximately equal to that of SCIs (∼ 5.0 × 104 molecules•cm-3, within an order of magnitude uncertainty) in the boreal forest and rural environments of Finland and Germany (Novelli et al, 2016, 2017). The atmospheric lifetime of anti-CH3CHOO reactivity toward Pa1 can be estimated as 1.3-13 × 103 s. The experimental rate coefficient of reaction R12 approximately equals ∼ 1.0 × 10-11 cm3 molecule-1 s-1 at 298 K (Lin et al., 2016). The concentration of water dimer is 5.5 × 1013 molecules•cm-3 at 3 km altitude (Long et al., 2016). The $\nu$R9/$\nu$R12 ratio is less than 1.4%, meaning that the anti-CH3CHOO + Pa1 reaction is minor loss process in the atmosphere. However, the [(H2O)2] is very low at the altitude above 15 km ( < 2.7 × 106 molecules cm-3) (Long et al., 2016), the anti-CH3CHOO + Pa1 reaction can compete well with the anti-CH3CHOO + (H2O)2 reaction, and thus contribute to the formation and growth of SOA. ' Kumar et al. (2014) proposed that the gas-phase decomposition of Pa1 has two competitive pathways, namely (i) HO-CH2OO-H → CH2O + H2O2 and (ii) HO-CH2OO-H → HCOOH + H2O.

The free energy barriers ∆Ga# in the presence of a single water molecule are 31.2 and 47.8 kcal•mol-1, respectively, which are 13.5 and 10.2 kcal•mol-1 lower than the uncatalyzed reactions. The result reveals that the formaldehyde-forming channel is preferable in the absence and presence of water molecule, and the role of water catalysis on the gas-phase Pa1 decomposition is significant. The ∆Ga# of bimolecular reaction of anti-CH3CHOO with Pa1 is 7.3 kcal•mol-1, which is 23.9 kcal•mol-1 lower than the formaldehyde-forming channel. It is concluded that the Pa1 + H2O reaction is less competitive as compared to the anti-CH3CHOO + Pa1 system. Corresponding descriptions have been added in the page 20 line 542-556, page 20 line 565 and page 21 line 566-579 of the revised manuscript: As discussed above, the anti-CH3CHOO + HO-CH2OO-H (Pa1) reaction (R9) is preferred over the other three pathways (R1a, R10 and R11). Therefore, it would be interesting to investigate whether the anti-CH3CHOO + Pa1 reaction can compete well with the anti-CH3CHOO + (H2O)2 (R12) system because the latter reaction is the dominant chemical sink (Taatjes et al., 2013; Anglada et al., 2016). The ratio of reaction rates of R9 and R12 is expressed as follows (10) The room temperature rate coefficient kR9 is $1.5 \times 10^{-9}$ cm3 molecule-1 s-1. Assuming that the concentration of Pa1 is approximately equal to that of SCIs ($\sim 5.0 \times 10^4$ molecules•cm-3, within an order of magnitude uncertainty) in the boreal forest and rural environments of Finland and Germany (Novelli et al., 2016; 2017). The atmospheric lifetime of anti-CH3CHOO reactivity toward Pa1 can be estimated as $1.3\text{-}13 \times 10^3$ s. The experimental rate coefficient of reaction R12 approximately equals $\sim 1.0 \times 10^{-11}$ cm3 molecule-1 s-1 at 298 K (Lin et al., 2016). The concentration of water dimer is $5.5 \times 10^{13}$ molecules•cm-3 at 3 km altitude (Long et al., 2016). The $\nu$R9/$\nu$R12 ratio is less than 1.4% when the [(H2O)2] is $\sim 10^{13}$ molecules cm-3, meaning that the anti-CH3CHOO + Pa1 reaction is minor loss process in the atmosphere. However, the [(H2O)2] is very low at the altitude above 15 km, the anti-CH3CHOO + Pa1 reaction can compete well with the anti-CH3CHOO + (H2O)2 reaction, and thus contribute to the formation and growth of SOA. Kumar et al. (2014) proposed that the gas-phase decomposition of Pa1 has two competitive pathways: (i) HO-CH2OO-H →

CH2O + H2O2 and (ii) HO-CH2OO-H → HCOOH + H2O. The ΔGa# in the presence of a single water molecular are 31.2 and 47.8 kcal•mol-1, respectively, which are 13.5 and 10.2 kcal•mol-1 lower than the uncatalyzed reactions. The result reveals that the formaldehyde-forming channel is preferable in the absence and presence of water molecule, and the role of water catalysis on the gas-phase decomposition of Pa1 is significant. The ΔGa# of bimolecular reaction of anti-CH3CHOO with Pa1 is 7.3 kcal•mol-1, which is 23.9 kcal•mol-1 lower than the formaldehyde-forming channel. It is concluded that the Pa1 + H2O reaction is less competitive as compared to the anti-CH3CHOO + Pa1 system. 8. An hydrogen misses in the structure of P1a in Figure 2. In addition some addition structures of the P2x compounds should be drawn if the different figures and the numbers should have a larger size. Response: Based on the Reviewer's suggestion, the PES of CH2OO reaction with HO-CH2OO-H (Pa1) is redrawn in Figure 2, and the str in the Figure 2.

Figure 2. PES (ΔG and ΔE (italic)) for the reaction of CH2OO with HO-CH2OO-H (Pa1) computed at the M06-2X/def2-TZVP//M06-2X/6-311+G(2df,2p) level of theory   References Anglada, J. M., and Solé, A.: Impact of water dimer on the atmospheric reactivity of carbonyl oxides, Phys. Chem. Chem. Phys., 18, 17698-17712, 10.1039/c6cp02531e, 2016. Boys, S. F., and Bernardi, F.: The calculation of small molecular interactions by the differences of separate total energies. Some procedures with reduced errors, Mol. Phys., 19, 553-566, 10.1080/00268977000101561, 1970. Chen, L., Wang. W. L., Zhou, L. T., Wang, W. N., Liu, F. Y., Li, C. Y., and Lü, J.: Role of water clusters in the reaction of the simplest Criegee intermediate CH2OO with water vapour, Theor. Chem. Acc., 135, 252-263, 10.1007/s00214-016-1998-2, 2016. Kumar, M., Busch, D. H., Subramaniam, B., and Thompson, W. H.: Role of tunable acid catalysis in decomposition of α-hydroxyalkyl hydroperoxides and mechanistic implications for tropospheric chemistry, J. Phys. Chem. A, 118, 9701-9711, 10.1021/jp505100x, 2014. Lin, L. C., Chang, H. T., Chang, C. H., Chao, W., Smith, M. C., Chang, C. H., Lin, J. J. M., and Takahashi, K.: Competition between H2O and (H2O)2 reactions with CH2OO/CH3CHOO, Phys. Chem. Chem. Phys., 18, 4557-

4568, 10.1039/C5CP06446E, 2016. Liu, J., Fang, S., Wang, Z., Yi, W., Tao, F. M., and Liu, J. Y.: Hydrolysis of sulfur dioxide in small clusters of sulfuric acid: mechanistic and kinetic study, Environ. Sci. Technol., 49, 13112-13120, 10.1021/acs.est.5b02977, 2015 Long, B., Bao, J. L., and Truhlar, D. G.: Atmospheric chemistry of Criegee intermediates: unimolecular reactions and reactions with water, J. Am. Chem. Soc., 138, 14409-14422, 10.1021/jacs.6b08655, 2016. Mendes, J., Zhou, C. W., and Curran, H. J.: Theoretical chemical kinetic study of the H-atom abstraction reactions from aldehydes and acids by H atoms and OH, HO2, and CH3 radicals, J. Phys. Chem. A, 118, 12089-12104, 10.1021/jp5072814, 2014. Novelli, A., Hens, K., Ernest, C. T., Martinez, M., Nölscher, A. C., Sinha, V., Paasonen, P., Petäjä, T., Sipilä, M., Elste, T., Plass-Dülmer, C., Phillips, G. J., Kubistin, D., Williams, J., Vereecken, L., Lelieveld, J., and Harder, H.: Identifying Criegee intermediates as potential oxidants in the troposphere, Atmos. Chem. Phys. Discuss., 10.5194/acp-2016-919, 2016. Novelli, A., Hens, K., Ernest, C. T., Martinez, M., Nölscher, A. C., Sinha, V., Paasonen, P., Petäjä, T., Sipilä, M., Elste, T., Plass-Dülmer, C., Phillips, G. J., Kubistin, D., Williams, J., Vereecken, L., Lelieveld, J., and Harder, H.: Estimating the atmospheric concentration of Criegee intermediates and their possible interference in a FAGE-LIF instrument, Atmos. Chem. Phys., 17, 7807-7826, 10.5194/acp-17-7807-2017, 2017. Ryzhkov, A. B., and Ariya, P. A.: The importance of water clusters (H2O)n (n=2, ... ,4) in the reaction of Criegee intermediate with water in the atmosphere, Chem. Phys. Lett., 419, 479-485, 10.1016/j.cplett.2005.12.016, 2006. Taatjes, C. A., Welz, O., Eskola, A. J., Savee, J. D., Scheer, A. M., Shallcross, D. E., Rotavera, B., Lee, E. P. F., Dyke, J. M., Mok, D. K. W., Osborn, D. L., and Percival, C. J.: Direct measurements of conformer-dependent reactivity of the Criegee intermediate CH3CHOO, Science, 340, 177-180, 10.1126/science.1234689, 2013. Zhang, P., Wang, W. L., Zhang, T. L., Chen, L., Du, Y. M., Li, C. Y., and Lü, J.: Theoretical study on the mechanism and kinetics for the self-reaction of C2H5O2 radicals, J. Phys. Chem. A, 116, 4610-4620, 10.1021/jp301308u, 2012.

Please also note the supplement to this comment:
https://www.atmos-chem-phys-discuss.net/acp-2018-935/acp-2018-935-AC2-
supplement.pdf
* * *
* * *
[Figure]

[Figure]

**Fig. 1.**

[Figure]

**Fig. 2.**

$\Delta G/\Delta E$ (kcal mol$^{-1}$)

(b)

(0.0)
Pa$_2$(syn-HO-C(CH$_3$)HOO-H)
+ 2syn-CH$_3$CHOO

(-2.4)
(-13.6)
IM5b

IM5a
(-2.4)
(-12.9)

(10.1)
(-1.4)
TS5a
TS5b
(9.6)
(-3.0)

Pa$_2$(syn-HO-C(CH$_3$)HOO-H)

syn-CH$_3$CHOO

P5a

P5b

syn-CH$_3$CHOO

(-33.2)
(-32.2)
P5a+syn-

(-34.3)
(-46.9)
P5b+syn-

(-38.0)
(-49.1)

IM6a
IM6b
(-38.8)
(-48.9)

P6c

TS6b
(-17.7)
(-28.1)

TS6a
(-27.0)
(-38.9)

IM6c
(-37.3)
(-61.1)
IM6d
(-39.6)
(-64.6)

P6a

P6d

TS6d
(-23.9)
(-49.4)
TS6c
(-25.7)
(-50.6)

P6b

(-58.0)
(-68.4)
P6b

(-71.1)
(-83.6)
P6a

(-70.3)
(-95.9)
P6d

(-73.4)
(-98.8)
P6c

**Fig. 3.**

[Figure]

**Fig. 4.**

[Figure]

**Fig. 5.**

**Supplement:**

Prof. Yu Huang
Key Lab of Aerosol Chemistry & Physics,
Institute of Earth Environment, Chinese Academy
of Sciences, Xi'an, 710061, China
Tel./Fax: (86) 29-62336261
E-mail: huangyu@ieecas.cn

Jan. 31, 2019

Dear reviewer,

**Revision for Manuscript acp-2018-935**

We thank you very much for giving us the opportunity to revise our manuscript. We highly appreciate the reviewer for their comments and suggestions on the manuscript entitled "**Mechanistic and Kinetics Investigations of Oligomer Formation from Criegee Intermediates Reactions with Hydroxyalkyl Hydroperoxides**". We have made revisions of our manuscript carefully according to the comments and suggestions of reviewer. The revised contents are marked in blue color. The response letter to reviewers is attached at the end of this cover letter.

We hope that the revised manuscript can meet the requirement of Atmospheric Chemistry & Physics. Any further modifications or revisions, please do not hesitate to contact us.

Look forward to hearing from you as soon as possible.

Best regards,

Yu Huang

**Comments of reviewer #2**

1. First of all, I think that the title of this study goes too far. The main focus of this study refers just to the reaction of Carbonyl oxides with HHPs, although a second step, namely the mechanisms for the interaction of the products of these reaction with Carbonyl oxides is taken into account.

**Response:** The title of this study has been revised to "Mechanistic and Kinetics Investigations of Oligomer formation from Criegee Intermediates Reactions with Hydroxyalkyl Hydroperoxides".

2. Along the text, the authors refer to several reaction products, as for instance, P2c, P2b and so on, but the structure of these compounds is not mentioned, which makes the work difficult to follow.

**Response:** Based on the Reviewer's suggestion, the structures of all reactants and products are added in the revised manuscript and supplement figures.

[Figure]

**Figure 2.** PES (ΔG and ΔE (italic)) for the reaction of CH₂OO with HO-CH₂OO-H (Pa₁) computed at the M06-2X/def2-TZVP//M06-2X/6-311+G(2df,2p) level of theory

[Figure]

**Figure 3.** PES (*ΔG and ΔE (italic)*) for the reactions of HO-C(CH₃)HOO-H with anti-(a) and syn-CH₃CHOO(b) calculated at the M06-2X/def2-TZVP//M06-2X/6-311+G(2df,2p) level of theory

[Figure]

***Figure 4.*** *PES (ΔG and ΔE (italic)) for the reaction of (CH₃)₂COO with HO-C(CH₃)₂OO-H(Pa₃) calculated at the M06-2X/def2-TZVP//M06-2X/6-311+G(2df,2p) level of theory*

[Figure]

***Figure 5.*** *PES (ΔG and ΔE (italic)) of distinct SCI reactions with HO-CH₂OO-H (Pa₁) calculated at the M06-2X/def2-TZVP//M06-2X/6-311+G(2df,2p) level of theory*

3. Some important references misses, as for instance CPL, 2001,337, 199, JPCA, 2001,105,446, JACS, 1997, 119, 330, CPC 2002, 2, 215, JPCA, 2003, 107, 5812, J. Atmos. Chem, 2000, 35, 165 and references therein.

**Response:** The references on the unimolecular decay of HHPs generated from isoprene ozonolysis, and OH radicals production from alkene ozonolysis are added in the Introduction of the revised manuscript. Thanks for suggesting these closely related references, which have now been cited in the revised manuscript.

The corresponding sentences have been added in the page 4 line 90-92, page 6 line 145-148 and page 3 line 57-61 of the revised manuscript in blue color:

*Winterhalter et al. (2001) studied the mechanism and products of gas phase ozonolysis of β-pinene, and found that the main products are the excited C9-CI plus HCHO. Aplincourt et al. (2003) investigated the unimolecular decay and water-catalyzed decomposition of HHPs generated from isoprene ozonolysis, and found that the main products are methyl vinyl ketone (MVK) or methacroleine (MAC) plus $H_2O_2$. The thermal unimolecular decay of vibrationally excited CIs is thought to be an important nonphotolytic source of atmospheric hydroxyl (OH) radicals, particularly in low light conditions, urban environments, and heavily forested areas (Lester et al., 2018; Foreman et al., 2016; Kidwell et al., 2016; Green et al., 2017; Zhang et al., 2002; Cremer et al., 2001; Anglada et al., 2002).*

4. Regarding the theoretical approach, the authors state that all stationary points have been computing using the M06-2X functional, and for some selected elementary reactions they have performed single point energy calculations at CCSD(T) level of theory, pointing out that the deviations in the free energy barriers computed with both approaches range between 1.5-1.6 kcal/mol. The authors should clarify in which cases they have computed the energy barriers using both approaches, if they have taken into account basis set superposition corrections. They should compare the results of both approaches, for instance with results from the literature involving the reaction with water vapor (section 3.1) with results from the literature, where energy barriers are reported at CCSD(T)/CBS level of theory.

**Response:** In the present study, the geometries of all stationary points on PES are optimized at the M06-2X/6-311+G(2df,2p) level of theory. For improved energies, single point calculations at the M06-2X/def2-TZVP level are performed. In order to evaluate the reliability of M06-2X functional in computing energies, the single point energies of species involved in the some selected elementary reactions (R1a-R1d, R3a-R3d and R5a-R5d) are recalculated at the CCSD(T)/6-311+G(2df,2p) level based on the M06-2X/6-311+G(2df,2p) optimized geometries. Furthermore, the basis set superposition error (BSSE) is performed using the counterpoise method proposed by Boys and Bernardi (Boys et al., 1970). The electronic-energy ($\Delta E_a^{\#}$) and free-energy ($\Delta G_a^{\#}$) barriers comparisons of both approaches for considering BSSE correction and not considering are listed in Table S1. As shown in Table S1, the BSSE correction contributes to the barriers ~ 1.4 (CCSD(T)) and ~ 0.4 (M06-2X) kcal mol$^{-1}$, respectively. The mean absolute deviations (MAD) of both approaches are 0.98 ($\Delta E_a^{\#}$) and 0.96 ($\Delta G_a^{\#}$) kcal mol$^{-1}$ when without

considering BSSE correction, while they are 0.38 ($\Delta E_a^{\#}$) and 0.34 ($\Delta G_a^{\#}$) kcal mol$^{-1}$ when considering BSSE correction. The result shows that the M06-2X method in combination with the BSSE correction afford energies similar to those determined by the accurate and well recognized CCSD(T) level calculation. Considering the computational costs, the M06-2X/def2-TZVP method is selected to perform the single-point energy calculation for the title reaction system.

For the bimolecular reaction of carbonyl oxide with water dimer, the barriers ($\Delta E_a^{\#}$ and ($\Delta G_a^{\#}$) are calculated using the M06-2X/def2-TZVP//M06-2X/6-311+G(2df,2p) method, and compared to the literature results. As shown in Table 1, the barrier differences between the computational and literature ones that were derived from the CCSD(T)/aug-cc-pVTZ//B3LYP/6-311+G(2df,2p) method (Anglada et al., 2016) are 0.5-1.0 kcal mol$^{-1}$. Such discrepancies may be attributed to the different theoretical methods used in computing energies. The barrier of Entry 4 is 3.3 kcal mol$^{-1}$, which is lower than the corresponding CCSD(T)/CBS result by 0.5 kcal mol$^{-1}$ (Anglada et al., 2016). The results show that the M06-2X method provides energies similar to those determined by the CCSD(T) level calculation.

Corresponding descriptions have been added in the page 8 line 198-215 and page 10 line 264-271 of the revised manuscript:

*In order to obtain a better evaluation on the reliability of M06-2X functional in computing energies, the single point energies of species included in some selected elementary reactions (R1a-R1d, R3a-R3d and R5a-R5d) are recalculated at the CCSD(T)/6-311+G(2df,2p) level based on the M06-2X/6-311+G(2df,2p) optimized geometries. Furthermore, the basis set superposition error (BSSE) is performed using the counterpoise method described by Boys and Bernardi (1970) to estimate the stability of the pre-reactive complexes. The electronic-energy ($\Delta E_a^{\#}$) and free-energy ($\Delta G_a^{\#}$) barriers comparisons of both approaches for considering BSSE correction and not considering are listed in Table S1. As shown in Table S1, the contributions of BSSE corrections to the barriers are ~ 1.4 (CCSD(T)) and ~ 0.4 (M06-2X) kcal mol$^{-1}$, respectively. The mean absolute deviations (MAD) of both approaches are 0.98 ($\Delta E_a^{\#}$) and 0.96 ($\Delta G_a^{\#}$) kcal mol$^{-1}$ when without considering BSSE correction, while they become 0.38 ($\Delta E_a^{\#}$) and 0.34 ($\Delta G_a^{\#}$) kcal mol$^{-1}$ when considering BSSE correction. The result shows that the M06-2X method in combination with the BSSE correction afford energies similar to those determined by the accurate and well recognized CCSD(T) level calculation. Considering the computational costs, the*

*M06-2X/def2-TZVP method is selected to perform the single-point energy calculation for the title reaction system.*

*As shown in Table 1, the barrier differences between the computational and literature ones that were derived from the CCSD(T)/aug-cc-pVTZ//B3LYP/6-311+G(2df,2p) method (Anglada et al., 2016) are 0.5-1.0 kcal mol$^{-1}$. Such discrepancies may be attributed to the different theoretical method used in computing energies. The barrier of Entry 4 is 3.3 kcal mol$^{-1}$, which is lower than the corresponding CCSD(T)/CBS result by 0.5 kcal mol$^{-1}$ (Anglada et al., 2016). The results show that the M06-2X method provides energies similar to those determined by the CCSD(T) level calculation.*

**Table 1** *Relative free-energies (kcal mol$^{-1}$) for the stationary points and free-energy ($\Delta G_a^{\#}$) barriers for the elementary pathways of distinct carbonyl oxides reactions with water dimer calculated at the M06-2X/def2-TZVP//M06-2X/6-311+G(2df,2p) level of theory. Labels A, B, C, and D are defined in Figure 1*

| Entry | R1 | R2 | DO2H4O4H3 | DO4H2O3H1 | A | B | C | D | $\Delta G_a^{\#}$ |
|---|---|---|---|---|---|---|---|---|---|
| 1 | H | H | -123.6 | 96.7 | -2.5(-2.9) | 0.2 | -40.5 | -39.9 | 2.7 [2.2] |
| 2 | H | H | 124.5 | -94.9 | -2.5(-3.0) | 0.2 | -39.6 | -40.2 | 2.7 |
| 3 | H | H | -143.8 | -116.9 | -2.2(-2.6) | 0.2 | -39.1 | -40.2 | 2.4 |
| 4 | H | H | 143.0 | 122.7 | -1.9(-2.3) | 1.4 | -40.0 | -40.3 | 3.3 {3.8} |
| 5 | CH$_3$ | H | -126.2 | 100.9 | -2.5(-3.0) | 4.1 | -32.4 | -31.9 | 6.6 [6.0] |
| 6 | CH$_3$ | H | 130.0 | -90.4 | -2.0(-2.5) | 4.3 | -31.6 | -32.2 | 6.3 |
| 7 | CH$_3$ | H | -146.0 | -116.3 | -2.1(-2.6) | 4.1 | -30.9 | -32.2 | 6.2 |
| 8 | CH$_3$ | H | 138.1 | 126.3 | -2.3(-2.8) | 4.7 | -31.7 | -31.9 | 7.0 |
| 9 | H | CH$_3$ | -122.1 | 95.0 | -4.3(-4.8) | 0.6 | -36.6 | -36.2 | 4.9 [3.9] |
| 10 | H | CH$_3$ | 125.6 | -93.6 | -3.9(-4.3) | 0.7 | -35.7 | -36.5 | 4.6 |
| 11 | H | CH$_3$ | -138.1 | -120.5 | -4.4(-4.9) | 1.1 | -34.9 | -36.5 | 5.5 |
| 12 | H | CH$_3$ | 139.2 | 123.4 | -3.6(-4.1) | 2.0 | -36.0 | -36.2 | 5.6 |
| 13 | CH$_3$ | CH$_3$ | -125.4 | 101.5 | -4.6(-5.1) | 4.2 | -29.7 | -29.1 | 8.8 [7.8] |
| 14 | CH$_3$ | CH$_3$ | 128.5 | -89.8 | -4.2(-4.8) | 4.5 | -28.6 | -29.7 | 8.7 |
| 15 | CH$_3$ | CH$_3$ | -145.4 | -117.6 | -4.9(-5.3) | 4.6 | -28.2 | -29.7 | 9.5 |
| 16 | CH$_3$ | CH$_3$ | 136.3 | 129.4 | -4.5(-5.0) | 5.2 | -29.2 | -29.1 | 9.7 |

*Values in parenthesis correspond to without considering the BSSE correction, values in brackets correspond to CCSD(T)/aug-cc-pVTZ//B3LYP/6-311+G(2df,2p), values in braces correspond to CCSD(T)/CBS//B3LYP/6-311+G(2df,2p)*

**Table S1** *Y/X (Y = M06-2X, CCSD(T), X = 6-311+G(2df,2p), def2-TZVP) calculated energy*

*barrier($\Delta E_a^{\#}$, $\Delta G_a^{\#}$) for the addition reactions of carbonyl oxides with HHPs based on the M06-2X/6-311+G(2df,2p) optimized geometries (kcal mol$^{-1}$)*

| Reactions | CCSD(T)/6-311+G(2df,2p) | | M06-2X/def2-TZVP | |
|---|---|---|---|---|
| | $\Delta E_a^{\#}$ | $\Delta G_a^{\#}$ | $\Delta E_a^{\#}$ | $\Delta G_a^{\#}$ |
| R1a | 9.4$^a$; 7.9$^b$ | 10.1$^a$; 8.6$^b$ | 8.0$^a$; 7.6$^b$ | 8.8$^a$; 8.4$^b$ |
| R1b | 13.0$^a$; 11.6$^b$ | 13.3$^a$; 11.9$^b$ | 11.9$^a$; 11.4$^b$ | 12.2$^a$; 11.7$^b$ |
| R1c | 8.1$^a$; 6.7$^b$ | 9.2$^a$; 7.8$^b$ | 6.9$^a$; 6.4$^b$ | 7.9$^a$; 7.4$^b$ |
| R1d | 8.6$^a$; 7.6$^b$ | 10.2$^a$; 9.2$^b$ | 7.0$^a$; 6.7$^b$ | 8.7$^a$; 8.4$^b$ |
| R3a | 6.6$^a$; 4.9$^b$ | 7.1$^a$; 5.4$^b$ | 5.8$^a$; 4.4$^b$ | 6.2$^a$; 5.8$^b$ |
| R3b | 8.4$^a$; 7.3$^b$ | 9.7$^a$; 8.6$^b$ | 7.3$^a$; 6.8$^b$ | 8.8$^a$; 8.3$^b$ |
| R3c | 6.5$^a$; 4.8$^b$ | 7.3$^a$; 5.6$^b$ | 5.8$^a$; 5.4$^b$ | 6.6$^a$; 6.2$^b$ |
| R3d | 8.4$^a$; 6.8$^b$ | 9.7$^a$; 8.1$^b$ | 7.3$^a$; 6.8$^b$ | 8.6$^a$; 8.1$^b$ |
| R5a | 12.3$^a$; 10.9$^b$ | 13.2$^a$; 11.8$^b$ | 11.5$^a$; 11.1$^b$ | 12.5$^a$; 12.1$^b$ |
| R5b | 11.0$^a$; 9.4$^b$ | 12.4$^a$; 10.8$^b$ | 10.6$^a$; 10.1$^b$ | 12.0$^a$; 11.5$^b$ |
| R5c | 12.0$^a$; 10.6$^b$ | 13.3$^a$; 11.9$^b$ | 11.3$^a$; 10.9$^b$ | 12.5$^a$; 12.1$^b$ |
| R5d | 12.2$^a$; 11.0$^b$ | 13.8$^a$; 12.6$^b$ | 11.4$^a$; 11.0$^b$ | 13.0$^a$; 12.6$^b$ |

*a and b represent without and with considering the BSSE correction*

5. Regarding the kinetics, the authors should clarify if they have considered the pre-reactive complexes in the kinetic study and if they play a role in the temperature dependence of the rate constants.

**Response:** As shown in Figure 5, the $CH_2OO$ + $HO-CH_2OO-H$ ($Pa_1$) reaction proceeds according to a two-step mechanism: (i) a fast thermal equilibrium between the reactants and intermediate IM1a, (ii) the addition of $CH_2OO$ leading to the formation of product P1a. The whole reaction process is expressed as follows:

$$CH_2OO + HO-CH_2OO-H \underset{k_{-1}}{\overset{k_1}{\rightleftharpoons}} IM1a \xrightarrow{k_2} TS1a \rightarrow P1a \qquad (5)$$

Applying the steady state approximation (SSA) for the intermediate IM1a, the overall rate coefficient is extrapolated to the eqn (6): (Zhang et al., 2012; Liu et al., 2015)

$$k_{ovr} = \frac{k_1 \times k_2}{k_{-1} + k_2} \qquad (6)$$

If $k_2 \ll k_{-1}$, the overall rate coefficient is written as follows: (Ryzhkov et al., 2006; Chen et al., 2016)

$$k_{ovr} = \frac{k_1 \times k_2}{k_{-1} + k_2} \approx \frac{k_1}{k_{-1}} k_2 = K_{eq} k_2 \qquad (7)$$

The equilibrium coefficient $K_{eq}$ is expressed as eqn (8):

$$K_{eq} = \sigma \frac{Q_{IM}(T)}{Q_A(T)Q_B(T)} \exp\left(\frac{G_R - G_{IM}}{RT}\right) \qquad (8)$$

where $\sigma$ is reaction symmetry number, $Q_{IM}(T)$, $Q_A(T)$ and $Q_B(T)$ denote the products of electronic, translational, rotational, torsional, and vibrational canonical partition functions for the intermediate, reactants A and B, respectively (Mendes, et al., 2014), T is the temperature in Kelvin, $G_R$ and $G_{IM}$ are the total free-energies of the reactant and intermediate, respectively.

Table R1 lists the partition functions, equilibrium coefficients ($K_{eq}$), and rate coefficients ($k_{2(IM1a\text{-}TS1a)}$ and $k_{ovr(R1a)}$) of the bimolecular reaction of $CH_2OO$ with $HO\text{-}CH_2OO\text{-}H$ (Pa$_1$). As shown the Table R1, the partition functions of reactants and intermediate increase with raising temperature, whereas the $k_{ovr(R1a)}$, $K_{eq}$ and $k_{2(IM1a\text{-}TS1a)}$ decrease with increasing temperature, indicating that there is not a direct correlation between $k_{ovr(R1a)}$ and partition function of pre-reactive intermediate. The result shows that the kinetics parameters strongly depend on the interaction between $K_{eq}$ and $k_{2(IM1a\text{-}TS1a)}$. Similar conclusions are also obtained from those of the *anti*-$CH_3CHOO$ + $HO\text{-}CH_2OO\text{-}H$(R9), *syn*-$CH_3CHOO$ + $HO\text{-}CH_2OO\text{-}H$(R10) and $(CH_3)_2CHOO$ + $HO\text{-}CH_2OO\text{-}H$(R11) systems (Table R2-R4).

**Table R1** The partition function, equilibrium coefficient ($K_{eq}$) (cm$^3$ molecule$^{-1}$), and rate coefficients ($k_{2(IM1a\text{-}TS1a)}$ and $k_{ovr(R1a)}$) (cm$^3$ molecule$^{-1}$ s$^{-1}$) of $CH_2OO$ reactions with $HO\text{-}CH_2OO\text{-}H$ (Pa$_1$) computed at different temperatures

| T/K | $Q_{CH2OO}$ | $Q_{HOCH2OOH}$ | $Q_{IMa}$ | $K_{eq}$ | $k_{2(IM1a\text{-}TS1a)}$ | $k_{ovr(R1a)}$ |
|---|---|---|---|---|---|---|
| 273 | $7.9 \times 10^{10}$ | $1.6 \times 10^{12}$ | $1.1 \times 10^{15}$ | $2.8 \times 10^{-6}$ | $4.3 \times 10^{-6}$ | $1.2 \times 10^{-11}$ |
| 280 | $8.8 \times 10^{10}$ | $1.9 \times 10^{12}$ | $1.5 \times 10^{15}$ | $2.4 \times 10^{-6}$ | $4.0 \times 10^{-6}$ | $9.4 \times 10^{-12}$ |
| 298 | $1.2 \times 10^{11}$ | $2.7 \times 10^{12}$ | $2.7 \times 10^{15}$ | $1.6 \times 10^{-6}$ | $3.3 \times 10^{-6}$ | $5.4 \times 10^{-12}$ |
| 300 | $1.2 \times 10^{11}$ | $2.8 \times 10^{12}$ | $2.9 \times 10^{15}$ | $1.6 \times 10^{-6}$ | $3.2 \times 10^{-6}$ | $5.1 \times 10^{-12}$ |
| 320 | $1.6 \times 10^{11}$ | $4.2 \times 10^{12}$ | $5.6 \times 10^{15}$ | $1.1 \times 10^{-6}$ | $2.7 \times 10^{-6}$ | $3.0 \times 10^{-12}$ |
| 340 | $2.1 \times 10^{11}$ | $6.2 \times 10^{12}$ | $1.1 \times 10^{16}$ | $8.1 \times 10^{-7}$ | $2.3 \times 10^{-6}$ | $1.9 \times 10^{-12}$ |
| 360 | $2.7 \times 10^{11}$ | $9.0 \times 10^{12}$ | $2.0 \times 10^{16}$ | $6.2 \times 10^{-7}$ | $2.0 \times 10^{-6}$ | $1.2 \times 10^{-12}$ |
| 380 | $3.5 \times 10^{11}$ | $1.3 \times 10^{13}$ | $3.6 \times 10^{16}$ | $4.8 \times 10^{-7}$ | $1.8 \times 10^{-6}$ | $8.6 \times 10^{-13}$ |
| 400 | $4.5 \times 10^{11}$ | $1.8 \times 10^{13}$ | $6.6 \times 10^{16}$ | $3.9 \times 10^{-7}$ | $1.6 \times 10^{-6}$ | $6.3 \times 10^{-13}$ |

**Table R2** The partition function, equilibrium coefficient ($K_{eq}$) (cm$^3$ molecule$^{-1}$), and rate coefficients ($k_{2(IM9\text{-}TS9)}$ and $k_{ovr(R9)}$) (cm$^3$ molecule$^{-1}$ s$^{-1}$) of *anti*-$CH_3CHOO$ reactions with $HO\text{-}CH_2OO\text{-}H$ (Pa$_1$) computed at different temperatures

| T/K | $Q_{anti\text{-}CH3CHOO}$ | $Q_{HOCH2OOH}$ | $Q_{IM9}$ | $K_{eq}$ | $k_{2(IM9\text{-}TS9)}$ | $k_{ovr(R9)}$ |
|---|---|---|---|---|---|---|
| 273 | $1.4 \times 10^{12}$ | $1.6 \times 10^{12}$ | $9.1 \times 10^{15}$ | $1.3 \times 10^{-4}$ | $4.4 \times 10^{-5}$ | $5.5 \times 10^{-9}$ |
| 280 | $1.6 \times 10^{12}$ | $1.9 \times 10^{12}$ | $1.2 \times 10^{16}$ | $9.6 \times 10^{-5}$ | $3.9 \times 10^{-5}$ | $3.7 \times 10^{-9}$ |
| 298 | $2.4 \times 10^{12}$ | $2.7 \times 10^{12}$ | $2.5 \times 10^{16}$ | $5.0 \times 10^{-5}$ | $2.9 \times 10^{-5}$ | $1.5 \times 10^{-9}$ |

| 300 | $2.5 \times 10^{12}$ | $2.8 \times 10^{12}$ | $2.7 \times 10^{16}$ | $4.7 \times 10^{-5}$ | $2.9 \times 10^{-5}$ | $1.3 \times 10^{-9}$ |
|---|---|---|---|---|---|---|
| 320 | $3.7 \times 10^{12}$ | $4.2 \times 10^{12}$ | $5.8 \times 10^{16}$ | $2.5 \times 10^{-5}$ | $2.2 \times 10^{-5}$ | $5.5 \times 10^{-10}$ |
| 340 | $5.4 \times 10^{12}$ | $6.2 \times 10^{12}$ | $1.2 \times 10^{17}$ | $1.5 \times 10^{-5}$ | $1.7 \times 10^{-5}$ | $2.6 \times 10^{-10}$ |
| 360 | $7.8 \times 10^{12}$ | $9.0 \times 10^{12}$ | $2.5 \times 10^{17}$ | $9.1 \times 10^{-6}$ | $1.4 \times 10^{-5}$ | $1.3 \times 10^{-10}$ |
| 380 | $1.1 \times 10^{13}$ | $1.3 \times 10^{13}$ | $5.1 \times 10^{17}$ | $5.9 \times 10^{-6}$ | $1.2 \times 10^{-5}$ | $7.1 \times 10^{-11}$ |
| 400 | $1.6 \times 10^{13}$ | $1.8 \times 10^{13}$ | $1.0 \times 10^{18}$ | $4.1 \times 10^{-6}$ | $1.0 \times 10^{-5}$ | $4.1 \times 10^{-11}$ |

**Table R3** The partition function, equilibrium coefficient ($K_{eq}$) (cm$^3$ molecule$^{-1}$), and rate coefficients ($k_{2(IM10\text{-}TS10)}$ and $k_{ovr(R10)}$) (cm$^3$ molecule$^{-1}$ s$^{-1}$) of *syn*-CH$_3$CHOO reactions with HO-CH$_2$OO-H (Pa$_1$) computed at different temperatures

| T/K | $Q_{syn\text{-}CH3CHOO}$ | $Q_{HOCH2OOH}$ | $Q_{IM10}$ | $K_{eq}$ | $k_{2(IM10\text{-}TS10)}$ | $k_{ovr(R10)}$ |
|---|---|---|---|---|---|---|
| 273 | $1.0 \times 10^{12}$ | $1.6 \times 10^{12}$ | $6.6 \times 10^{15}$ | $1.5 \times 10^{-6}$ | $5.6 \times 10^{-9}$ | $8.1 \times 10^{-15}$ |
| 280 | $1.2 \times 10^{12}$ | $1.9 \times 10^{12}$ | $8.8 \times 10^{15}$ | $1.2 \times 10^{-6}$ | $6.0 \times 10^{-9}$ | $7.5 \times 10^{-15}$ |
| 298 | $1.7 \times 10^{12}$ | $2.7 \times 10^{12}$ | $1.8 \times 10^{16}$ | $8.6 \times 10^{-7}$ | $7.3 \times 10^{-9}$ | $6.3 \times 10^{-15}$ |
| 300 | $1.8 \times 10^{12}$ | $2.8 \times 10^{12}$ | $1.9 \times 10^{16}$ | $8.3 \times 10^{-7}$ | $7.5 \times 10^{-9}$ | $6.2 \times 10^{-15}$ |
| 320 | $2.6 \times 10^{12}$ | $4.2 \times 10^{12}$ | $4.1 \times 10^{16}$ | $5.8 \times 10^{-7}$ | $9.1 \times 10^{-9}$ | $5.3 \times 10^{-15}$ |
| 340 | $3.7 \times 10^{12}$ | $6.2 \times 10^{12}$ | $8.7 \times 10^{16}$ | $4.3 \times 10^{-7}$ | $1.1 \times 10^{-8}$ | $4.6 \times 10^{-15}$ |
| 360 | $5.3 \times 10^{12}$ | $9.0 \times 10^{12}$ | $1.8 \times 10^{17}$ | $3.3 \times 10^{-7}$ | $1.3 \times 10^{-8}$ | $4.1 \times 10^{-15}$ |
| 380 | $7.5 \times 10^{12}$ | $1.3 \times 10^{13}$ | $3.6 \times 10^{17}$ | $2.6 \times 10^{-7}$ | $1.5 \times 10^{-8}$ | $3.8 \times 10^{-15}$ |
| 400 | $1.1 \times 10^{13}$ | $1.8 \times 10^{13}$ | $7.2 \times 10^{17}$ | $2.1 \times 10^{-7}$ | $1.7 \times 10^{-8}$ | $3.5 \times 10^{-15}$ |

**Table R4** The partition function, equilibrium coefficient ($K_{eq}$) (cm$^3$ molecule$^{-1}$), and rate coefficients ($k_{2(IM11\text{-}TS11)}$ and $k_{ovr(R11)}$) (cm$^3$ molecule$^{-1}$ s$^{-1}$) of (CH$_3$)$_2$CHOO reactions with HO-CH$_2$OO-H (Pa$_1$) computed at different temperatures

| T/K | $Q_{(CH3)2CHOO}$ | $Q_{HOCH2OOH}$ | $Q_{IM11}$ | $K_{eq}$ | $k_{2(IM10\text{-}TS10)}$ | $k_{ovr(R10)}$ |
|---|---|---|---|---|---|---|
| 273 | $7.2 \times 10^{12}$ | $1.6 \times 10^{12}$ | $4.0 \times 10^{16}$ | $8.8 \times 10^{-5}$ | $3.1 \times 10^{-8}$ | $2.7 \times 10^{-12}$ |
| 280 | $8.5 \times 10^{12}$ | $1.9 \times 10^{12}$ | $5.5 \times 10^{16}$ | $6.9 \times 10^{-5}$ | $3.4 \times 10^{-8}$ | $2.3 \times 10^{-12}$ |
| 298 | $1.3 \times 10^{13}$ | $2.7 \times 10^{12}$ | $1.2 \times 10^{17}$ | $3.7 \times 10^{-5}$ | $4.1 \times 10^{-8}$ | $1.5 \times 10^{-12}$ |
| 300 | $1.4 \times 10^{13}$ | $2.8 \times 10^{12}$ | $1.3 \times 10^{17}$ | $3.5 \times 10^{-5}$ | $4.2 \times 10^{-8}$ | $1.5 \times 10^{-12}$ |
| 320 | $2.2 \times 10^{13}$ | $4.2 \times 10^{12}$ | $3.2 \times 10^{17}$ | $1.9 \times 10^{-5}$ | $5.2 \times 10^{-8}$ | $1.0 \times 10^{-12}$ |
| 340 | $3.5 \times 10^{13}$ | $6.2 \times 10^{12}$ | $7.3 \times 10^{17}$ | $1.2 \times 10^{-5}$ | $6.2 \times 10^{-8}$ | $7.2 \times 10^{-13}$ |
| 360 | $5.5 \times 10^{13}$ | $9.0 \times 10^{12}$ | $1.7 \times 10^{18}$ | $7.4 \times 10^{-6}$ | $7.4 \times 10^{-8}$ | $5.4 \times 10^{-13}$ |
| 380 | $8.6 \times 10^{13}$ | $1.3 \times 10^{13}$ | $3.8 \times 10^{18}$ | $5.0 \times 10^{-6}$ | $8.6 \times 10^{-8}$ | $4.2 \times 10^{-13}$ |
| 400 | $1.3 \times 10^{14}$ | $1.8 \times 10^{13}$ | $8.3 \times 10^{18}$ | $3.5 \times 10^{-6}$ | $9.8 \times 10^{-8}$ | $3.4 \times 10^{-13}$ |

The corresponding sentences have been added in the page 9 line 219-237 of the revised manuscript in blue color:

*As shown in Figure 5, the CH$_2$OO + HO-CH$_2$OO-H (Pa$_1$) reaction proceeds according to a two-step mechanism: (i) a fast thermal equilibrium between the reactants and intermediate IM1a, (ii) the addition of CH$_2$OO leading to the formation of product P1a. The whole reaction process is expressed as follows:*

$$CH_2OO + HO\text{-}CH_2OO\text{-}H \underset{k_{-1}}{\overset{k_1}{\rightleftharpoons}} IM1a \xrightarrow{k_2} TS1a \rightarrow P1a \qquad (4)$$

*Applying the steady state approximation (SSA) for the intermediate IM1a, the overall rate coefficient is extrapolated to the eqn (5) (Zhang et al., 2012; Liu et al., 2015)*

$$k_{ovr} = \frac{k_1 \times k_2}{k_{-1} + k_2} \qquad (5)$$

*If $k_2 << k_{-1}$, the overall rate coefficient is written as follows: (Chen et al., 2016b; Ryzhkov et al., 2006)*

$$k_{ovr} = \frac{k_1 \times k_2}{k_{-1} + k_2} \approx \frac{k_1}{k_{-1}} k_2 = K_{eq} k_2 \qquad (6)$$

*The equilibrium coefficient $K_{eq}$ is expressed as eqn (7):*

$$K_{eq} = \sigma \frac{Q_{IM}(T)}{Q_A(T)Q_B(T)} \exp\left(\frac{G_R - G_{IM}}{RT}\right) \qquad (7)$$

*where $\sigma$ is reaction symmetry number, $Q_{IM}(T)$, $Q_A(T)$ and $Q_B(T)$ denote the products of electronic, translational, rotational, torsional, and vibrational canonical partition functions for the intermediate, reactants A and B, respectively (Mendes et al., 2014), T is the temperature in Kelvin, $G_R$ and $G_{IM}$ are the total free-energies of the reactant and complex, respectively.*

6. The authors report rate constants for the reactions of the carbonyl oxides considered with HHP's (Table 2), but no mention is done for the reactions of P1x with Carbonyl oxides. Moreover, that authors should clarify if they have considered all different conformers of the stationary points in the kinetic study. In addition, they should estimate the errors in the these calculated rate constants, since they can be between one and two orders of magnitude according to the errors in the computed free energy barriers.

**Response:** Based on the Reviewer's suggestion, the rate coefficients of carbonyl oxides reactions with P1x, P3x, P5x, and P7x are calculated using a combination of canonical transition state theory (CTST) and an asymmetric Eckart tunneling correction at 273-400 K. And the different conformers of stationary points in the kinetics study are taken into account, with the obtained results listed in Table S5-S8. As shown in Table S5, the predicted rate coefficients for the reaction of $CH_2OO$ with P1a decrease with increasing temperature, with a similar trend observed for $CH_2OO$ + P1b, $CH_2OO$ + P1c and $CH_2OO$ + P1d systems. The result implies that the oligomer

formation from $CH_2OO$ reaction with HHP is preferable under low temperature conditions. Similar conclusions are obtained from the *anti*-$CH_3CHOO$ + P3x (Table S6), *syn*-$CH_3CHOO$ + P5x (Table S7) and $(CH_3)_2CHOO$ + P7x (Table S8) systems. In order to avoid redundancy, we do not repeat them here in detail. Considering the errors of the computed free energy barriers, the uncertainty of rate coefficient is estimated within an order of magnitude.

Corresponding descriptions have been added in the page 18 line 500-506 and page 19 line 521-528 of the revised manuscript:

*the rate coefficients of distinct SCI reactions with $HO\text{-}CH_2OO\text{-}H$ ($Pa_1$) are computed using a combination of canonical transition state theory (CTST) and an asymmetric Eckart tunneling correction based on the free energies obtained at the M06-2X level, in the temperature range from 273 to 400 K. And the different conformers of stationary points in the kinetics study are taken into account, with the results listed in Table 2 and Table S5-S8.*

*As shown in Table S5, the rate coefficients of $CH_2OO$ + P1a, $CH_2OO$ + P1b, $CH_2OO$ + P1c and $CH_2OO$ + P1d reactions decrease with increasing temperature, indicating that the oligomer formation from $CH_2OO$ reactions with HHP is preferable under low temperature conditions. Similar conclusions are also obtained from the anti-$CH_3CHOO$ + P3x (Table S6), syn-$CH_3CHOO$ + P5x (Table S7) and $(CH_3)_2CHOO$ + P7x (Table S8) systems. In order to avoid redundancy, we do not repeat them here in detail. Considering the errors of the computed free energy barriers, the uncertainty of rate coefficient is estimated within an order of magnitude.*

***Table S5*** *Rate coefficients ($cm^3$ molecule$^{-1}$ s$^{-1}$) of $CH_2OO$ reactions with P1a, P1b, P1c and P1d computed at different temperatures*

| T/K | $k_{(CH2OO+P1a)}$ | $k_{(CH2OO+P1b)}$ | $k_{(CH2OO+P1c)}$ | $k_{(CH2OO+P1d)}$ |
|---|---|---|---|---|
| 273 | $5.1 \times 10^{-12}$ | $3.5 \times 10^{-12}$ | $8.2 \times 10^{-13}$ | $4.8 \times 10^{-11}$ |
| 280 | $4.0 \times 10^{-12}$ | $2.9 \times 10^{-12}$ | $7.1 \times 10^{-13}$ | $3.7 \times 10^{-11}$ |
| 298 | $2.3 \times 10^{-12}$ | $1.9 \times 10^{-12}$ | $5.2 \times 10^{-13}$ | $1.9 \times 10^{-11}$ |
| 300 | $2.2 \times 10^{-12}$ | $1.8 \times 10^{-12}$ | $5.0 \times 10^{-13}$ | $1.8 \times 10^{-11}$ |
| 320 | $1.3 \times 10^{-12}$ | $1.2 \times 10^{-12}$ | $3.7 \times 10^{-13}$ | $1.0 \times 10^{-11}$ |
| 340 | $8.3 \times 10^{-13}$ | $8.5 \times 10^{-13}$ | $2.8 \times 10^{-13}$ | $5.9 \times 10^{-12}$ |
| 360 | $5.6 \times 10^{-13}$ | $6.3 \times 10^{-13}$ | $2.3 \times 10^{-13}$ | $3.8 \times 10^{-12}$ |
| 380 | $4.0 \times 10^{-13}$ | $4.9 \times 10^{-13}$ | $1.9 \times 10^{-13}$ | $2.5 \times 10^{-12}$ |
| 400 | $3.0 \times 10^{-13}$ | $3.9 \times 10^{-13}$ | $1.6 \times 10^{-13}$ | $1.8 \times 10^{-12}$ |

**Table S6** *Rate coefficients (cm$^3$ molecule$^{-1}$ s$^{-1}$) of anti-CH$_3$CHOO reactions with P3a, P3b, P3c and P3d computed at different temperatures*

| T/K | $k_{(anti+P3a)}$ | $k_{(anti+P3b)}$ | $k_{(anti+P3c)}$ | $k_{(anti+P3d)}$ |
|---|---|---|---|---|
| 273 | $3.5 \times 10^{-9}$ | $2.3 \times 10^{-9}$ | $1.7 \times 10^{-7}$ | $2.1 \times 10^{-9}$ |
| 280 | $2.3 \times 10^{-9}$ | $1.4 \times 10^{-9}$ | $1.0 \times 10^{-7}$ | $1.4 \times 10^{-9}$ |
| 298 | $8.8 \times 10^{-10}$ | $5.8 \times 10^{-10}$ | $2.9 \times 10^{-8}$ | $5.9 \times 10^{-10}$ |
| 300 | $8.0 \times 10^{-10}$ | $5.3 \times 10^{-10}$ | $2.6 \times 10^{-8}$ | $5.4 \times 10^{-10}$ |
| 320 | $3.2 \times 10^{-10}$ | $2.3 \times 10^{-10}$ | $7.9 \times 10^{-9}$ | $2.3 \times 10^{-10}$ |
| 340 | $1.4 \times 10^{-10}$ | $1.1 \times 10^{-10}$ | $2.8 \times 10^{-9}$ | $1.1 \times 10^{-10}$ |
| 360 | $7.1 \times 10^{-11}$ | $5.8 \times 10^{-11}$ | $1.1 \times 10^{-9}$ | $5.9 \times 10^{-11}$ |
| 380 | $3.8 \times 10^{-11}$ | $3.3 \times 10^{-11}$ | $4.9 \times 10^{-10}$ | $3.4 \times 10^{-11}$ |
| 400 | $2.2 \times 10^{-11}$ | $2.0 \times 10^{-11}$ | $2.4 \times 10^{-10}$ | $2.0 \times 10^{-11}$ |

**Table S7** *Rate coefficients (cm$^3$ molecule$^{-1}$ s$^{-1}$) of syn-CH$_3$CHOO reactions with P5a, P5b, P5c and P5d computed at different temperatures*

| T/K | $k_{(syn+P5a)}$ | $k_{(syn+P5b)}$ | $k_{(syn+P5c)}$ | $k_{(syn+P5d)}$ |
|---|---|---|---|---|
| 273 | $2.1 \times 10^{-11}$ | $1.7 \times 10^{-13}$ | $2.1 \times 10^{-11}$ | $2.4 \times 10^{-13}$ |
| 280 | $1.5 \times 10^{-11}$ | $1.4 \times 10^{-13}$ | $1.5 \times 10^{-11}$ | $2.0 \times 10^{-13}$ |
| 298 | $7.5 \times 10^{-12}$ | $1.0 \times 10^{-13}$ | $7.6 \times 10^{-12}$ | $1.4 \times 10^{-13}$ |
| 300 | $6.9 \times 10^{-12}$ | $9.7 \times 10^{-14}$ | $7.0 \times 10^{-12}$ | $1.4 \times 10^{-13}$ |
| 320 | $3.5 \times 10^{-12}$ | $6.9 \times 10^{-14}$ | $3.6 \times 10^{-12}$ | $9.6 \times 10^{-14}$ |
| 340 | $1.9 \times 10^{-12}$ | $5.2 \times 10^{-14}$ | $2.0 \times 10^{-12}$ | $7.1 \times 10^{-14}$ |
| 360 | $1.1 \times 10^{-12}$ | $4.0 \times 10^{-14}$ | $1.2 \times 10^{-12}$ | $5.5 \times 10^{-14}$ |
| 380 | $7.1 \times 10^{-13}$ | $3.0 \times 10^{-14}$ | $7.3 \times 10^{-13}$ | $4.4 \times 10^{-14}$ |
| 400 | $4.7 \times 10^{-13}$ | $2.6 \times 10^{-14}$ | $4.9 \times 10^{-13}$ | $3.6 \times 10^{-14}$ |

**Table S8** *Rate coefficients (cm$^3$ molecule$^{-1}$ s$^{-1}$) of (CH$_3$)$_2$CHOO reactions with P7a, P7b, P7c and P7d computed at different temperatures*

| T/K | $k_{((CH3)2CHOO +P7a)}$ | $k_{((CH3)2CHOO +P7b)}$ | $k_{((CH3)2CHOO +P7c)}$ | $k_{((CH3)2CHOO +P7d)}$ |
|---|---|---|---|---|
| 273 | $7.8 \times 10^{-14}$ | $9.1 \times 10^{-14}$ | $1.8 \times 10^{-12}$ | $4.5 \times 10^{-13}$ |
| 280 | $6.8 \times 10^{-14}$ | $8.2 \times 10^{-14}$ | $1.5 \times 10^{-12}$ | $3.9 \times 10^{-13}$ |
| 298 | $5.0 \times 10^{-14}$ | $6.6 \times 10^{-14}$ | $1.0 \times 10^{-12}$ | $2.8 \times 10^{-13}$ |
| 300 | $4.8 \times 10^{-14}$ | $6.5 \times 10^{-14}$ | $9.9 \times 10^{-13}$ | $2.7 \times 10^{-13}$ |
| 320 | $3.6 \times 10^{-14}$ | $5.3 \times 10^{-14}$ | $6.9 \times 10^{-13}$ | $1.9 \times 10^{-13}$ |
| 340 | $2.8 \times 10^{-14}$ | $4.5 \times 10^{-14}$ | $5.1 \times 10^{-13}$ | $1.5 \times 10^{-13}$ |
| 360 | $2.2 \times 10^{-14}$ | $3.9 \times 10^{-14}$ | $3.9 \times 10^{-13}$ | $1.2 \times 10^{-13}$ |

| 380 | $1.9 \times 10^{-14}$ | $3.5 \times 10^{-14}$ | $3.1 \times 10^{-13}$ | $9.6 \times 10^{-14}$ |
| 400 | $1.6 \times 10^{-14}$ | $3.1 \times 10^{-14}$ | $2.5 \times 10^{-13}$ | $8.0 \times 10^{-14}$ |

7. With respect to the atmospheric implications, the authors compare the reaction rates of the reaction investigated with those between carbonyl oxides with formic acid. In my opinion, the reactions rates of carbonyl oxides with water and water dimer, but also the reactions rates of HHPs with water should be also taken into account, because the high concentration of water vapor in the atmosphere. For the last, there are free energy barriers in the literature to compare with.

**Response:** As shown in Table 2, the rate coefficient of *anti*-$CH_3CHOO$ + $Pa_1$ reaction (R9) is significantly higher than that of the other three pathways (R1a, R10 and R11). Therefore, it would be interesting to investigate whether the *anti*-$CH_3CHOO$ + $Pa_1$ reaction can compete well with the *anti*-$CH_3CHOO$ + $(H_2O)_2$ (R12) system because the latter reaction is the dominant chemical sink (Anglada et al., 2016; Taatjes, et al., 2013). The ratio of reaction rates of R9 and R12 is expressed as follows

$$\frac{\upsilon_{R9}}{\upsilon_{R12}} = \frac{k_{R9}[CI][Pa_1]}{k_{R12}[CI][(H_2O)_2]} = \frac{k_{R9}[Pa_1]}{k_{R12}[(H_2O)_2]} \qquad (9)$$

The room temperature rate coefficient $k_{R9}$ is $1.5 \times 10^{-9}$ cm$^3$ molecule$^{-1}$ s$^{-1}$. Assuming that the concentration of $Pa_1$ is approximately equal to that of SCIs (~ $5.0 \times 10^4$ molecules cm$^{-3}$, within an order of magnitude uncertainty) in the boreal forest and rural environments of Finland and Germany (Novelli et al, 2016, 2017). The atmospheric lifetime of *anti*-$CH_3CHOO$ reactivity toward $Pa_1$ can be estimated as $1.3$-$13 \times 10^3$ s. The experimental rate coefficient of reaction R12 approximately equals ~ $1.0 \times 10^{-11}$ cm$^3$ molecule$^{-1}$ s$^{-1}$ at 298 K (Lin et al., 2016). The concentration of water dimer is $5.5 \times 10^{13}$ molecules cm$^{-3}$ at 3 km altitude (Long et al., 2016). The $v_{R9}/v_{R12}$ ratio is less than 1.4%, meaning that the *anti*-$CH_3CHOO$ + $Pa_1$ reaction is minor loss process in the atmosphere. However, the $[(H_2O)_2]$ is very low at the altitude above 15 km ($< 2.7 \times 10^6$ molecules cm$^{-3}$) (Long et al., 2016), the *anti*-$CH_3CHOO$ + $Pa_1$ reaction can compete well with the *anti*-$CH_3CHOO$ + $(H_2O)_2$ reaction, and thus contribute to the formation and growth of SOA. `

Kumar et al. (2014) proposed that the gas-phase decomposition of $Pa_1$ has two competitive pathways, namely (i) HO-$CH_2OO$-H $\rightarrow$ $CH_2O$ + $H_2O_2$ and (ii) HO-$CH_2OO$-H $\rightarrow$ HCOOH + $H_2O$. The free energy barriers $\Delta G_a^{\#}$ in the presence of a single water molecule are 31.2 and 47.8 kcal mol$^{-1}$, respectively, which are 13.5 and 10.2 kcal mol$^{-1}$ lower than the uncatalyzed reactions.

The result reveals that the formaldehyde-forming channel is preferable in the absence and presence of water molecule, and the role of water catalysis on the gas-phase $Pa_1$ decomposition is significant. The $\Delta G_a^{\#}$ of bimolecular reaction of *anti*-$CH_3CHOO$ with $Pa_1$ is 7.3 kcal mol$^{-1}$, which is 23.9 kcal mol$^{-1}$ lower than the formaldehyde-forming channel. It is concluded that the $Pa_1$ + $H_2O$ reaction is less competitive as compared to the *anti*-$CH_3CHOO$ + $Pa_1$ system.

Corresponding descriptions have been added in the page 20 line 542-556, page 20 line 565 and page 21 line 566-579 of the revised manuscript:

*As discussed above, the anti-CH$_3$CHOO + HO-CH$_2$OO-H (Pa$_1$) reaction (R9) is preferred over the other three pathways (R1a, R10 and R11). Therefore, it would be interesting to investigate whether the anti-CH$_3$CHOO + Pa$_1$ reaction can compete well with the anti-CH$_3$CHOO + (H$_2$O)$_2$ (R12) system because the latter reaction is the dominant chemical sink (Taatjes et al., 2013; Anglada et al., 2016). The ratio of reaction rates of R9 and R12 is expressed as follows*

$$\frac{\upsilon_{R9}}{\upsilon_{R12}} = \frac{k_{R9}[CI][Pa_1]}{k_{R12}[CI][(H_2O)_2]} = \frac{k_{R9}[Pa_1]}{k_{R12}[(H_2O)_2]} \qquad (10)$$

*The room temperature rate coefficient k$_{R9}$ is 1.5 × 10$^{-9}$ cm$^3$ molecule$^{-1}$ s$^{-1}$. Assuming that the concentration of Pa$_1$ is approximately equal to that of SCIs (~5.0 × 10$^4$ molecules cm$^{-3}$, within an order of magnitude uncertainty) in the boreal forest and rural environments of Finland and Germany (Novelli et al., 2016; 2017). The atmospheric lifetime of anti-CH$_3$CHOO reactivity toward Pa$_1$ can be estimated as 1.3-13 × 10$^3$ s. The experimental rate coefficient of reaction R12 approximately equals ~ 1.0 × 10$^{-11}$ cm$^3$ molecule$^{-1}$ s$^{-1}$ at 298 K (Lin et al., 2016). The concentration of water dimer is 5.5 × 10$^{13}$ molecules cm$^{-3}$ at 3 km altitude (Long et al., 2016).*

*The v$_{R9}$/v$_{R12}$ ratio is less than 1.4% when the [(H$_2$O)$_2$] is ~ 10$^{13}$ molecules cm$^{-3}$, meaning that the anti-CH$_3$CHOO + Pa$_1$ reaction is minor loss process in the atmosphere. However, the [(H$_2$O)$_2$] is very low at the altitude above 15 km, the anti-CH$_3$CHOO + Pa$_1$ reaction can compete well with the anti-CH$_3$CHOO + (H$_2$O)$_2$ reaction, and thus contribute to the formation and growth of SOA. Kumar et al. (2014) proposed that the gas-phase decomposition of Pa$_1$ has two competitive pathways: (i) HO-CH$_2$OO-H → CH$_2$O + H$_2$O$_2$ and (ii) HO-CH$_2$OO-H → HCOOH + H$_2$O. The ΔG$_a^{\#}$ in the presence of a single water molecular are 31.2 and 47.8 kcal mol$^{-1}$, respectively, which are 13.5 and 10.2 kcal mol$^{-1}$ lower than the uncatalyzed reactions. The result reveals that the formaldehyde-forming channel is preferable in the absence and presence of water molecule, and*

*the role of water catalysis on the gas-phase decomposition of Pa₁ is significant. The $\Delta G_a^{\#}$ of bimolecular reaction of anti-CH₃CHOO with Pa₁ is 7.3 kcal mol⁻¹, which is 23.9 kcal mol⁻¹ lower than the formaldehyde-forming channel. It is concluded that the Pa₁ + H₂O reaction is less competitive as compared to the anti-CH₃CHOO + Pa₁ system.*

8. An hydrogen misses in the structure of P1a in Figure 2. In addition some addition structures of the P2x compounds should be drawn if the different figures and the numbers should have a larger size.

**Response:** Based on the Reviewer's suggestion, the PES of CH₂OO reaction with HO-CH₂OO-H (Pa₁) is redrawn in Figure 2, and the str in the Figure 2.

[Figure]

***Figure 2.*** *PES (ΔG and ΔE (italic)) for the reaction of CH₂OO with HO-CH₂OO-H (Pa₁) computed at the M06-2X/def2-TZVP//M06-2X/6-311+G(2df,2p) level of theory*

References

Anglada, J. M., and Solé, A.: Impact of water dimer on the atmospheric reactivity of carbonyl oxides, Phys. Chem. Chem. Phys., 18, 17698-17712, 10.1039/c6cp02531e, 2016.

Boys, S. F., and Bernardi, F.: The calculation of small molecular interactions by the differences of separate total energies. Some procedures with reduced errors, Mol. Phys., 19, 553-566, 10.1080/00268977000101561, 1970.

Chen, L., Wang. W. L., Zhou, L. T., Wang, W. N., Liu, F. Y., Li, C. Y., and Lü, J.: Role of water clusters in the reaction of the simplest Criegee intermediate $CH_2OO$ with water vapour, Theor. Chem. Acc., 135, 252-263, 10.1007/s00214-016-1998-2, 2016.

Kumar, M., Busch, D. H., Subramaniam, B., and Thompson, W. H.: Role of tunable acid catalysis in decomposition of α-hydroxyalkyl hydroperoxides and mechanistic implications for tropospheric chemistry, J. Phys. Chem. A, 118, 9701-9711, 10.1021/jp505100x, 2014.

Lin, L. C., Chang, H. T., Chang, C. H., Chao, W., Smith, M. C., Chang, C. H., Lin, J. J. M., and Takahashi, K.: Competition between $H_2O$ and $(H_2O)_2$ reactions with $CH_2OO/CH_3CHOO$, Phys. Chem. Chem. Phys., 18, 4557-4568, 10.1039/C5CP06446E, 2016.

Liu, J., Fang, S., Wang, Z., Yi, W., Tao, F. M., and Liu, J. Y.: Hydrolysis of sulfur dioxide in small clusters of sulfuric acid: mechanistic and kinetic study, Environ. Sci. Technol., 49, 13112-13120, 10.1021/acs.est.5b02977, 2015

Long, B., Bao, J. L., and Truhlar, D. G.: Atmospheric chemistry of Criegee intermediates: unimolecular reactions and reactions with water, J. Am. Chem. Soc., 138, 14409-14422, 10.1021/jacs.6b08655, 2016.

Mendes, J., Zhou, C. W., and Curran, H. J.: Theoretical chemical kinetic study of the H-atom abstraction reactions from aldehydes and acids by H atoms and OH, $HO_2$, and $CH_3$ radicals, J. Phys. Chem. A, 118, 12089-12104, 10.1021/jp5072814, 2014.

Novelli, A., Hens, K., Ernest, C. T., Martinez, M., Nölscher, A. C., Sinha, V., Paasonen, P., Petäjä, T., Sipilä, M., Elste, T., Plass-Dülmer, C., Phillips, G. J., Kubistin, D., Williams, J., Vereecken, L., Lelieveld, J., and Harder, H.: Identifying Criegee intermediates as potential oxidants in the troposphere, Atmos. Chem. Phys. Discuss., 10.5194/acp-2016-919, 2016.

Novelli, A., Hens, K., Ernest, C. T., Martinez, M., Nölscher, A. C., Sinha, V., Paasonen, P., Petäjä, T., Sipilä, M., Elste, T., Plass-Dülmer, C., Phillips, G. J., Kubistin, D., Williams, J.,

Vereecken, L., Lelieveld, J., and Harder, H.: Estimating the atmospheric concentration of Criegee intermediates and their possible interference in a FAGE-LIF instrument, Atmos. Chem. Phys., 17, 7807-7826, 10.5194/acp-17-7807-2017, 2017.

Ryzhkov, A. B., and Ariya, P. A.: The importance of water clusters $(H_2O)_n$ (n=2, ... ,4) in the reaction of Criegee intermediate with water in the atmosphere, Chem. Phys. Lett., 419, 479-485, 10.1016/j.cplett.2005.12.016, 2006.

Taatjes, C. A., Welz, O., Eskola, A. J., Savee, J. D., Scheer, A. M., Shallcross, D. E., Rotavera, B., Lee, E. P. F., Dyke, J. M., Mok, D. K. W., Osborn, D. L., and Percival, C. J.: Direct measurements of conformer-dependent reactivity of the Criegee intermediate $CH_3CHOO$, Science, 340, 177-180, 10.1126/science.1234689, 2013.

Zhang, P., Wang, W. L., Zhang, T. L., Chen, L., Du, Y. M., Li, C. Y., and Lü, J.: Theoretical study on the mechanism and kinetics for the self-reaction of C2H5O2 radicals, J. Phys. Chem. A, 116, 4610-4620, 10.1021/jp301308u, 2012.